# Multi-agent imitation learning with function approximation: Linear Markov games and beyond

**Luca Viano** [* 1]  **Till Freihaut** [* 2]  **Emanuele Nevali** [1]  **Volkan Cevher** [1]  **Matthieu Geist** [3]  **Giorgia Ramponi** [2]

## Abstract

In this work, we present the first theoretical analysis of multi-agent imitation learning (MAIL) in linear Markov games where both the transition dynamics and each agent's reward function are linear in some given features. We demonstrate that by leveraging this structure, it is possible to replace the state-action level *all policy deviation concentrability coefficient* (Freihaut et al., 2025b) with a concentrability coefficient defined at the feature level which can be much smaller than the state-action analog when the features are informative about *states' similarity*. Furthermore, to circumvent the need for any concentrability coefficient, we turn to the interactive setting. We provide the first, computationally efficient, interactive MAIL algorithm for linear Markov games and show that its sample complexity depends only on the dimension of the feature map $d$. Building on these theoretical findings, we propose a deep MAIL interactive algorithm which clearly outperforms BC on games such as Tic-Tac-Toe and Connect4.

## 1. Introduction

Recently, there has been growing interest in multi-agent imitation learning (MAIL) (Ramponi et al., 2023; Tang et al., 2024; Freihaut et al., 2025a;b). To ensure non-exploitable learning, these works focus on equilibrium solutions rather than value-based gaps. The consensus across this literature is that achieving approximate equilibrium behavior by learning from expert data is significantly harder than merely recovering the experts' value. For instance, Tang et al. (2024) provides the first hardness result and an error propagation analysis for $N$-player Markov games, showing that non-interactive MAIL can fail to minimize the ex-

ploitability of agents even when recovering the expert value is successful. This hardness has been further quantified by Freihaut et al. (2025a;b), who provide the first sample complexity analysis for this setting. They establish the necessity of an *all policy deviation concentrability coefficient* $\mathcal{C}_{\max}$ for all non-interactive MAIL algorithms and subsequently introduce computationally and statistically efficient algorithms for interactive MAIL that bypass the dependence on $\mathcal{C}_{\max}$. However, these contributions are restricted to tabular Markov games and are therefore applicable only to small state and action spaces. In this work, we take a significant step forward in studying function approximation in the context of MAIL, rendering our algorithms applicable to settings requiring neural network function approximation.

To facilitate the transfer to (linear) function approximation settings, it is crucial to understand the limitations of MAIL in the tabular case, particularly regarding the hardness quantity $\mathcal{C}_{\max}$. Intuitively, $\mathcal{C}_{\max}$ measures the distributional shift between the states visited by a best response to an arbitrary policy and those covered by the observed Nash equilibrium. While the necessity of $\mathcal{C}_{\max}$ was derived by Freihaut et al. (2025b), their lower bounds rely on examples where a state outside the support of the observed Nash equilibrium cannot be connected to a similar state within the support of the expert data. Consequently, it remains an open question whether an agent that intelligently abstracts the state space can transfer the knowledge learned in observed states to similar but unseen ones, thereby bypassing the dependency on $\mathcal{C}_{\max}$. Furthermore, the rate-optimal learning algorithm for interactive MAIL proposed by Freihaut et al. (2025b) relies on a reward-free warm-up phase, and it is unclear if this approach remains compatible with the function approximation setting. A detailed discussion of related work is provided in Appendix B.

To address these open questions and extend MAIL to the (linear) function approximation setting, we present the following contributions. **(1)** In the setting of linear Markov games, we demonstrate the emergence of a new concentrability coefficient $\mathcal{C}_{\varphi,\max}$ in the behavior cloning upper bound. This coefficient depends on the feature map $\varphi$ and allows an agent to generalize across states given well-crafted features. Most importantly, it is upper bounded by and po-

---

[1]EPFL [2]University of Zurich [3]Earth Species Project. Correspondence to: Luca Viano <luca.viano@epfl.ch>, Till Freihaut <freihaut@ifi.uzh.ch>.

*Proceedings of the 43rd International Conference on Machine Learning*, Seoul, South Korea. PMLR 306, 2026. Copyright 2026 by the author(s).

tentially much smaller than $\mathcal{C}_{\max}$. **(2)** We present the first interactive MAIL algorithm for the linear function approximation setting. The key ingredient is a novel reward-free warm-up phase that scales effectively to function approximation settings and extends to the infinite horizon setting with minor adjustments. **(3)** Inspired by the linear function approximation results, we propose a deep MAIL algorithm for competitive environments and provide experimental evaluations for both the linear and deep MAIL settings.

## 2. Preliminaries

This work focuses on the setting of Markov games. Notably, our results apply to $N$ player general-sum games, however for clarity of presentation in the main text, we focus on stationary two-player zero-sum Markov games (MGs) introduced next.

**Finite Horizon two player zero-sum Markov games.** In general, a Markov game is defined as the tuple $\mathcal{G} = (\mathcal{X}, \mathcal{A}, H, r, P, \nu_1)$, where $\mathcal{X}$ is the state space, $\mathcal{A} = \mathcal{A}^1 \times \mathcal{A}^2$ is the joint action space, which is the product space of the individual action spaces $\mathcal{A}^1, \mathcal{A}^2$, $H$ is the horizon, and $P : \mathcal{X} \times \mathcal{A}^1 \times \mathcal{A}^2 \to \Delta_{\mathcal{X}}$ is a transition kernel. In other words, $P(x' \mid x, a^1, a^2)$ denotes the probability of landing in $x'$ from $x$ under the action pair $a^1, a^2$ and a reward $r^n : \mathcal{X} \times \mathcal{A}^1 \times \mathcal{A}^2 \to \mathbb{R}$ for $n = 1, 2$ with $r^n(x, a^1, a^2) \in [-1, 1]$. Additionally, we denote the initial state distribution $\nu_1 \in \Delta_{\mathcal{X}}$. Moreover, we denote the space of (possibly non-stationary) policies for each player as $\Pi^n : \mathcal{X} \times [H] \to \Delta_{\mathcal{A}^n}$ for $n = 1, 2$. In particular, a non stationary policy is a collection of $H$ mappings from states to distributions over $\mathcal{A}$, i.e. $\pi^n = \{\pi_1^n, \ldots, \pi_H^n\}$ where $\pi_h^n(a|x)$ is the probability that player $n$ chooses action $a$ in state $x$ when this state is visited after $h$ steps. Since we assume that the game is *zero-sum*, we have that $r^1(x, a^1, a^2) = -r^2(x, a^1, a^2)$ for all $x, a^1, a^2 \in \mathcal{X} \times \mathcal{A}^1 \times \mathcal{A}^2$. We will also find convenient to use the game theoretic $-n$ notation to denote all the players but the one indexed by $n$. Clearly, in the two players setting $\pi^{-1} = \pi^2$ and conversely. The maximal cardinality of the action spaces is denoted by $A_{\max} := \max_n |\mathcal{A}^n|$.

**Policies, occupancy measures and value functions.** Then, for any policy pair $(\pi^1, \pi^2)$ let us call a *trajectory* the stochastic process $(X_1, A_1^1, A_1^2, \ldots, X_H, A_H^1, A_H^2)$ where $X_1 \sim \nu_1$, $A_h^1 \sim \pi_h^1(\cdot|X_h)$, $A_h^2 \sim \pi_h^2(\cdot|X_h)$ and $X_{h+1} \sim P(\cdot|X_h, A_h^1, A_h^2)$ for each $h \in [H]$. Furthermore, let us define the state visitation distribution at stage $h \in [H]$ and state $x \in \mathcal{X}$ induced by a policy pair $\pi^1, \pi^2$ as

$$\nu_h^{\pi^1, \pi^2}(x) := \mathbb{P}_{\pi^1, \pi^2}[X_h = x].$$

Similarly, we define the state actions occupancy measure as

$$\mu_h^{\pi^1, \pi^2}(x, a^1, a^2) := \mathbb{P}_{\pi^1, \pi^2}[X_h = x, A_h^1 = a^1, A_h^2 = a^2].$$

Moreover, we define the state value function of the policy pair $\pi^1, \pi^2$ for the player indexed by $n \in \{1, 2\}$ as

$$V_{n,h}^{\pi^1, \pi^2}(x) = \mathbb{E}_{\pi^1, \pi^2}\left[\sum_{h'=h}^{H} r^n(X_{h'}, \{A_{h'}^n\}_{n=1}^2) \mid X_h = x\right].$$

Moreover, we define the state action value function for the $n^{\text{th}}$ player denoted as $Q_{n,h}^{\pi^1, \pi^2}(x, a^n)$ as

$$\mathbb{E}_{\pi^1, \pi^2}\left[\sum_{h'=h}^{H} r^n(X_{h'}, \left\{A_{h'}^{n'}\right\}_{n'=1}^2) \mid X_h = x, A_h^n = a^n\right].$$

Additionally, we adopt the convention that when $h = 1$ we drop the stage subscript. In particular, we have that $V_n^{\pi^1, \pi^2} := V_{n,1}^{\pi^1, \pi^2}$ and $Q_n^{\pi^1, \pi^2} := Q_{n,1}^{\pi^1, \pi^2}$ for all $n \in \{1, 2\}$. Next, we can introduce our solution concept which is the one of Nash Equilibria.

**Nash Equilibria.** A policy profile $\pi_{\text{NE}}^1, \pi_{\text{NE}}^2$ is called a Nash equilibrium if no agent has interest in deviating from those strategies. In particular, we have that $\pi_{\text{NE}} := \pi_{\text{NE}}^1, \pi_{\text{NE}}^2$ is a $\varepsilon$-approximate Nash equilibrium if

$$\text{NG}(\pi_{\text{NE}}) := \max_{n \in \{1,2\}} \max_{\pi^n \in \Pi^n} \left\langle \nu_1, V_n^{\pi^n, \pi_{\text{NE}}^{-n}} - V_n^{\pi_{\text{NE}}} \right\rangle \le \varepsilon.$$

where NG stands for Nash gap. If $\varepsilon = 0$, then $\pi_{\text{NE}}$ is an (exact) Nash equilibrium. Following standard terminology, we will often refer to the left hand side above as the exploitability or Nash gap. Additionally, we introduce a best response set to a given policy $\pi^n$ for $n \in \{1, 2\}$ by

$$\text{br}(\pi^n) := \arg\max_{\pi^{-n} \in \Pi^{-n}} \langle \nu_1, V_n^{\pi^n, \pi^{-n}} \rangle.$$

Note that in a Nash equilibrium both policies are best responses to each other.

### 2.1. Linear Markov games

In order to design computationally and statistically efficient algorithms under this setting we make the following structural assumption on the reward and the transitions of the game (Cui et al., 2023; Zhong et al., 2022; Wang et al., 2023). Our assumption is, however, significantly weaker than the ones in prior works because we require linearity of transitions and rewards averaged by the expert policy. In contrast, prior works required linearity when the averaging policy can be chosen as any Markov policy (Cui et al., 2023) or any linear Markov policy (Wang et al., 2023).

**Assumption 2.1** (Stationary linear MGs)**.** For each player $n \in \{1, 2\}$, there exists a known $d$-dimensional feature mapping $\varphi : \mathcal{X} \times \mathcal{A}^n \to \mathbb{R}$, such that for any state $x \in \mathcal{X}$ and action $a \in \mathcal{A}^n$,

$$\sum_{a^{-n}\in\mathcal{A}^{-n}} \pi_E^{-n}(a^{-n} \mid x)P(x' \mid x,a,a^{-n}) = \varphi_n(x,a)^\top M_{-n}(x')$$

$$\sum_{a^{-n}\in\mathcal{A}^{-n}} \pi_E^{-n}(a^{-n} \mid s)r^n(x,a,a^{-n}) = \varphi_n(x,a)^\top w_{-n}$$

where $M_{-n} : \mathcal{X} \to \mathbb{R}^d$ and $w_{-n} \in \mathbb{R}^d$ are unknown to the players. Following the convention of Jin et al. (2019); Luo et al. (2021), we assume $\|\varphi(x,a)\|2 \le 1 \;\; \forall x \in \mathcal{X}, a \in \mathcal{A}^n, n \in \{1,2\}$ and that $\max_{n\in\{1,2\}} \max(\|M_{-n}\|_2, \|w_{-n}\|_2) \le B$ for some $B \ge 1$. Last, as we are in a zero-sum setting, we have $\varphi_1(x,a^1)^\top w_2 = r^1(x,a^1,a^2) = -r^2(x,a^1,a^2) = -\varphi_2(x,a^2)^\top w_1$.

To develop some intuition about the setting, we introduce an important particular case: Tabular Markov games, which have been considered in the previous closest related work (Freihaut et al., 2025a;b).

*Example* 2.2. **Tabular MGs are Linear MGs:** Let $d_1 = |\mathcal{X}| |\mathcal{A}^1|$, $d_2 = |\mathcal{X}| |\mathcal{A}^2|$ and $\varphi_1(x,a^1) = e_{(x,a^1)}$, $\varphi_2(x,a^2) = e_{(x,a^2)}$ be the canonical basis in $\mathbb{R}^{d_1}$ and $\mathbb{R}^{d_2}$, then we recover tabular Markov games. For the detailed argument, we point the reader to Cui et al. (2023, Example 1).

Note that Assumption 2.1 is also known as the *independent linear Markov game* setup which has previously been studied by Cui et al. (2023). While we consider only two players in the main text, our results transfer to the $N$-player setting[1]. Additionally, this assumption comes natural for Imitation Learning as each player is learning their policy individually, seeing the other agents as a part of the environment. The other potential assumption is a *global* function approximation setting, where the feature map depends on all agents $\varphi(x,a^1,a^2)$, see e.g. (Xie et al., 2020; Huang et al., 2021). However, translating this setting to the tabular case results in algorithms that scale exponentially in the number of players, which is known as the *curse of multi-agents*.

**Linearity of state action value functions.** An important consequence of Assumption 2.1 is that the state action value functions of all players are linear. This means that for policies $\pi^1, \pi^2$ and each player index $n \in \{1,2\}$ and stage $h \in [H]$, there exists an unknown vector $\theta_{n,h}^{\pi^1,\pi^2}$ such that[2]

$$Q_{n,h}^{\pi^1,\pi^2}(x,a) = \varphi(x,a)^\top \theta_{n,h}^{\pi^1,\pi^2},$$

and $\theta_{n,h}^{\pi^1,\pi^2} \in \Theta$ with $\Theta = \{\theta \in \mathbb{R}^d \mid \|\theta\| \le B_\theta\}$ for some scalar $B_\theta$. We will make crucial use of this fact in designing the decision space of our algorithms.

---

## 2.2. Imitation Learning in MGs.

In Imitation Learning in MGs the learning algorithm is tasked with learning an approximate Nash equilibrium without knowing the reward function, the transition dynamics and the initial distribution of the underlying Markov game. The only way that the learner can access information about an *expert* Nash equilibrium denoted via $\pi_E = (\pi_E^1, \pi_E^2)$ is via a state action dataset $\mathcal{D}_E = \{\mathcal{D}_E^1, \mathcal{D}_E^2\}$. We distinguish two different settings for the generation of $\mathcal{D}_E$.

**Non-interactive:** The learning algorithm receives a dataset $\mathcal{D}_E = \{X_{E,h}^i, \{A_{E,h}^{n,i}\}_{n=1}^2\}_{i=1}^{\tau_E}$ where each triplet is sampled from the expert occupancy measure $X_{E,h}^i, \{A_{E,h}^{n,i}\}_{n=1}^2 \sim \mu_h^{\pi_E^1,\pi_E^2}$ for all $i \in [\tau_E]$. Notation wise, we denote the datasets $\mathcal{D}_E^1 = \{X_{E,h}^i, A_{E,h}^{1,i}\}_{i=1}^{\tau_E}$ and $\mathcal{D}_E^2 = \{X_{E,h}^i, A_{E,h}^{2,i}\}_{i=1}^{\tau_E}$ which contain the joint states and only the first and second player actions respectively.

**Interactive MAIL:** The learning algorithm can query the expert Nash profile at any state encountered while interacting in the game.

The interactive setting is a much more powerful access model to the expert policy as it does not require that the states in which the expert is queried belong to the support of the expert state occupancy measure $\nu^{\pi_E^1,\pi_E^2}$.

## 3. Main result for non-interactive setting

In this section we present our algorithms and main results for the non-interactive setting. To ensure our algorithms can be implemented efficiently, we assume the expert dataset $\mathcal{D}_E$ is generated by a Nash equilibrium which can be recovered as the limit of the Nash equilibrium in a regularized game. Specifically, this equilibrium is obtained by augmenting each player's payoff with a strongly concave function (e.g., entropy) weighted by a regularization parameter $1/\eta$[3] (to be introduced shortly). In order to formally state the assumption we introduce, for $n \in \{1,2\}$, the policy class $\Pi_{\text{softlin},h}^n$ defined as

$$\left\{ \exists \theta_{n,h} \in \Theta \mid \pi_h^n(a \mid x) = e^{\eta\,\varphi(x,a)^\top \theta_{n,h} - \zeta_{n,h}(x)} \right\}$$

where $\zeta_{n,h}(x) = \log \sum_{a'\in\mathcal{A}^n} \exp\big(\eta\,\varphi(x,a')^\top \theta_{n,h}\big)$ ensure proper normalization of the policies in the class. Moreover, let us introduce the notations

$$\Pi_{\text{softlin}}^n = \mathop{\times}_{h=1}^{H} \Pi_{\text{softlin},h}^n, \;\; \Pi_{\text{softlin}} = \mathop{\times}_{n=1}^{N} \Pi_{\text{softlin}}^n,$$

$$\Pi_{\text{softlin}}^{-n} = \mathop{\times}_{n'=1,n'\neq n}^{N} \Pi_{\text{softlin}}^{n'}.$$

---

For the two player case, we set $N = 2$ in the previous definitions. We are now ready to present our assumption on the expert NE.

**Assumption 3.1.** We assume that the expert policy collecting the data $(\pi_E^1, \pi_E^2)$ is in the set of limit points of the set $\Pi_{\text{softlin}}$. That is, $\pi_{E,h}^n \in \lim_{\eta \to \infty} \Pi_{\text{softlin}}^n$ for all players, i.e. for $n \in \{1, 2\}$.

The above assumption says that the features of the linear MG should be expressive enough to approximately realize the expert policy. However, our bounds for the non-interactive setting will depend also on the concentrability coefficient (see Definition 3.2), that requires the features to be *not too expressive*, in the sense that they should not carry information about the state action pair which are irrelevant for describing the state action value function. To make the claim formal, we define the concentrability coefficient as follows.

**Definition 3.2. Features concentrability coefficient.** For each player $n \in \{1, 2\}$, we define the features expectation vector,

$$\varphi_h^{\pi_\star^n, \pi_E^{-n}} = \sum_{x', a^n} \varphi(x', a^n) \pi_{\star,h}^n(a^n|x') \nu_h^{\pi_\star^n, \pi_E^{-n}}(x'),$$

and the $n^{\text{th}}$-player features level concentrability coefficient,

$$\mathcal{C}_{\varphi,\max}^n = \max_{h \in [H]} \max_{\pi^{-n} \in \Pi_{\text{softlin}}^{-n}} \max_{\pi_\star^n \in \text{br}(\pi^{-n})} \left\| \varphi_h^{\pi_\star^n, \pi_E^{-n}} \right\|_{(\Lambda_{E,h}^n)^{-1}},$$

where the expert features covariance matrix $\Lambda_{E,h}^n$ for the $n^{\text{th}}$ player is defined as

$$\Lambda_{E,h}^n = \mathbb{E}\left[ \varphi(X_{E,h}, A_{E,h}^n) \varphi(X_{E,h}, A_{E,h}^n)^\top \right] + \lambda I,$$

for $X_{E,h} \sim \nu_h^{\pi_E}, A_{E,h}^n \sim \pi_{E,h}^n(\cdot|X_{E,h})$ and $\lambda \geq 0$. Finally, we define the features concentrability coefficient as the maximum between the concentrability of each player. Formally, we get $\mathcal{C}_{\varphi,\max} := \max\{\mathcal{C}_{\varphi,\max}^{\pi_1}, \mathcal{C}_{\varphi,\max}^{\pi_2}\}$.

Next, we can present behavioral cloning and a new sample complexity analysis which will incur only a dependence on $\mathcal{C}_{\varphi,\max}$ instead of the (always equal or larger) state action concentrability coefficient $\mathcal{C}_{\max}$ defined by Freihaut et al. (2025a) as

$$\mathcal{C}_{\max} := \max_{n \in \{1,2\}} \max_{h \in [H]} \max_{\pi^{-n} \in \Pi_{\text{softlin}}^{-n}} \max_{\pi_\star^n \in \text{br}(\pi^{-n})} \left\| \frac{\nu_h^{\pi_\star^n, \pi_E^n}}{\nu_h^{\pi_E}} \right\|_\infty.$$

**Behavioral Cloning.** For each player, Behavioral Cloning (BC) outputs the policy which maximizes the likelihood of the observed actions for that player. That is, $\pi_{\text{out}}^n = \arg\max_{\pi \in \Pi_{\text{softlin}}^n} \sum_{i=1}^{\tau_E} \sum_{h=1}^H \log \pi_h(A_{E,h}^{i,n}|X_{E,h}^n)$ for $n = \{1, 2\}$, and enjoys the following guarantees.

**Theorem 3.3.** *Main result: non-interactive case.* Let the feature concentrability coefficient and the the expert features covariances be given as defined in Defintion 3.2 with $\lambda = \tau_E^{-1}$, where $\tau_E$ is the total number of trajectories. Furthermore, assume that the Nash equilibrium that collected the dataset $\mathcal{D}_E$ satisfies Assumption 3.1. Then, the output of BC, ran over the class $\Pi_{\text{softlin}}$ with $\eta = \log \tau_E / H$, is an $\varepsilon$-approximate Nash equilibrium with probability of at least $1 - \delta$ if the expert dataset $\mathcal{D}_E$ contains $\tau_E = \tilde{\mathcal{O}}\left( \frac{H^5 \mathcal{C}_{\max,\varphi}^2 dB^2 \log(A_{\max} B_\theta \delta^{-1})}{\varepsilon^2} \right)$ many trajectories.

At a technical level, the most important observation is to exploit the linearity of the state occupancy measures in $M_{-n}$ to perform a change of measure argument at features level rather than at the state action level. The formal statement of this step is in Lemma D.6. We make some important remarks in the following.

**Rate optimality and necessity of $\mathcal{C}_{\varphi,\max}$.** Despite its simplicity, BC is known to be minimax optimal with respect to $\varepsilon$ and that the polynomial dependence in $\mathcal{C}_{\varphi,\max}$ is necessary for each non-interactive algorithm. This fact has been shown for BC in tabular Markov games by Freihaut et al. (2025b) and such result continues to hold for Linear MGs under the light of Example 2.2. Indeed, under one hot features, the choice of $\lambda = 0$ and deterministic occupancy measure $\mu_h^{\pi_\star^n, \pi_E^{-n}}$ for all $n \in \{1, 2\}$[4] we have that $\mathcal{C}_{\varphi,\max}$ reduces to $\mathcal{C}_{\max}$ whose necessity is implied by Freihaut et al. (2025b, Theorem 3.1).

**Which are good features for BC?** On the positive side, we notice that for some features, beyond the tabular case, $\mathcal{C}_{\varphi,\max}$ can be much smaller than $\mathcal{C}_{\max}$. In order to understand why this is the case, let us recall that in the tabular case, the concentrability coefficient $\mathcal{C}_{\max}$ is infinite when there exist a deviation $\pi_\star^n$ such that when playing against the Nash policy for the other player induces an occupancy measure $\nu_h^{\pi_\star^n, \pi_E^{-n}}$ which has support on states which have zero mass under $\nu_h^{\pi_E^n, \pi_E^{-n}}$. Under tabular features, there is no information about how to act in those states. However, under the linear MG structure we can observe generalization across states when the features of two states are not orthogonal. In this way, the algorithm can have information about how to act even in states which do not appear in the expert dataset if they have features which are similar enough to expert states. Such similarity is formally measured by the feature level concentrability $\mathcal{C}_{\varphi,\max}$.

Let us illustrate the concept above with a simple case of maximally similar features, i.e. $\varphi(x, a^n) = c \in (0, 1]$ for all $x \in \mathcal{X}, n \in \{1, 2\}$ and $a^n \in \mathcal{A}^n$. For simplicity, let us set

---

[4]Notice that the lower bound construction in (Freihaut et al., 2025b) satisfies these requirements.

$\lambda = 0$ since $c > 0$ ensures that the inverse covariance matrix exists and equals $(\Lambda_{E,h}^n)^{-1} = c^{-2}$. Therefore, $\mathcal{C}_{\varphi,\max} = \max_{n \in \{1,2\}} \max_{x,a^n} \|\varphi(x,a^n)\|_{(\Lambda_{E,h}^n)^{-1}} = \sqrt{cc^{-2}c} = 1$. Therefore, in this example, $\mathcal{C}_{\varphi,\max}$ attains its minimum possible value while at the same time $\mathcal{C}_{\max}$ is possibly infinite.

*However, are the constant features good features ?* The answer is clearly no. Indeed, these features are not *expressive enough* to satisfy the Linear MG and expert realizability assumption (i.e. Assumption 3.1 and Assumption 2.1) unless the next state is always sampled from the same distribution, i.e., ignoring the conditioning on current state action. All in all, we make the following observation regarding the best choice for the features.

> **Take Away:** The best features for BC are the ones that satisfy Assumption 2.1 with lowest values of $\mathcal{C}_{\varphi,\max}$, i.e. the ones that allow the best generalization across states, while being expressive enough to realize the Linear MG assumption.

Moreover, the best features are usually not one-hot encoding. Indeed, we show in our experiments (see Section 5) that it is usually possible to find features which allows BC to improve upon its own performance with tabular features.

**Is the interactive setting still needed in Linear MGs ?** Although, smaller or equal than $\mathcal{C}_{\max}$, also $\mathcal{C}_{\varphi,\max}$ can be very large or unbounded if there exists a best response which can generate an expected feature vector pointing in a direction poorly covered by the expert dataset. In such situations, it is highly desirable to deploy algorithms with concentrability-free guarantees. In light of the lower bound of Freihaut et al. (2025b), such algorithms can exist only in the interactive setting, which we introduce next.

## 4. Interactive Linear MAIL

In the last section, we have seen that BC can effectively minimize the Nash gap in linear Markov games if the features are well designed so that (i) the value of the concentrability coefficient is small and (ii) the features are expressive enough to satisfy Linear MGs and expert realizability. Coming up with good features is highly environment-dependent and far from trivial in general.

This motivates interactive imitation learning, under which we can design algorithms which effectively minimize the Nash gap even if the features level concentrability coefficient is unbounded.

### 4.1. The algorithm: LSVI-UCB-ZERO-BC

Our algorithm is built upon the scheme of combining reward-free RL and imitation learning, first introduced by Freihaut

et al. (2025b) in the tabular case. In particular, the algorithm can be composed into two main phases:

- **Exploration phase:** During this phase, we construct an *exploratory* dataset for each player $\mathcal{D}^1$ and $\mathcal{D}^2$. The goal is to collect expert actions from the perspective of player $n$ under any possible deviation of the opponent, i.e. player $-n$. Interestingly, we will present an efficient implementation of this idea via an appropriate instantiation of a no-regret learner rather than looping explicitly over all the possible deviations, as done by Tang et al. (2024), which is computationally inefficient.

- **Imitation Phase:** After collecting $\mathcal{D}^n$ for $n \in \{1,2\}$, we simply apply behavioral cloning on this newly created datasets.

Before moving to a more detailed description of each phase, we state the sample complexity guarantees for the resulting algorithm given in Algorithm 1.

**Theorem 4.1.** *Main result: interactive case. Let $\pi_{\text{out}} = (\pi_{\text{out}}^1, \pi_{\text{out}}^2)$ be the output of Algorithm 1 ran setting $K = \tilde{\mathcal{O}}\left(\frac{H^6 d^4 B^4 \log(A_{\max} B_\theta \delta^{-1})}{\varepsilon^2}\right)$, then $\pi_{\text{out}}$ is an $\varepsilon$-approximate Nash equilibrium with probability at least $1 - \delta$. The total number of expert queries is $KH = \tilde{\mathcal{O}}\left(\frac{H^7 d^4 B^4 \log(B_\theta A_{\max} \delta^{-1})}{\varepsilon^2}\right)$.*

The main consequence of Theorem 4.1 is explained in the following take away.

> **Take Away:** In the interactive setting, Algorithm 1 avoids the dependence on $\mathcal{C}_{\varphi,\max}$. Moreover, the sample complexity scales only with the features dimension $d$ and not with the number of states of the game.

A remaining concern can be that in long horizon problems learning non-stationary policies as in Algorithm 1 can create memory issues. For such situations, modeling the game with the discounted infinite horizon framework as done by Freihaut et al. (2025a) is more attractive.

**Extension to the infinite horizon setting.** Interestingly, the design philosophy of LSVI-UCB-ZERO-BC extends to the infinite horizon setting. Indeed, we can just replace LSVI-UCB with an algorithm that achieves optimal regret bounds in discounted linear MDPs and run it with zero reward and slightly larger bonuses. In particular, we use RMAX-RAVI-LSVI-UCB (Moulin et al., 2025b). This enables us to answer affirmatively the open question of Freihaut et al. (2025b) of achieving rate optimal bounds in the discounted setting. For this extension, we need to assume

**Algorithm 1** LSVI-UCB-ZERO-BC

> **Input:** Horizon $H$, Number of episodes $K$.
> **Output:** Learned policy pair $\pi_{\text{out}} = (\pi_{\text{out}}^1, \pi_{\text{out}}^2)$.
> **% Loop over players:**
> **for** $n \in \{1, 2\}$ **do**
>   **% Exploration Phase:**
>   Define the MDP $\mathcal{M}_n$ where the policy of player $n$ is fixed to $\pi_E^n$.
>   Create the dataset for player $n$, updating the policy of player $-n$:
>
> $$\mathcal{D}^n \leftarrow \text{LSVI-UCB-ZERO}(\mathcal{M}_n, \texttt{active} = -n).$$
>
>   **% Imitation Phase:**
>   Run BC on dataset $\mathcal{D}^n$:
>
> $$\pi_{\text{out}}^n \leftarrow \arg\min_{\pi^n \in \Pi_{\text{softlin}}^n} \sum_{(X,A) \in \mathcal{D}^n} -\log \pi^n(A|X)$$
>
> **end for**

**Algorithm 2** LSVI-UCB-ZERO $(\mathcal{M}_n, \texttt{active} = -n)$

> Set $\beta = \widetilde{\mathcal{O}}(dH \log(1/\delta))$
> **for** $k = 1, \dots, K$ **do**
>   **for** $h = 1, \dots, H$ **do**
>     $X_h^{-n,k} \sim \nu_h^{\pi^{-n,k}\pi_E^n}, A_h^{-n,k} \sim \pi_h^{-n,k}(X_h^{-n,k})$
>     $A_{E,h}^{n,k} \sim \pi_{E,h}^n.$
>   **end for**
>   $V_{H+1}^k(\cdot) \leftarrow 0, \Lambda_{-n,h}^0 = I.$
>   **for** $h = H, \dots, 1$ **do**
>     $\varphi_h^{-n,k} = \varphi(X_h^{-n,k}, A_h^{-n,k}).$
>     $\Lambda_h^{-n,k} = \Lambda_h^{-n,k-1} + \varphi_h^{-n,k}(\varphi_h^{-n,k})^T.$
>     $w_h^k = (\Lambda_h^k)^{-1} \sum_{\tau=1}^{k-1} \varphi_{-n,h}^\tau \cdot V_h^k(X_h^{-n,\tau}).$
>     **% Update with zero reward and larger bonus**
>
> $$Q_h^k(x,a) = \min\{H, 0 + (w_h^k)^\top \varphi(x,a) + (\beta + 1) \|\varphi(x,a)\|_{(\Lambda_h^{-n,k})^{-1}}\}$$
>
>     $V_h^k(s) \leftarrow \max_{a \in \mathcal{A}^{-n}} Q_h^k(x,a).$
>     $\pi_h^{-n,k}(x) \leftarrow \arg\max_{a \in \mathcal{A}^{-n}} Q_h^k(x,a).$
>   **end for**
> **end for**
> **Return:** expert dataset $\mathcal{D}^n = \left\{ X_h^{-n,k}, A_{E,h}^{n,k} \right\}_{k=1}^K.$

that $\langle \varphi, \mathbf{1} \rangle = 1$, which can be achieved by features normalization. The formal results are presented in Appendix G.

We now describe each phase in detail.

**The exploration phase.** In prior MAIL works (Freihaut et al., 2025b), the exploration phase is handled instantiating $XH$ different RL problems each of which is responsible to learn how to reach a particular state at a specific stage, if possible. While this approach can be generalized to the linear setting (Wagenmaker et al., 2022), we opt for a simpler alternative explained next. In particular, recall that in this phase the agent $-n$ acts in a linear MDP induced by fixing the policy of the opponent, i.e. of the $n^{\text{th}}$ player, to a Nash $\pi_E^n$. Then, we can use a no-regret algorithm for linear MDPs instantiated for the reward sequence $\{r_h^{-n,k}(x, a^{-n})\}_{k=1}^K$ such that $r_h^{-n,k}(x, a^{-n}) = \|\varphi(x, a^{-n})\|_{(\Lambda_h^{-n,k})^{-1}}$ where $\Lambda_h^{-n,k}$ is the covariance matrix of the data collected so far (see Algorithm 2 for the detailed definition). We emphasize that this reward sequence is synthetic and used only for exploration, independent of the reward of the underlying MG. This idea is inspired by the first reward-free exploration algorithm for linear MDPs (Wang et al., 2020b). We dubbed this procedure LSVI-UCB-ZERO because running LSVI-UCB with the rewards described above can be seen as running LSVI-UCB with zero everywhere reward and (slightly) larger bonus. Indeed, as highlighted in the update for $Q_h^k$ in Algorithm 2 the weighted feature norm that serves as bonus is multiplied by $(\beta + 1)$ rather than $\beta$ as in the standard analysis of LSVI-UCB (Jin et al., 2019). This name takes inspiration from UCB-ZERO (Zhang et al., 2020), which made the same observation in the tabular setting in the context of

reward-agnostic RL. Algorithm 2 gives the pseudocode.

Notice that while running this phase, we need interactive access to the expert $\pi_E^n$. Indeed, let us assume that player 1 is learning while the second player's policy is fixed to $\pi_E^2$. Then, at each round $k$ and stage $h$ from the current state[5] $X_h^{1,k}$ we sample an action from the no regret learner, i.e. $A_h^{1,k}$ and one action from the expert opponent policy denoted via $A_{E,h}^{2,k} \sim \pi_{E,h}^2(\cdot | X_h^{1,k})$. Finally, the next state is sampled as $X_{h+1}^{1,k} \sim P(\cdot | X_h^{1,k}, A_h^{1,k}, A_{E,h}^{2,k})$.

Before leaving this phase we generate the dataset to be imitated by the second player at stage $h$ as $\mathcal{D}_h^2 = \{X_h^{1,k}, A_{E,h}^{2,k}\}_{k=1}^K$. This dataset is passed to the next phase while the action sequences $\{A_h^{1,k}\}_{k,h}$ is discarded. The analogous procedure with inverted roles is run for the other player.

**The imitation phase.** In the imitation phase, the player with index $n$, maximizes the likelihood of the dataset $\mathcal{D}^n = \cup_{h=1}^H \mathcal{D}_h^n$,

$$\pi_{\text{out}}^n = \arg\max_{\pi \in \Pi_{\text{softlin}}^n} \sum_{k=1}^K \sum_{h=1}^H \log \pi_h(A_{E,h}^{n,k} | X_h^{-n,k}).$$

---

[5] The superscript 1 hereafter denotes the fact that $X_h^{1,k}$ is the state reached at stage $h$ of episode $k$ when the player 1 is updating the policies and 2 is frozen and follows the expert policy.

The procedure exactly matches the BC procedure but with the dataset generated by the exploration phase. The full two-phase algorithm is outlined in Algorithm 1.

*Remark* 4.2. Notice that Algorithms 1,2 are presented here for two players games, therefore the notation $-n$ stands for a single player in this case. For games with more than two players those algorithms are not directly applicable but the same conceptual ideas apply. We present the algorithm for the $N$ players case in Algorithm 3 in Appendix F.

Having described the algorithm, we present a sketch of the analysis in the next section.

### 4.2. Analysis Sketch

While the algorithmic components are similar to the ones of the tabular case, the required analysis differs significantly. In the tabular case (Freihaut et al., 2025b), the exploration dataset covered all significant states, which enabled a change of measure at states level and then enabled an application of classical BC results for the second case. This time we have to perform the change of measure at features level in order to avoid dependence on the number of states. In particular, for each player $n \in \{1, 2\}$ the change of measures will depend on the covariance feature matrices $\Lambda_h^{-n,K}$ for $n = 1, 2$ constructed after $K$ iterations of Algorithm 2.

**Lemma 4.3.** *Let $\Lambda_h^{-n,K}$ be the output covariance matrix and $\{\pi_k^{-n}\}_{k=1}^{K}$ be the sequence of policies generated by Algorithm 2 invoked with* active $= -n$, *i.e.* $\Lambda_h^{-n,K} = \sum_{k=1}^{K} \varphi(X_h^{-n,k}, A_h^{-n,k}) \varphi(X_h^{-n,k}, A_h^{-n,k})^\top + I$. *Then, the Nash gap for $\pi_{\mathrm{out}}$, the output policy of Algorithm 1, can be upper bounded, with probability at least $1 - \delta$, as*

$$\mathrm{NG}(\pi_{\mathrm{out}}) \leq H \sum_{h=1}^{H} \max_{\substack{n \in \{1,2\} \\ \pi^{-n} \in \Pi^{-n}}} \left\| \varphi_h^{\pi_E^n, \pi^{-n}} \right\|_{(\Lambda_h^{-n,K})^{-1}}$$
$$\cdot \mathcal{O}\left( \sqrt{\Delta\mathrm{TV}_{n,h}^2(\pi_{\mathrm{out}}) + B^2 d \log(K/\delta)} \right)$$

*where $\Delta\mathrm{TV}_{n,h}^2(\pi_{\mathrm{out}})$ is defined as follows*

$$\sum_{k=1}^{K} \sum_{x \in \mathcal{X}} \nu_h^{\pi_E^n, \pi^{-n,k}}(x) \mathrm{TV}^2(\pi_{\mathrm{out},h}^n, \pi_{E,h}^n)(x),$$

*and* $\mathrm{TV}$ *is the total variation (for a definition see Table A).*

We notice that Lemma 4.3 avoids completely the dependence on $\mathcal{C}_{\varphi,\max}$ replacing it by $\max_{\pi^{-n} \in \Pi^{-n}} \left\| \varphi_h^{\pi_E^n, \pi^{-n},} \right\|_{(\Lambda_h^{-n,K})^{-1}}$. The main difference between the two quantities is that in $\Lambda_h^{-n,K}$ the features are evaluated at the states and actions sampled while running LSVI-UCB-ZERO and not via Nash vs Nash dynamics. Notably, we can control the latter quantity

by proving that when the matrix $\Lambda_h^{-n,K}$ is constructed via LSVI-UCB-ZERO, the $\Lambda_h^{-n,K}$-weighted norm of any expected feature vector decreases at a $\mathcal{O}(K^{-1/2})$ rate.

**Lemma 4.4.** *For any $n, h \in \{1, 2\} \times [H]$, let $\Lambda_h^{-n,K}$ be the output covariance matrix after $K$ iterations of Algorithm 2 with* active $= -n$. *Then with probability $1 - \delta$ it holds that*

$$\sum_{h=1}^{H} \max_{\pi^{-n} \in \Pi^{-n}} \left\| \varphi_h^{\pi_E^n, \pi^{-n}} \right\|_{(\Lambda_h^{-n,K})^{-1}} \leq \widetilde{\mathcal{O}}\left( \sqrt{\frac{d^3 H^4 B^2 \iota}{K}} \right).$$

*where $\iota = \log(dKH/\delta)$.*

At this point, the proof is concluded by invoking the guarantees for the conditional maximum likelihood estimator (MLE) under misspecification to prove the following high probability bound on $\Delta\mathrm{TV}_{n,h}^2(\pi_{\mathrm{out}})$.

**Lemma 4.5.** *For any failure probability $\delta \in (0, 1]$, let us consider $\eta = \log(K)/H$ in the definition of the policy class $\Pi_{\mathrm{softlin}}$, then for any $n, h \in \{1, 2\} \times [H]$ it holds that with probability at least $1 - \delta$, we have $\Delta\mathrm{TV}_{n,h}^2(\pi_{\mathrm{out}}) \leq \widetilde{\mathcal{O}}\left( d \log\left( K A_{\max} B_\theta \delta^{-1} \right) \right)$.*

The final result given in Theorem 4.1 follows from combining the change of measure in Lemma 4.3, the bound on the weighted norm of any feature expectation vector in Lemma 4.4 and finally the MLE guarantees in Lemma 4.5.

Having concluded the proof we illustrate in the next section how the analysis under linear function approximation can guide the design of a practical algorithm based on neural network function approximation.

### 4.3. Beyond Linar MGs: extension to Deep MAIL

Motivated by the developed theory for the linear function approximation setting, we provide a practical extension to the general function approximation setting. In particular, we provide a deep multi-agent imitation learning algorithm for competitive games with the goal to minimize the Nash gap. For the extension to the deep learning setting, we note that in Algorithm1 the exploration bonus is defined using features that realize the action value functions of any policy profile. We carry this idea over to the function approximation setting by replacing such features with the last layer of the neural network trained to predict the cumulative future exploratory reward, i.e. a *critic* network in Deep RL jargon. More precisely, the critic network is updated by running DQN (Mnih et al., 2013) on the replay buffer that contains the historical data. Importantly, this implies that we now have to deal with features that depend on historical data as the neural network gets updated over time. The full pseudo-code, dubbed DQN-Explore-BC, is given in Algorithm 5 in Appendix H, which mirrors at a conceptual level Algorithm 1 with the important difference being that the

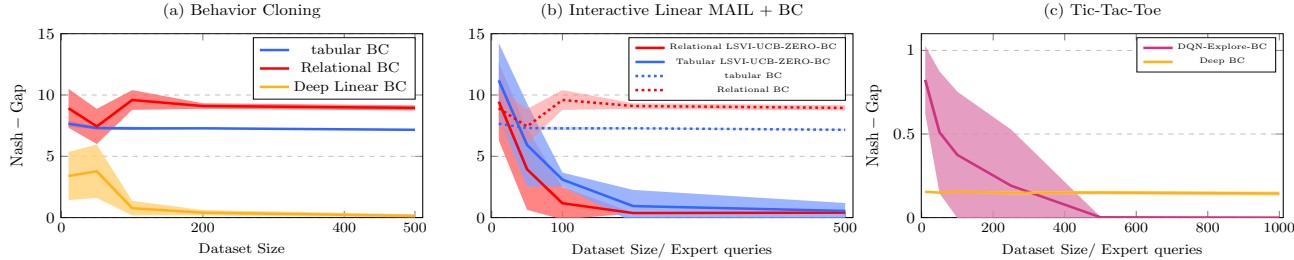

*Figure 1.* Experimental results in the linear Gridworld ((a), (b)) and deep Tic-Tac-Toe case ((c)).

*Table 1.* Win rate comparison against opponents of varying optimality. DQN-Explore-BC outperforms BC across all levels in Connect4.

| OPPONENT | DQN-EXPLORE-BC | BC |
|---|---|---|
| APPROX. BR | **0.21 ± 0.04** | 0.00 ± 0.00 |
| SOLVER NOISE 1 | **0.32 ± 0.04** | 0.00 ± 0.00 |
| SOLVER NOISE 2 | **0.60 ± 0.04** | 0.05 ± 0.01 |
| SOLVER NOISE 3 | **0.81 ± 0.01** | 0.21 ± 0.02 |
| SOLVER NOISE 4 | **0.92 ± 0.00** | 0.40 ± 0.04 |
| SOLVER NOISE 5 | **0.97 ± 0.01** | 0.54 ± 0.05 |

exploration bonus is created by reevaluating the historical data each time the features mapping changes.

## 5. Experiments

In this section, we present our experimental results. The full code is publicly available here. We first validate our theory using linear function approximation (Sections 3 and 4) before evaluating the deep extension of our algorithm (Algorithm 5).

**Experiments for linear function approximation.** We compare BC against our interactive MAIL algorithm (Algorithm 1) in the zero-sum Gridworld environment used by Freihaut et al. (2025b). The $3 \times 3$ grid contains a goal state in the top right corner; the first agent to reach it receives a reward of $+1$, while the other receives $-1$. The state space consists of 72 states (representing the joint positions of two agents, excluding overlaps) and 4 directional actions. We compute a Nash equilibrium using zero-sum value iteration (Perolat et al., 2015).

We evaluate three feature maps: tabular features ($d = 288$) to reproduce results from Freihaut et al. (2025b), relational features ($d = 80$) based on relative state similarities and a linear neural network ($d = 262$) (see Appendix I for details). Figure 1 (a) shows the results for BC. We observe that, although fitting perfectly the training set, BC fails to minimize the Nash gap with both tabular and relational features. This indicates that for both feature maps, $\mathcal{C}_{\varphi,\max}$ is large and even unbounded in the tabular case. Conversely,

the Nash gap converges to $0$ with the deep linear features, suggesting successful generalization, i.e. a small value for $\mathcal{C}_{\varphi,\max}$. While finding a feature map with finite $\mathcal{C}_{\varphi,\max}$ enables BC to succeed, identifying such maps is difficult, motivating the interactive setting. As shown in Figure 1 (b), our interactive algorithm, LSVI-UCB-ZERO-BC, effectively minimizes the Nash gap for both tabular and relational features. As predicted by our theory, convergence is faster with the lower-dimensional relational map.

**Deep experiments.** We evaluate DQN-Explore-BC (Algorithm 5) against Deep BC on Tic-Tac-Toe and Connect4. Both games are strongly solved (Kalra, 2022; Böck, 2025). Tic-Tac-Toe ends in a draw under optimal play, while Player 1 can force a win in Connect4. We use optimal solvers as expert queries. Figure 1 (c) demonstrates that while Deep BC fails to minimize the Nash gap in Tic-Tac-Toe, DQN-Explore-BC succeeds. For Connect4, where exact exploitability calculation is intractable, we evaluate the win rate of the learned Player 1 policy against opponents of varying skill levels. As shown in Table 1, DQN-Explore-BC significantly outperforms the pure deep BC. We provide further details in Appendix I.

## 6. Conclusion

In this work, we have provided the first theoretical sample complexity analysis for linear Markov games in both the non-interactive (Theorem 3.3) and interactive (Theorem 4.1) MAIL settings. We demonstrated that by leveraging the structure of the underlying game, it is possible to relax the negative results associated with the non-interactive setting of tabular Markov games (Freihaut et al., 2025b). Furthermore, we provided the first interactive MAIL algorithm for linear Markov games that extends to the infinite horizon setting, addressing an open question by Freihaut et al. (2025b). Finally, inspired by our theoretical findings, we developed a deep multi-agent imitation learning algorithm (Algorithm 5) that scales to complex environments.

Our work opens a number of interesting perspectives, discussed further in Appendix C. First, as common in other works even in the single agent case, we rely on an expert

policy realizability assumption (Assumption 3.1), and it would be interesting to weaken this by relying only on structural properties of the game. We have shown that $\mathcal{C}_{\varphi,\max}$ is a central quantity to assess the "hardness" of a game. Estimating $\mathcal{C}_{\varphi,\max}$ would therefore provide a proxy to know if BC can be safely applied or not. Related to this, our work could be extended to warm-starting from an existing dataset, calling for an adaptive amount of interactions.

## Acknowledgements

This project has received funding from the SNSF Starting Grant No PS00-2_234751. Luca Viano was supported by Swiss Data Science Center under fellowship number P22_03. Research was sponsored by the Army Research Office and was accomplished under Grant Number W911NF-24-1-0048. This work was funded by the Swiss National Science Foundation (SNSF) under grant number 2000-1-240094

## Impact Statement

This paper presents work whose goal is to advance the field of Machine Learning. There are many potential societal consequences of our work, none which we feel must be specifically highlighted here.

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

# Contents of Appendix

This appendix provides supplementary material to support the main findings of the paper.

- Appx. A contains a summary table of all the notation used throughout this work.

- Appx. B provides an extensive discussion on related works, in particular it includes *theoretical multi-agent imitation learning*, *deep multi-agent imitation learning*, *reward-free reinforcement learning* and *single-agent imitation learning*.

- Appx. C lays out interesting directions of future work.

- Appx. D provides the proofs for the non-interactive setting. The proofs are directly presented for the $N$ player general-sum setting.

- Appx. E contains the proofs for the interactive setting for the two player case.

- Appx. F extends the proofs from the two player zero-sum setting to the $N$ player general-sum setting as well as the extension to the infinite horizon setting.

- Appx. H states the algorithm for the Deep multi-agent imitation learning setting and provides some further details.

- Appx. I contains all necessary details to reproduce the experiments.

- Appx. J lists important technical Lemmas used throughout the work and provides their proofs.

# A. Notation

| Notation | Description |
|---|---|
| $[X]$ | Set of $X$ elements: $[X] := \{1, \ldots X\}$, where $X \in \mathbb{N}$ |
| $\mathrm{TV}(\cdot, \cdot)$ | Total Variation distance of two distributions $\pi^1, \pi^2$; $\mathrm{TV}(\pi^1, \pi^2) := \sum_a \left\| \pi^1(a) - \pi^2(a) \right\|$. |
| $\mathcal{G}$ | Markov game |
| $\mathcal{X}$ | joint State space |
| $\mathcal{A}^n$ | Action space of player $n \in [N]$ |
| $A_{\max}$ | Max Action space size $\max_{n \in [N]} \lvert \mathcal{A}^n \rvert$ |
| $P$ | Transition function |
| $r^n$ | Reward function of player $n$ |
| $H$ | finite time horizon |
| $\gamma$ | Discount factor |
| $\nu_1$ | Initial state distribution |
| $\Delta_{\mathcal{X}}$ | Probability simplex over state space $\mathcal{X}$ |
| $\pi^n$ | $\mathcal{X} \to \Delta_{\mathcal{A}^n}$, Policy of player $n \in [N]$ |
| $a^n$ | action of player $n \in [N]$ |
| $a^{-n}$ | action of all players but player $n \in [N]$ |
| $V_{n,h}^{\pi^n, \pi^{-n}}(x)$ | State value function of player $n$ at state $x$ and step $h$ |
| $Q_{n,h}^{\pi^n, \pi^{-n}}(x, a^n, a^{-n})$ | State-action value of player $n$ at state $x$ and step $h$ |
| $\nu_h^{\pi^n, \pi^{-n}}(x)$ | State visitation distribution of joint policy $(\pi^n, \pi^{-n})$ |
| $\mu_h^{\pi^1, \pi^2}(x, a^1, a^2)$ | State action distribution |
| $\mathrm{NG}(\pi)$ | Nash gap of joint policy $\pi$ |
| $\mathrm{br}(\pi^n)$ | Best response to policy $\pi^n$ |
| $\theta_{n,h}^{\pi^n, \pi^{-n}}$ | parameter of player $n$ at step $h$ with policy $\pi^n$, facing $\pi^{-n}$ |
| $B_\theta$ | scalar value bound on parameter norm |
| $B$ | scalar bound of $\max_{n \in [N]}(\lVert M_{-n} \rVert, \lVert w_{-n} \rVert)$ |
| $\varphi(x, a^n)$ | feature map of player $n$ at state action $(x, a^n)$ |
| $\mathcal{D}_{\mathrm{E}}$ | Nash equilibrium expert dataset |
| $\mathcal{D}_{\mathrm{E}}^n$ | Nash equilibrium expert dataset of player $n$ |
| $X_{\mathrm{E},h}^i$ | state of $i$-th trajectory with $i \in [\tau_{\mathrm{E}}]$ at state $h$ under expert occupancy measure |
| $A_{\mathrm{E},h}^{n,i}$ | action of player $n$ of $i$-th trajectory with $i \in [\tau_{\mathrm{E}}]$ at step $h$ under expert occupancy measure |
| $\mathcal{C}_{\varphi,\max}$ | Concentrability coefficient in linear Markov games |
| $A_h^{n,k}$ | action at iteration $k \in [K]$ and step $h$ in Algorithm 2 invoked with $\mathrm{active} = n$. |
| $X_h^{n,k}$ | state at iteration $k \in [K]$ and step $h$ in Algorithm 2 invoked with $\mathrm{active} = n$. |

# B. Related Works

The related works can be split into four areas: *theoretical multi-agent Imitation learning*, *deep multi-agent imitation learning*, *reward-free reinforcement learning* and *single-agent imitation Learning*. We will provide a detailed discussion for all related areas next.

**Theoretical multi-agent imitation learning.** While interest in theoretical guarantees for multi-agent imitation learning has increased in recent years, the area remains relatively underexplored. The first authors to adopt the Nash gap as the optimality objective in imitation learning for normal form games are Waugh et al. (2011). More recently, the Nash gap has also been studied in the context of imitation learning for mean field games (Ramponi et al., 2023). The authors analyze behavioral cloning and adversarial imitation learning and derive upper bounds that scale exponentially with the horizon. Motivated by this hardness result for directly minimizing the Nash gap, they introduce a proxy objective based on a mean field control formulation, which yields an upper bound that scales quadratically with the horizon. In contrast, our work focuses on algorithms that directly minimize the Nash gap in the $N$ player setting, rather than designing surrogate metrics whose minimization implies a small Nash gap through error propagation arguments specific to the mean field regime.

The first work to explicitly consider the Nash gap as the learning objective in the $N$ player multi-agent imitation learning setting is the seminal work of Tang et al. (2024). The authors show that recovering the value of a Nash equilibrium in a multi-agent setting exhibits strong parallels with the single agent case. However, when the learner aims to minimize the Nash gap, they establish a hardness result demonstrating the existence of a Markov game in which any non-interactive learner incurs regret that scales linearly with the horizon. To address this challenge, they propose interactive imitation learning algorithms, namely MALICE and BLADES, which are capable of minimizing the Nash gap. The accompanying analysis for behavioral cloning, MALICE, and BLADES is based on error propagation arguments, and does not provide formal sample complexity guarantees. Moreover, a closer inspection reveals that the computational complexity of these algorithms is exponential in the number of states.

Building on these results, Freihaut et al. (2025a) provide the first sample complexity analysis for behavioral cloning when minimizing the Nash gap. They show that the resulting upper bound depends on the all policy deviation concentrability coefficient $\mathcal{C}_{\max}$. In addition, they establish a hardness result showing that if a related quantity $\mathcal{C}(\pi_E^1, \pi_E^2)$ is infinite, then the Nash gap necessarily scales linearly with the horizon. The authors then move to the interactive setting and introduce MURMAIL, the first algorithm that is both computationally and statistically efficient for minimizing the Nash gap. However, the sample complexity of MURMAIL scales as $\varepsilon^{-8}$, which is suboptimal compared to the $\varepsilon^{-2}$ rate achieved by behavioral cloning.

This gap was recently closed by Freihaut et al. (2025b). In that work, the authors reconcile the upper bound dependence on $\mathcal{C}_{\max}$ with the hardness result involving $\mathcal{C}(\pi_E^1, \pi_E^2)$ by showing that $\mathcal{C}_{\max}$ is indeed the fundamental quantity governing non-interactive multi-agent imitation learning. They further prove a statistical lower bound of order $\Omega(\mathcal{C}_{\max}\varepsilon^{-2})$, establishing that behavioral cloning is rate optimal in the non-interactive setting. By introducing a new framework that combines reward free reinforcement learning and imitation learning, they also propose MAIL WARM, an interactive algorithm with sample complexity scaling as $\varepsilon^{-2}$, which is rate optimal. Nevertheless, all of these results are restricted to tabular settings with small state and action spaces. In addition, MAIL WARM builds on EULER (Zanette & Brunskill, 2019) and therefore does not extend to the discounted infinite horizon setting. In contrast, we provide the first analysis of multi-agent imitation learning with linear function approximation, applicable to both finite horizon and discounted infinite horizon settings, thereby addressing the open questions highlighted by Freihaut et al. (2025b).

**Deep multi-agent imitation learning.** Most empirical work on multi-agent imitation learning rely on strong assumptions or focus on cooperative settings. To the best of our knowledge, the only work that explicitly considers demonstrations generated by Nash equilibrium policies is that of Song et al. (2018), which can be viewed as a natural extension of GAIL (Ho & Ermon, 2016) from the single agent to the multi-agent setting. Their analysis relies on strong assumptions, such as the existence of a unique Nash equilibrium, and their empirical evaluation focuses on recovering policies with high value rather than minimizing the Nash gap. Therefore, this work can be seen orthogonal to ours. Another non-cooperative line of work studies inverse reinforcement learning in multi-agent settings (Yu et al., 2019), where the authors propose an optimality objective tailored to an adversarial reward learning framework. This objective is closely related to QRE in Markov Games and enables the recovery of highly correlated reward functions under function approximation. This direction is orthogonal to our work, as we focus on directly learning expert behavior and target Nash equilibrium solutions.

In the cooperative setting, all agents share a common reward function and aim to optimize a joint expected return. Consequently, existing works adopt value-based performance metrics, which differ fundamentally from the Nash gap objective considered here. While these approaches are therefore orthogonal to ours, we briefly summarize recent developments for completeness. Bui et al. (2024) extend IQ Learn (Garg et al., 2021) to the multi-agent setting by training local IQ Learn agents and combining them through a mixing network, enabling centralized training with decentralized execution. Li et al. (2025) extend this approach to continuous control tasks via an alternative characterization. In a related direction, Bui et al. (2025) consider settings in which demonstrations may originate from suboptimal experts and propose a two stage approach that first classifies trajectories as expert or non expert before learning a robust policy from the labeled data.

**Reward-free reinforcement learning.** Reward free reinforcement learning was first introduced in the seminal work of Jin et al. (2020). The authors study tabular Markov decision processes and formalize reward free reinforcement learning as a two phase procedure consisting of an exploration phase followed by a planning phase. During exploration the learner does not observe rewards and instead aims to visit all states that are significant in the sense that there exists a policy such that they are reachable with high probability. The resulting data collection guarantees that once a reward function is revealed in the second phase, the learner can efficiently compute a near optimal policy. Subsequent works refined the sample complexity bounds in the tabular finite horizon setting (Ménard et al., 2020; Kaufmann et al., 2021). An extension beyond the tabular

setting to linear Markov decision processes was proposed by Wang et al. (2020a), and later improvements showed that in linear Markov decision processes reward free reinforcement learning is, from an information theory perspective, not harder than standard reward-aware reinforcement learning (Wagenmaker et al., 2022). These works build on linear analogues of the procedure introduced by Jin et al. (2020). In our setting, the framework of Wang et al. (2020a) is particularly relevant as it enables a translation to the discounted infinite horizon case through the use of no regret learning and recent advances in this line of work (Moulin et al., 2025b). We emphasize that our goal is not to advance the theory of reward free reinforcement learning itself, but rather to leverage existing results as a component in our study of multi-agent imitation learning, and we include this discussion for completeness.

**Single agent imitation learning.** There has been substantial recent progresses in the theoretical understanding of single agent imitation learning. In the fully offline setting, behavior cloning (BC) (Pomerleau, 1991) has been revisited by Foster et al. (2024), who analyze BC through the lens of supervised learning with the log loss between the learner and expert policies. They establish expert sample complexity bounds that are independent of the horizon under deterministic stationary expert policies and sparse reward functions. Horizon dependence only arises when rewards are dense or when the class containing the expert policy is non-stationary. Furthermore, they show that without additional structural assumptions, interactive imitation learning algorithms cannot improve upon BC in a worst case sense.

This conclusion contrasts with classical results based on error propagation analyses, which demonstrate that interactive imitation learning methods such as DAgger (Ross et al., 2011), Logger (Li & Zhang, 2022), and reward or action value function moments matching (Swamy et al., 2021) can outperform BC when performance is measured using the $0/1$ loss or total variation distance. Improved error propagation guarantees are also possible when the learner is allowed to reset to states sampled from the expert state occupancy measure (Swamy et al., 2023).

Beyond action level supervision, imitation learning in the single agent setting can succeed without observing expert actions when the learner has access to trajectories from the environment. For instance, several works show that imitation is possible using only expert state observations (Sun et al., 2019; Kidambi et al., 2021; Viel et al., 2025), or by leveraging reward features in linear Markov decision processes (Moulin et al., 2025b; Viano et al., 2024; 2022). Taken together, these results indicate that in the single agent case, interaction with the expert does not fundamentally improve worst case performance, whereas access to the environment dynamics or sampling capability enables either improved horizon dependence in tabular settings or imitation without explicit action supervision.

In recent works (Freihaut et al., 2025a;b) sharp separation between single agent and multi-agent imitation learning in the tabular setting has been highlighted, driven by the necessity of the all policy deviation concentrability coefficient $\mathcal{C}_{\max}$. Although tabular Markov games are a special case of linear Markov games, we show that in the presence of function approximation, behavior cloning can potentially achieve stronger guarantees even when $\mathcal{C}_{\max} = \infty$, by replacing it with the weaker quantity $\mathcal{C}_{\varphi,\max}$. Additionally, allowing interaction in multi-agent imitation learning is primarily used to get algorithms independent of $\mathcal{C}_{\max}$ instead of sharpening horizon dependencies.

## C. Future Directions

We highlight here a couple of interesting open questions left open by our work.

**Avoiding the expert policy realizability assumption.** A limitation of our analysis is relying on assumption 3.1 which requires to approximately realize the expert policy in the softmax linear policy class $\Pi_{\text{softlin}}$. Albeit weak and quite common, even in the single agent setting (Foster et al., 2024; Rohatgi et al., 2025) it does exclude some Nash Equilibria that might be observed in the dataset. In these situations, our algorithms could still be applied but the theoretical guarantees would not hold. It would be therefore interesting to drop completely Assumption 3.1 and develop a method that relies only on structural properties of the game. A possible solution would be building on the recent work (Moulin et al., 2025a) that, in the single agent setting, successfully imitates an expert without imposing any realizability assumption while staying completely offline.

**Estimation of $\mathcal{C}_{\varphi,\max}$.** Our theory highlights that in games where $\mathcal{C}_{\varphi,\max}$ is small BC can be applied without issues. However it remains open how $\mathcal{C}_{\varphi,\max}$ can be computed a priori in order to predict the performance of a non interactive algorithm before running it. This might be practically very important when interaction with the expert is costly and algorithm failures can not be tolerated for safety reasons. Our first attempts failed and highlighted that the problem might be hard even

in presence of a best response oracle. Nevertheless, we think this problem deserve future study.

**Adaptive amount of interaction.** Estimation of $\mathcal{C}_{\varphi,\max}$ would also allow to decide the amount of expert interaction needed in case prior data are available. Indeed, in this work, we considered running LSVI-UCB-ZERO-BC starting with an empty dataset but we could imagine the scenario in which we already have some precollected data. In this case, it would be interesting to develop an algorithm which only explores in regions which are not well represented by the given dataset and that hopefully requires no interactions at all if the initial dataset is informative enough, which could be the case if $\mathcal{C}_{\varphi,\max}$ is a small constant and the dataset is already of size $\mathcal{O}(\varepsilon^{-2})$.

# D. Proofs for Section 3

Since the analysis for BC in the $N$-players case has just a minimal complication overhead compared to the two players setting, we present the proof of Theorem 3.3 in the following more general form, which allows for a general number of players $N$ and does not leverage at all the zero-sum structure. The main result for $N$ players is given in Theorem D.5. Similarly to the two-players case, the goal is to find a policy profile $\pi$ with low Nash Gap which in $N$-players game is defined as

$$\mathrm{NG}(\pi_{\mathrm{out}}) = \max_{n \in [N]} \max_{\pi_{\star}^n \in \Pi^n} \left\langle \nu_1, V_n^{\pi_{\star}^n, \pi_{\mathrm{out}}^{-n}} - V_n^{\pi_{\mathrm{out}}} \right\rangle$$

We will consider a dataset collected by a potentially nonstationary expert policy $\pi_{\mathrm{E}}$ profile. In particular, let $\pi_{\mathrm{E},h}^n$ be the policy of agent $n$ at step $h$. Moreover, let $\nu_h^\pi$ be the state occupancy measure for a policy $\pi$ after $h$ steps which is

$$\nu_h^\pi(x) = \mathbb{P}_\pi\left[X_h = x\right]$$

where $X_1, \ldots, X_H$ is generated according to the following interaction protocol:

- Sample $X_0 \sim \nu_1$,

- For $h \in [H]$:
  - Sample $A_h^n \sim \pi_h^n$ for each $n \in [N]$.
  - Sample $X_{h+1} \sim P(\cdot|X_h, \{A_h^n\}_{n=1}^N)$

In particular, we sample the expert dataset $\mathcal{D}_n^{\mathrm{E}} = \left\{X_{\mathrm{E},h}^{n,i}, A_{\mathrm{E},h}^{n,i}\right\}$ as follows: $X_{\mathrm{E},h}^{n,i} \sim \nu_h^{\pi_{\mathrm{E}}}, A_{\mathrm{E},h}^{n,i} \sim \pi_{\mathrm{E},h}^n(\cdot|X_{\mathrm{E},h}^{n,i})$. At this point, before moving to the results for this setting, let us recall that $N$-players Linear Markov games are defined as follows.

**Assumption D.1.** (Stationary $N$-players Linear Markov game) For any agent $n \in [N]$, there exists a known $d$-dimensional feature mapping $\varphi : \mathcal{X} \times \mathcal{A}^n \to \mathbb{R}^d$, such that for any state $x \in \mathcal{X}$, action $a^n \in \mathcal{A}^n$, and any Markov policy $\pi$,

$$\sum_{a^{-n} \in \mathcal{A}^{-n}} \pi^{-n}(a^{-n}|x) P(x'|x, a_n, a^{-n}) = \varphi(x, a^n)^\top M_{\pi^{-n}}(x'),$$

$$\sum_{a^{-n} \in \mathcal{A}^{-n}} \pi^{-n}(a^{-n}|x) r(x, a^n, a^{-n}) = \varphi(x, a^n)^\top w_{\pi^{-n}},$$

where $M_{\pi^{-n}} : \Pi^{-n} \times \mathcal{X} \to \mathbb{R}^d$ and $w_{\pi^{-n}} : \Pi^{-n} \to \mathbb{R}^d$ are both unknown to the agent. Following the convention (Jin et al., 2020; Luo et al., 2021), we assume $\|\varphi(x, a^n)\|_2 \leq 1$ for all $x \in \mathcal{X}, a^n \in \mathcal{A}^n, n \in [N]$ and that $\max(\|M_{\pi^{-n}}\|_2, \|w_{\pi^{-n}}\|_2) \leq B, \pi^{-n} \in \Pi^{-n}$, and $n \in [N]$. Finally, we also assume that $\langle \varphi, \mathbf{1} \rangle = 1$.[6]

This assumption also implies the state-action value functions for all the agents are linear, i.e., for each policy $\pi$ there exists some unknown $d$-dimensional feature mapping $\theta_{h,i}^\pi \in \mathbb{R}^d$ with $\|\theta_{h,i}^\pi\|_2 \leq B_\theta$ such that

$$Q_{n,h}^\pi(x, a^n) = \varphi(x, a^n)^\top \theta_{n,h}^\pi, \quad \forall x \in \mathcal{X}, a^n \in \mathcal{A}^n, n \in [N].$$

Therefore, there exists a weight $\theta_{n,h}^{\mathrm{E}}$ such that we have that the state action value functions at the equilibrium can be written as

$$Q_{n,h}^{\mathrm{E}}(x, a^n) = \varphi(x, a^n)^\top \theta_{n,h}^{\mathrm{E}}$$

---

[6]This fact is needed only in the infinite horizon setting. It can be enforced simply by feature normalization.

and a Nash equilibrium profile can be recovered satisfying these constraints

$$Q_{n,h}^{\mathrm{E}}(x, a^n) \leq Q_{n,h}^{\mathrm{E}}(x, \pi_{\mathrm{E}}^n) \quad \forall n \in [N] \quad \forall a^n \in \mathcal{A}^n.$$

Solving this problem can be PPAD-hard, however this fact shows that one equilibrium that requires $n |\mathcal{X}| A_{\max} H$ memory in a tabular setting can now be stored with only $NdH$ parameters. Therefore, we can apply behavior cloning over a restricted hypothesis class for the expert policy denote by $\Pi_{\mathrm{lin}} = \Pi_{\mathrm{lin}}^1 \times \cdots \times \Pi_{\mathrm{lin}}^N$ defined as follows

$$\Pi_{\mathrm{lin}}^n = \left\{ \pi^n | \exists \theta_n = (\theta_{n,1}, \ldots, \theta_{n,h}) \quad \text{s.t.} \quad \theta_{n,h}^\top \varphi(x, a^n) \leq \theta_{n,h}^\top \varphi(x, \pi^n) \right\}$$

Unfortunately, this policy class is not convex and not Lipschitz continuous in the parameters $\theta_n$. Therefore, we consider the following class which avoids the above issues

$$\Pi_{\mathrm{softlin}}^n = \left\{ \pi^n | \exists \theta_n = (\theta_{n,1}, \ldots, \theta_{n,H}) \quad \text{s.t.} \quad \pi_h^n(a^n|x) = \frac{\exp\left(\eta\varphi(x, a^n)^\top \theta_{n,h}\right)}{\sum_{\tilde{a}^n \in \mathcal{A}^n} \exp\left(\eta\varphi(x, \tilde{a}^n)^\top \theta_{n,h}\right)} \right\}.$$

and we impose the following assumption which allow the Nash equilibrium of the original game to be almost realized for $\eta$ large enough.

**Assumption D.2.** We require that the expert policy collecting the data $\pi_{\mathrm{E},h}$ is in the set of limit points of the set $\Pi_{\mathrm{softlin}}$. That is, $\pi_{\mathrm{E}}^n \in \lim_{\eta \to \infty} \Pi_{\mathrm{softlin}}^n$.

**Discussion of the assumption.** To gain intuition about the above assumption we can consider the following example in which the expert chooses an action in the Nash support with exponentially high probability rather than with probability $1$. which leverages the concept of Quantile Response Equilibria in a normal form game is defined as follows.

**Definition D.3. Quantile Response Equilibria.** Let $Q$ be a collection of payoff matrices $\{Q_1, \ldots, Q_N\}$. We define the regularize payoff matrix for the player $n \in [N]$

$$\tilde{Q}(a^1, \ldots, a^N) = Q(a^1, \ldots, a^N) - \frac{1}{\eta} \sum_{a^n \in \mathcal{A}^n} \pi(a^n) \log \pi(a^n)$$

Let us define the value function for each $n \in [N]$, the state value function

$$\tilde{V}^{\pi^n, \bar{\pi}^{-n}} := \sum_{a^n \in \mathcal{A}^n} \sum_{a^{-n} \in \mathcal{A}^{-n}} \pi^n(a^n) \bar{\pi}^{-n}(a^{-n}) Q(a^1, \ldots, a^N).$$

Then, a policy $\pi_{\mathrm{QRE}}$ is a Quantile Response Equilibrium iff

$$\max_{n \in \{1,2\}} \max_{\pi^n \in \Pi^n} \tilde{V}_n^{\pi^n, \pi_{\mathrm{QRE}}^{-n}} - \tilde{V}_n^{\pi_{\mathrm{QRE}}} = 0$$

The following lemma justifies shows that the class $\Pi_{\mathrm{softlin}}^n$ can realize the quantile response equilibria of the game instantiated with the action value functions of the observed Nash equilibrium.

**Lemma D.4.** *Let us consider a collection of $|\mathcal{X}|$ normal form games each with payoff matrices given by the set $\{Q_1^{\pi_{\mathrm{E}}}(x, \cdot), \ldots, Q_N^{\pi_{\mathrm{E}}}(x, \cdot)\}$ which are the action value functions of the expert policy in the linear markov games at hand. Then, the quantile response equilibria of this game belongs to $\Pi_{\mathrm{softlin}}$.*

*Proof.* The proof leverages the fact that at each state $x \in \mathcal{X}$, the quantile response equilibrium policy takes the form $\pi_{\mathrm{QRE}}^n(a^n|x) = \frac{\exp^{\eta Q^{\pi_{\mathrm{E}}}(x, a^n)}}{\sum_{\tilde{a}^n \in \mathcal{A}^n} \exp^{\eta Q^{\pi_{\mathrm{E}}}(x, \tilde{a}^n)}}$. Then, by linearity of the Markov game there must exist a vector $\theta^E$ such that $Q^{\pi_{\mathrm{E}}}(x, \tilde{a}^n) = \varphi(x, \tilde{a}^n)^T \theta^E$. Therefore, $\pi_{\mathrm{QRE}}$ is the softmax of a linear form and lies in $\Pi_{\mathrm{softlin}}$. $\square$

Finally, it is known that in the limit of $\eta \to \infty$ it is known ( see McKelvey & Palfrey (1995)) that any limit point of a QRE is a Nash equilibrium of the normal form games. Therefore, for any $x \in \mathcal{X}$, $\pi_{\mathrm{QRE}}(\cdot|x)$ tends to the Nash equilibrium of the normal form games with payoff matrices $\{Q_1^{\pi_{\mathrm{E}}}(x, \cdot), \ldots, Q_N^{\pi_{\mathrm{E}}}(x, \cdot)\}$ which is clearly a Nash equilibrium of the original game. Moreover, by continuity in $\eta$ it means that all non-isolated Nash equilibria are limit points of some QRE. If we

were, therefore, imposing that only the Nash value can be written in linear form, we would need to assume that the dataset is collected exactly by one of the Nash equilibrium which can be recovered as limit of a QRE[7]. Our assumption is less restrictive than this because we can realize even action value functions which are not attained by an equilibrium profile. Therefore, $\Pi_{\text{softlin}}^n$ is richer and more likely to realize the observe expert behaviour for large $\eta$.

We now have all the elements to state the main result for non-interactive imitation learning in general $N$-players games.

**Theorem D.5.** *Main result for BC in $N$-players games Let the features concentrability coefficient be defined as*

$$\mathcal{C}_{\varphi,\max} := \max_{n\in[N],h\in[H]} \max_{\pi^{-n}\in\Pi_{\text{softlin}}^{-n}} \max_{\pi_\star^n\in\text{br}(\pi^{-n})} \left\| \varphi_h^{\pi_\star^n,\pi_{\text{E}}^{-n}} \right\|_{\left(\Lambda_{\text{E},h}^n\right)^{-1}},$$

*where , setting $\lambda = 1/\tau_{\text{E}}$ , we define the expert features covariance matrix as*

$$\Lambda_{\text{E},h}^n = \sum_{x',a^n} \varphi(x',a^n)\varphi(x',a^n)^\top \pi_{\text{E}}^n(a^n|x')\nu_h^{\pi_{\text{E}}}(x') + \lambda I.$$

*Then, the output of BC is an $\varepsilon$-approximate Nash equilibrium with probability $1-\delta$ if run taking as input a dataset containing*
$$\tau_{\text{E}} = \widetilde{\mathcal{O}}\left( \frac{H^5 N^4 \mathcal{C}_{\max,\varphi}^2 dB^2 \log(A_{\max}B_\theta\delta^{-1})}{\varepsilon^2} \right) \text{ trajectories from a Nash equilibrium.}$$

### D.1. Proof of Theorem D.5 (N-player general-sum version of Theorem 3.3)

In this section, we present the proof of Theorem D.5 which is the $N$ player general-sum version of Theorem 3.3. The main idea of the proof is to split the sum into a exploitation gap and a value-based gap. Note that the value based gap only appears in the $N$-player general-sum setting and could be omitted in the two player zero-sum case. To bound the exploitation gap a change of measure is required. Therefore, we will make use of the following Lemma, which will be proven after the proof of our main result (see Lemma D.2).

**Lemma D.6.** *Let $W : \mathcal{X} \to \mathbb{R}^+$ be any state dependent function. Moreover, let the feature expectation at step $h \in [H]$ for a policy profile $\pi_\star^n, \pi_{\text{E}}^{-n}$ for an arbitrary $\pi_\star^n$ be defined as*

$$\varphi_{h-1}^{\pi_\star^n,\pi_{\text{E}}^{-n}} = \sum_{x',a^n} \varphi(x',a^n)\pi_\star^n(a^n|x')\nu_{h-1}^{\pi_\star^n,\pi_{\text{E}}^{-n}}(x')$$

*and let the expert features covariance matrix at stage $h \in [H]$ be defined as*

$$\Lambda_{\text{E},h-1}^n = \sum_{x',a^n} \varphi(x',a^n)\varphi(x',a^n)^\top \pi_{\text{E}}^n(a^n|x')\nu_{h-1}^{\pi_{\text{E}}}(x') + \lambda I.$$

*Then, the following change of measure argument holds true*

$$\sum_{h=1}^H \mathbb{E}_{X\sim\nu_h^{\pi_\star^n,\pi_{\text{E}}^{-n}}} [W(X)] \leq \sum_{h=1}^H \left\| \varphi_{h-1}^{\pi_\star^n,\pi_{\text{E}}^{-n}} \right\|_{\left(\Lambda_{\text{E},h-1}^n\right)^{-1}} \sqrt{\mathbb{E}_{X\sim\nu_h^{\pi_{\text{E}}}}[W^2(X)] + \lambda B^2 W_{\max}^2}.$$

Having stated the lemma that enables us to do the change of measure, we proceed with the proof of Theorem D.5.

*Proof of Theorem D.5.* Now, we can start with the analysis of behavioral cloning, which optimizes the following objective for each player $n \in [N]$,

$$\pi_{\text{out}}^n = \underset{\pi^n\in\Pi_{\text{softlin}}}{\arg\max} \sum_{h=1}^H \sum_{i=1}^{\tau_{\text{E}}} \log \pi_h^n(A_{\text{E},h}^{n,i}|X_{\text{E},h}^{n,i}),$$

---

[7]Eibelshäuser & Poensgen (2019, Theorem 5) showed existence of a unique limiting Nash equilibrium starting from the unique QRE that is obtained for $\eta = 0$. Changing the initial value of $\eta$, different limits can be obtained but there are also some (non-isolated) Nash equilibria which can not be recovered as limit of a QRE, namely the isolated Nash equilibria. As a practical example of a Nash equilibrium we can recover consider the zero sum normal form games with payoff matrix $\begin{pmatrix} 1 & 0 \\ 1 & 0 \end{pmatrix}$. For the row player every policy is a Nash equilibrium and no matter what the column player does both actions are best responses. Therefore, the state action value functions against any policy of the column player will always have equal entries for both actions. As a consequence only the uniform distribution over row player's actions can be recovered as limit of a logit quantile response equilibria.

where $X_{\mathrm{E},h}^{n,i}$, $A_{\mathrm{E},h}^{n,i}$ are the state-actions pairs in the dataset $\mathcal{D}_{\mathrm{E}}^n$. We have the following decomposition.

$$\max_{\pi_\star^n \in \Pi^n} \left\langle \nu_1, V_n^{\pi_\star^n, \pi_{\mathrm{out}}^{-n}} - V_n^{\pi_{\mathrm{out}}} \right\rangle = \underbrace{\max_{\pi_\star^n \in \Pi^n} \left\langle \nu_1, V_n^{\pi_\star^n, \pi_{\mathrm{out}}^{-n}} - V_n^{\pi_{\mathrm{E}}} \right\rangle}_{T_1} + \underbrace{\left\langle \nu_1, V_n^{\pi_{\mathrm{E}}} - V_n^{\pi_{\mathrm{out}}} \right\rangle}_{T_2} \tag{1}$$

We will bound the two terms separately. For the first term ($T_1$), we have

$$\max_{\pi_\star^n \in \Pi^n} \left\langle \nu_1, V_n^{\pi_\star^n, \pi_{\mathrm{out}}^{-n}} - V_n^{\pi_{\mathrm{E}}} \right\rangle \overset{(i)}{\leq} \max_{\pi_\star^n \in \Pi^n} \left\langle \nu_1, V_n^{\pi_\star^n, \pi_{\mathrm{out}}^{-n}} - V_n^{\pi_\star^n, \pi_{\mathrm{E}}^{-n}} \right\rangle$$

$$= \max_{\pi_\star^n \in \Pi^n} \sum_{h=1}^{H} \left\langle r_h^n, \mu_h^{\pi_\star^n, \pi_{\mathrm{out}}^{-n}} - \mu_h^{\pi_\star^n, \pi_{\mathrm{E}}^{-n}} \right\rangle$$

$$\overset{(ii)}{\leq} \max_{\pi_\star^n \in \Pi^n} \sum_{h=1}^{H} \|r_h^n\|_\infty \left\| \mu_h^{\pi_\star^n, \pi_{\mathrm{out},h}^{-n}} - \mu_h^{\pi_\star^n, \pi_{\mathrm{E},h}^{-n}} \right\|_1$$

$$\overset{(iii)}{\leq} \max_{\pi_\star^n \in \Pi^n} \sum_{h=1}^{H} \sum_{h'=1}^{h} \mathbb{E}_{X \sim \nu_{h'}^{\pi_\star^n, \pi_{\mathrm{E}}^{-n}}} \mathrm{TV}(\pi_{\mathrm{out},h'}^{-n}, \pi_{\mathrm{E},h'}^{-n})(X)$$

$$\leq H \max_{\pi_\star^n \in \Pi^n} \sum_{h=1}^{H} \mathbb{E}_{X \sim \nu_{h}^{\pi_\star^n, \pi_{\mathrm{E}}^{-n}}} \mathrm{TV}(\pi_{\mathrm{out},h}^{-n}, \pi_{\mathrm{E},h}^{-n})(X),$$

where in $(i)$ we used the fact that $\pi_{\mathrm{E}}$ is a Nash equilibrium and therefore the policies are best responses to each other, in $(ii)$ we used Hölder's inequality with $p = \infty$ and $q = 1$ and in $(iii)$ we used that the rewards are bounded as well as the performance difference lemma combined with Hölder's inequality (see e.g. Kakade & Langford (2002)). Therefore, summing over the players index $n \in [N]$, we obtain

$$\sum_{n=1}^{N} \max_{\pi_\star^n \in \Pi^n} \left\langle \nu_1, V_n^{\pi_\star^n, \pi_{\mathrm{out}}^{-n}} - V_n^{\pi_{\mathrm{E}}} \right\rangle \leq \sum_{n=1}^{N} \max_{\pi_\star^n \in \Pi^n} H \sum_{h=1}^{H} \mathbb{E}_{X \sim \nu_{h}^{\pi_\star^n, \pi_{\mathrm{E}}^{-n}}} \left[ \sum_{n' \neq n} \mathrm{TV}(\pi_{\mathrm{out},h}^{n'}, \pi_{\mathrm{E},h}^{n'})(X) \right]$$

$$\leq \sum_{n=1}^{N} \max_{\pi_\star^n \in \Pi^n} H \sum_{h=1}^{H} \sum_{n' \neq n} \mathbb{E}_{X \sim \nu_{h}^{\pi_\star^{n'}, \pi_{\mathrm{E}}^{-n'}}} \left[ \mathrm{TV}(\pi_{\mathrm{out},h}^{n}, \pi_{\mathrm{E},h}^{n})(X) \right].$$

At this point, we need (Lemma D.6), which enables a change of measure that does not introduce a dependence on $\mathcal{C}_{\max}$ but only on $\mathcal{C}_{\varphi,\max}$. It is of key importance to use the linearity of the transition dynamics which implies linearity of the state occupancy measure of any policy.

Invoking Lemma D.6 with $W(X) = \mathrm{TV}(\pi_{\mathrm{out},h}^{n}, \pi_{\mathrm{E},h}^{n})(X)$, we obtain

$$\sum_{n=1}^{N} \max_{\pi_\star^n \in \Pi^n} \left\langle \nu_1, V_n^{\pi_\star^n, \pi_{\mathrm{out}}^{-n}} - V_n^{\pi_{\mathrm{E}}} \right\rangle \leq$$

$$H \sum_{n=1}^{N} \max_{\pi_\star^n \in \Pi^n} \sum_{h=1}^{H} \sum_{n' \neq n} \left\| \varphi_{h-1}^{\pi_\star^n, \pi_{\mathrm{E}}^{-n}} \right\|_{(\Lambda_{\mathrm{E},h-1}^n)^{-1}} \sqrt{2 \mathbb{E}_{X \sim \nu_{h}^{\pi_{\mathrm{E}}}} \left[ \mathrm{TV}^2(\pi_{\mathrm{out},h}^{n}, \pi_{\mathrm{E},h}^{n})(X) \right] + 4\lambda B^2}$$

For the second term ($T_2$) in (1), no change of measure is required. Therefore, we simply obtain

$$\sum_{n=1}^{N} \left\langle \nu_1, V_n^{\pi_{\mathrm{E}}} - V_n^{\pi_{\mathrm{out}}} \right\rangle \leq HN \sum_{h=1}^{H} \sum_{n=1}^{N} \mathbb{E}_{X \sim \nu_{h}^{\pi_{\mathrm{E}}}} \left[ \mathrm{TV}\left(\pi_{\mathrm{out},h}^{n}, \pi_{\mathrm{E},h}^{n}\right)(X) \right].$$

Therefore, we obtain that the Nash gap is upper-bounded by

$$\sum_{n=1}^{N} \max_{\pi_\star^n \in \Pi^n} \left\langle \nu_1, V_n^{\pi_\star^n, \pi_{\text{out}}^{-n}} - V_n^{\pi_{\text{out}}} \right\rangle \leq HN \sum_{n=1}^{N} \sqrt{H \sum_{h=1}^{H} \mathbb{E}_{X \sim \nu_h^{\pi_E}} \left[ \text{TV}^2 \left( \pi_{\text{out},h}^n, \pi_{E,h}^n \right)(X) \right]}$$

$$+ H \sum_{n=1}^{N} \max_{\pi_\star^n \in \Pi^n} \max_{h \in [H]} \sum_{n' \neq n}^{N} \left\| \varphi_{h-1}^{\pi_\star^n, \pi_E^{-n}} \right\|_{(\Lambda_{E,h-1}^n)^{-1}} \sqrt{2H \sum_{h=1}^{H} \mathbb{E}_{X \sim \nu_h^{\pi_E}} \left[ \text{TV}^2(\pi_{\text{out},h}^n, \pi_{E,h}^n)(X) \right] + 4\lambda B^2 H^2}.$$

At this point, we upper bound the total variation distance by the Hellinger divergence

$$\mathbb{E}_{X \sim \nu_h^{\pi_E}} \left[ \text{TV}^2(\pi_{\text{out},h}^n, \pi_{E,h}^n)(X) \right] \leq 4\mathbb{E}_{X \sim \nu_h^{\pi_E}} \left[ D_{\text{Hel}}^2(\pi_{\text{out},h}^n, \pi_{E,h}^n)(X) \right]$$

where $D_{\text{Hel}}$ is the Hellinger divergence defined as

$$D_{\text{Hel}}^2(p, q) = \sum_{z \in \mathcal{Z}} \left( \sqrt{p(z)} - \sqrt{q(z)} \right)^2$$

for some $p, q \in \Delta_{\mathcal{Z}}$ for some finite set $\mathcal{Z}$. At this point, we can upper bound the sum of the expected local Hellinger divergences with the divergence between trajectories invoking Rohatgi et al. (2025, Lemma H.31),

$$\sum_{h=1}^{H} \mathbb{E}_{X \sim \nu_h^{\pi_E}} \left[ D_{\text{Hel}}^2(\pi_{\text{out},h}^n, \pi_{E,h}^n)(X) \right] \leq 4H D_{\text{Hel}}^2(\mathbb{P}^{\pi_{\text{out}}^n, \pi_E^{-n}}, \mathbb{P}^{\pi_E}),$$

where $\mathbb{P}^\pi$ denotes the probability over trajectories induced by the profile $\pi$. Moreover, invoking Rohatgi et al. (2025, Theorem 4.2) where $B_{\text{ratio}} = \max_{n,h \in [N] \times [H]} \max_{\pi \in \Pi_{\text{softlin}}^n} \left\| \frac{\pi_E}{\pi} \right\|_\infty$, we obtain

$$D_{\text{Hel}}^2(\mathbb{P}^{\pi_{\text{out}}^n, \pi_E^{-n}}, \mathbb{P}^{\pi_E}) \leq \inf_{\epsilon : \epsilon > 0} \left\{ \epsilon H + \frac{\log(|\mathcal{C}_\epsilon(\log \Pi_{\text{softlin}})|/\delta)}{\tau_E} \right\} + \frac{H \log(eB_{\text{ratio}}) \log(1/\delta)}{\tau_E}$$

$$+ H \log(eB_{\text{ratio}}) \min_{\underline{\pi}^n \in \Pi_{\text{softlin}}^n} \mathbb{E}_{X \sim \nu_h^{\pi_E}} \left[ D_{\text{Hel}}^2(\underline{\pi}_h^n, \pi_{E,h}^n)(X) \right]$$

$$\leq \inf_{\epsilon : \epsilon > 0} \left\{ \epsilon H + \frac{\log(|\mathcal{C}_\epsilon(\log \Pi_{\text{softlin}})|/\delta)}{\tau_E} \right\} + \frac{H \log(eB_{\text{ratio}}) \log(1/\delta)}{\tau_E}$$

$$+ H \log(eB_{\text{ratio}}) \underbrace{\min_{\underline{\pi}^n \in \Pi_{\text{softlin}}^n} \mathbb{E}_{X \sim \nu_h^{\pi_E}} \left[ \text{TV}(\underline{\pi}_h^n, \pi_{E,h}^n)(X) \right]}_{T_{\text{mis}}}$$

where the last inequality holds due to the fact that the squared Hellinger distance is upper bounded by the total variation distance. The mispecification term ($T_{\text{mis}}$) accounts for the case that the expert equilibrium providing demonstrations is a pure equilibrium or an equilibrium which put probability mass on certain action lower than $\exp(\eta Q_{\min})/(\exp(\eta Q_{\min}) + (A_{\max}-1)\exp(\eta Q_{\max}))$ then such equilibrium can not be realized by the class $\Pi_{\text{softlin}}$. In particular, for $Q_{\min} = 0$ and $Q_{\max} = H$ the minimum achievable value is $\pi_{\min} = 1/(1+(A_{\max}-1)\exp(\eta H))$. At this point, for any state $x \in \mathcal{X}$ we have that the worst case approximation error is where the player $n$ uses a deterministic Nash, i.e. always players the action $a_{n,h}^\star$ at each stage $h \in [H]$. Therefore, we have that

$$\min_{\underline{\pi} \in \Pi_{\text{softlin}}} \text{TV}(\pi_{E,h}^n, \underline{\pi}_h)(x) = \min_{\underline{\pi} \in \Pi_{\text{softlin}}} 1 - \underline{\pi}_h(a_n^\star|x) + \sum_{a \neq a_n^\star} \underline{\pi}_h(a|x) \leq 2(|\mathcal{A}_n| - 1)\underline{\pi}_{\min}$$

$$= \frac{2(A_{\max} - 1)}{1 + (A_{\max} - 1)\exp(\eta H)}.$$

With similar calculations, we can compute that $B_{\text{ratio}} \leq 1 + (A_{\max} - 1)\exp(\eta H)$. Finally, we can bound the log covering number of the log policy class using Lemma J.3. All in all, we obtain

$$D_{\text{Hel}}^2(\mathbb{P}^{\pi_{\text{out}}^n, \pi_E^{-n}}, \mathbb{P}^{\pi_E}) \leq \inf_{\epsilon : \epsilon > 0} \left\{ \epsilon H + \frac{d\eta H \log\left(\frac{2A_{\max}^2 \eta B_\theta}{\epsilon \delta}\right)}{\tau_E} \right\} + \frac{\eta H^2 \log(2eA_{\max}) \log(1/\delta)}{\tau_E}$$

$$+ \frac{2\eta H^2 (A_{\max} - 1) \log(2eA_{\max})}{1 + (A_{\max} - 1)\exp(\eta H)}.$$

Hence, choosing $\epsilon = (\tau_{\mathrm{E}})^{-1}$, we get

$$D_{\mathrm{Hel}}^2(\mathbb{P}^{\pi_{\mathrm{out}}^n, \pi_{\mathrm{E}}^{-n}}, \mathbb{P}^{\pi_{\mathrm{E}}}) \leq \frac{2d\eta H \log\left(\frac{2A_{\max}^2 \eta B_\theta \tau_{\mathrm{E}}}{\delta}\right)}{\tau_E} + \frac{\eta H^2 \log(2eA_{\max}) \log(1/\delta)}{\tau_E}$$
$$+ \frac{2\eta H^2(A_{\max} - 1) \log(2eA_{\max})}{1 + (A_{\max} - 1) \exp(\eta H)}.$$

In turns, the above bound yields

$$\sum_{h=1}^{H} \mathbb{E}_{X \sim \nu_h^{\pi_{\mathrm{E}}}} \left[ \mathrm{TV}^2(\pi_{\mathrm{out},h}^n, \pi_{\mathrm{E},h}^n)(X) \right] \leq \frac{8d\eta H^2 \log\left(\frac{2A_{\max}^2 \eta B_\theta \tau_{\mathrm{E}}}{\delta}\right)}{\tau_E} + \frac{4\eta H^3 \log(2eA_{\max}) \log(1/\delta)}{\tau_E}$$
$$+ \frac{8\eta H^3(A_{\max} - 1) \log(2eA_{\max})}{1 + (A_{\max} - 1) \exp(\eta H)}.$$

Therefore, setting $\eta$ in the definition of $\Pi_{\mathrm{softlin}}$ as $\eta = \log(\tau_{\mathrm{E}})/H$ gives with probability at least $1 - \delta$

$$\sum_{h=1}^{H} \mathbb{E}_{X \sim \nu_h^{\pi_{\mathrm{E}}}} \left[ \mathrm{TV}^2(\pi_{\mathrm{out},h}^n, \pi_{\mathrm{E},h}^n)(X) \right] \leq \frac{34dH^2 \log(A_{\max} B_\theta \tau_{\mathrm{E}} \log(\tau_{\mathrm{E}}) \delta^{-1})}{\tau_{\mathrm{E}}}.$$

Therefore, plugging in this final bound in the upper bound on the Nash gap

$$H \sum_{n=1}^{N} \max_{\pi_\star^n \in \Pi^n} \left\langle \nu_1, V_n^{\pi_\star^n, \pi_{\mathrm{out}}^{-n}} - V_n^{\pi_{\mathrm{out}}} \right\rangle \leq H^{5/2} N^2 \sqrt{\frac{34d \log(A_{\max} B_\theta \tau_{\mathrm{E}} \log(\tau_{\mathrm{E}}) \delta^{-1})}{\tau_{\mathrm{E}}}}$$
$$+ \sum_{n=1}^{N} \max_{\pi_\star^n \in \Pi^n} H^{5/2} \sum_{n' \neq n}^{N} \max_{h \in [H]} \left\| \varphi_h^{\pi_\star^n, \pi_{\mathrm{E}}^{-n}} \right\|_{\left(\Lambda_{\mathrm{E},h}^n\right)^{-1}} \sqrt{\frac{68d \log(A_{\max} B_\theta \tau_{\mathrm{E}} \log(\tau_{\mathrm{E}}) \delta^{-1})}{\tau_{\mathrm{E}}} + 4\lambda B^2}.$$

Now, setting $\lambda = \tau_{\mathrm{E}}^{-1}$, we obtain that with probability at least $1 - \delta$ it holds that

$$\sum_{n=1}^{N} \max_{\pi_\star^n \in \Pi^n} \left\langle \nu_1, V_n^{\pi_\star^n, \pi_{\mathrm{out}}^{-n}} - V_n^{\pi_{\mathrm{out}}} \right\rangle$$
$$\leq H^{5/2} N^2 \left( \max_{n,h} \max_{\pi_\star^n \in \Pi^n} \left\| \varphi_h^{\pi_\star^n, \pi_{\mathrm{E}}^{-n}} \right\|_{\left(\Lambda_{\mathrm{E},h}^n\right)^{-1}} + 1 \right) \sqrt{\frac{72dB^2 \log(A_{\max} B_\theta \tau_{\mathrm{E}} \log(\tau_{\mathrm{E}}) \delta^{-1})}{\tau_{\mathrm{E}}}}$$
$$\leq \mathcal{O}\left( H^{5/2} N^2 \mathcal{C}_{\varphi,\max} \sqrt{\frac{72dB^2 \log(A_{\max} B_\theta \tau_{\mathrm{E}} \delta^{-1})}{\tau_{\mathrm{E}}}} \right),$$

where we introduced $\mathcal{C}_{\varphi,\max} := \max_{n \in [N], h \in [H]} \max_{\pi^{-n} \in \Pi_{\mathrm{softlin}}^{-n}} \max_{\pi_\star^n \in \mathrm{br}(\pi^{-n})} \left\| \varphi_h^{\pi_\star^n, \pi_{\mathrm{E}}^{-n}} \right\|_{\left(\Lambda_{\mathrm{E},h}^n\right)^{-1}}$. Finally, setting $\tau_{\mathrm{E}} = \widetilde{\mathcal{O}}\left( \frac{H^5 N^4 \mathcal{C}_{\max,\varphi}^2 dB^2 \log(A_{\max} B_\theta \delta^{-1})}{\varepsilon^2} \right)$ we get the final result. Notably, thanks to the linearity of the dynamics we managed to avoid completely the dependence on $\mathcal{C}_{\max}$ replacing it with $\mathcal{C}_{\varphi,\max}$. $\qquad \square$

## D.2. Proof of Lemma D.6

*Proof.* For any state $x \in \mathcal{X}$, let us consider a function $W(x) \in \mathbb{R}^+$ bounded by $W_{\max}$. Then, notice that by linearity of the transition dynamics, we have that

$$
\begin{aligned}
\nu_h^{\pi_\star^n, \pi_{\mathrm{E}}^{-n}}(x) &= \sum_{x', a^n} \sum_{a^{-n}} P(x|x', a^n, a^{-n}) \pi_{\mathrm{E}, h-1}^{-n}(a^{-n}|x') \pi_{\star, h-1}^n(a^n|x') \nu_{h-1}^{\pi_\star^n, \pi_{\mathrm{E}}^{-n}}(x') \\
&= \sum_{x', a^n} \left\langle \varphi(x', a^n), M_{\pi_{\mathrm{E}}^{-n}}(x) \right\rangle \pi_{\star, h-1}^n(a^n|x') \nu_{h-1}^{\pi_\star^n, \pi_{\mathrm{E}}^{-n}}(x') \\
&= \left\langle \sum_{x', a^n} \varphi(x', a^n) \pi_{\star, h-1}^n(a^n|x') \nu_{h-1}^{\pi_\star^n, \pi_{\mathrm{E}}^{-n}}(x'), M_{\pi_{\mathrm{E}}^{-n}}(x) \right\rangle \\
&= \left\langle \varphi_{h-1}^{\pi_\star^n, \pi_{\mathrm{E}}^{-n}}, M_{\pi_{\mathrm{E}}^{-n}}(x) \right\rangle,
\end{aligned}
$$

where we introduced the notation $\varphi_{h-1}^{\pi_\star^n, \pi_{\mathrm{E}}^{-n}} = \sum_{x', a^n} \varphi(x', a^n) \pi_{\star, h-1}^n(a^n|x') \nu_{h-1}^{\pi_\star^n, \pi_{\mathrm{E}}^{-n}}(x')$. Using the linearity of the state occupancy measure, we obtain the following,

$$
\begin{aligned}
\sum_{h=1}^H \mathbb{E}_{X \sim \nu_h^{\pi_\star^n, \pi_{\mathrm{E}}^{-n}}} [W(X)] &= \sum_{h=1}^H \sum_{x \in \mathcal{X}} \left\langle \varphi_{h-1}^{\pi_\star^n, \pi_{\mathrm{E}}^{-n}}, M_{\pi_{\mathrm{E}}^{-n}}(x) \right\rangle W(x) \\
&= \sum_{h=1}^H \left\langle \varphi_{h-1}^{\pi_\star^n, \pi_{\mathrm{E}}^{-n}}, \sum_{x \in \mathcal{X}} M_{\pi_{\mathrm{E}}^{-n}}(x) W(x) \right\rangle \\
&\leq \sum_{h=1}^H \left\| \varphi_{h-1}^{\pi_\star^n, \pi_{\mathrm{E}}^{-n}} \right\|_{(\Lambda_{\mathrm{E}, h-1}^n)^{-1}} \left\| \sum_{x \in \mathcal{X}} M_{\pi_{\mathrm{E}}^{-n}}(x) W(x) \right\|_{\Lambda_{\mathrm{E}, h-1}^n},
\end{aligned} \tag{2}
$$

where we introduced the quantity $\Lambda_{\mathrm{E}, h-1}^n = \sum_{x', a^n} \varphi(x', a^n) \varphi(x', a^n)^\top \pi_{\mathrm{E}, h-1}^n(a^n|x') \nu_{h-1}^{\pi_{\mathrm{E}}}(x') + \lambda I$. Then, let us compute

$$
\begin{aligned}
&\Lambda_{\mathrm{E}, h-1}^n \sum_{x \in \mathcal{X}} M_{\pi_{\mathrm{E}}^{-n}}(x) W(x) \\
&= \sum_{x', a^n} \sum_{x \in \mathcal{X}} \varphi(x', a^n) \varphi(x', a^n)^\top M_{\pi_{\mathrm{E}}^{-n}}(x) W(x) \pi_{\mathrm{E}, h-1}^n(a^n|x') \nu_{h-1}^{\pi_{\mathrm{E}}}(x') + \lambda \sum_{x \in \mathcal{X}} M_{\pi_{\mathrm{E}}^{-n}}(x) W(x) \\
&= \sum_{x', a^n} \sum_{x \in \mathcal{X}} \varphi(x', a^n) P(x|x', a^n, \pi_{\mathrm{E}}^{-n}) W(x) \pi_{\mathrm{E}, h-1}^n(a^n|x') \nu_{h-1}^{\pi_{\mathrm{E}}}(x') + \lambda \sum_{x \in \mathcal{X}} M_{\pi_{\mathrm{E}}^{-n}}(x) W(x).
\end{aligned}
$$

At this point,

$$
\begin{aligned}
&\sum_{\tilde{x} \in \mathcal{X}} M_{\pi_{\mathrm{E}}^{-n}}^\top(\tilde{x}) W(\tilde{x}) \Lambda_{\mathrm{E}, h-1}^n \sum_{x \in \mathcal{X}} M_{\pi_{\mathrm{E}}^{-n}}(x) W(x) \\
&= \sum_{x', a^n} \sum_{x \in \mathcal{X}} \sum_{\tilde{x} \in \mathcal{X}} M_{\pi_{\mathrm{E}}^{-n}}^\top(\tilde{x}) \varphi(x', a^n) P(x|x', a^n, \pi_{\mathrm{E}}^{-n}) W(x) W(\tilde{x}) \pi_{\mathrm{E}, h-1}^n(a^n|x') \nu_{h-1}^{\pi_{\mathrm{E}}}(x') + \lambda \left\| \sum_{x \in \mathcal{X}} M_{\pi_{\mathrm{E}}^{-n}}(x) W(x) \right\|^2 \\
&= \sum_{x', a^n} \sum_{x \in \mathcal{X}} \sum_{\tilde{x} \in \mathcal{X}} P(\tilde{x}|x', a^n, \pi_{\mathrm{E}}^{-n}) P(x|x', a^n, \pi_{\mathrm{E}}^{-n}) W(x) W(\tilde{x}) \pi_{\mathrm{E}, h-1}^n(a^n|x') \nu_{h-1}^{\pi_{\mathrm{E}}}(x') + \lambda \left\| \sum_{x \in \mathcal{X}} M_{\pi_{\mathrm{E}}^{-n}}(x) W(x) \right\|^2
\end{aligned}
$$

Rearranging the last sum, we obtain that

$$
\left\| \sum_{x \in \mathcal{X}} M_{\pi_{\mathrm{E}}^{-n}}(x) W(x) \right\|_{\Lambda_{\mathrm{E},h-1}^n}^2 = \sum_{x',a^n} \left( \sum_{x \in \mathcal{X}} P(x|x',a^n,\pi_{\mathrm{E}}^{-n}) W(x) \right)^2 \pi_{\mathrm{E},h-1}^n(a^n|x') \nu_{h-1}^{\pi_{\mathrm{E}}}(x')
$$

$$
+ \lambda \left\| \sum_{x \in \mathcal{X}} M_{\pi_{\mathrm{E}}^{-n}}(x) W(x) \right\|^2
$$

$$
\leq \sum_{x',a^n} \sum_{x \in \mathcal{X}} P(x|x',a^n,\pi_{\mathrm{E}}^{-n}) W^2(x) \pi_{\mathrm{E},h-1}^n(a^n|x') \nu_{h-1}^{\pi_{\mathrm{E}}}(x')
$$

$$
+ \lambda B^2 W_{\max}^2
$$

where Jensen's inequality guarantees that $\sum_{x \in \mathcal{X}} P(x|x',a^n,\pi_{\mathrm{E}}^{-n}) W^2(x) \geq (\sum_{x \in \mathcal{X}} P(x|x',a^n,\pi_{\mathrm{E}}^{-n}) W(x))^2$. Finally, noticing that $\sum_{x',a^n} P(x|x',a^n,\pi_{\mathrm{E}}^{-n}) \pi_{\mathrm{E},h-1}^n(a^n|x') \nu_{h-1}^{\pi_{\mathrm{E}}}(x') = \nu_h^{\pi_{\mathrm{E}}}(x)$, we get

$$
\left\| \sum_{x \in \mathcal{X}} M_{\pi_{\mathrm{E}}^{-n}}(x) W(x) \right\|_{\Lambda_{\mathrm{E},h-1}^n}^2 \leq \sum_{x \in \mathcal{X}} W^2(x) \nu_h^{\pi_{\mathrm{E}}}(x) + \lambda B^2 W_{\max}^2 .
$$

Therefore, putting all the pieces together we obtain,

$$
\sum_{h=1}^{H} \mathbb{E}_{X \sim \nu_h^{\pi_\star^n, \pi_{\mathrm{E}}^{-n}}} [W(X)] \leq \sum_{h=1}^{H} \left\| \varphi_{h-1}^{\pi_\star^n, \pi_{\mathrm{E}}^{-n}} \right\|_{(\Lambda_{\mathrm{E},h-1}^n)^{-1}} \sqrt{\mathbb{E}_{X \sim \nu_h^{\pi_{\mathrm{E}}}} [W^2(X)] + \lambda B^2 W_{\max}^2} . \tag{3}
$$

$\square$

# E. Proofs of Section 4

In this section we present the analysis of LSVI-UCB-ZERO-BC. In this cases, there are some technical differences between the two and $N$ players setting. Therefore, we choose to present the proofs separately: we start with the simpler two players setup and move to the $N$-players setup in Appendix F. Before stating the proofs of the technical lemmas used to prove the main theorem of the interactive setting (Theorem 4.1), we provide an analysis sketch. While the algorithmic components are similar to the ones of the tabular case, the required analysis differs significantly. In the tabular case (Freihaut et al., 2025b), the exploration dataset covered all significant states, which enabled a change of measure at states level and then enabled an application of classical BC results for the second case. This time we have to perform the change of measure at features level in order to avoid dependence on the number of states. In particular, for each player $n \in \{1,2\}$ the change of measures will depend on the covariance feature matrices $\Lambda_h^{-n,K}$ for $n = 1,2$ constructed after $K$ iterations of Algorithm 2. This is given in Lemma 4.3.

We notice that Lemma 4.3 avoids completely the dependence on $\mathcal{C}_{\varphi,\max}$ replacing it by $\max_{\pi^{-n} \in \Pi^{-n}} \left\| \varphi_h^{\pi_E^n, \pi^{-n},} \right\|_{(\Lambda_h^{-n,K})^{-1}}$.

The main difference between the two quantities is that in $\Lambda_h^{-n,K}$ the features are evaluated at the states and actions sampled while running LSVI-UCB-ZERO and not via Nash vs Nash dynamics. Notably, we can control the latter quantity by proving that when the matrix $\Lambda_h^{-n,K}$ is constructed via LSVI-UCB-ZERO, the $\Lambda_h^{-n,K}$-weighted norm of any expected feature vector decreases at a $\mathcal{O}(K^{-1/2})$ rate. This result is given in Lemma 4.4.

At this point, the proof is concluded by invoking the guarantees for the conditional maximum likelihood estimator (MLE) under misspecification to prove the following high probability bound on $\Delta \mathrm{TV}_{n,h}^2(\pi_{\mathrm{out}})$ given in Lemma 4.5.

The final result given in Theorem 4.1 follows from combining the change of measure in Lemma 4.3, the bound on the weighted norm of any feature expectation vector in Lemma 4.4 and finally the MLE guarantees in Lemma 4.5.

In the following subsections, we give the proofs of the introduced lemmas.

### E.1. Proof of Lemma 4.3

**Lemma 4.3.** *Let $\Lambda_h^{-n,K}$ be the output covariance matrix and $\left\{\pi_k^{-n}\right\}_{k=1}^K$ be the sequence of policies generated by Algorithm 2 invoked with* active $= -n$, *i.e.* $\Lambda_h^{-n,K} = \sum_{k=1}^K \varphi(X_h^{-n,k}, A_h^{-n,k})\varphi(X_h^{-n,k}, A_h^{-n,k})^\top + I$. *Then, the Nash gap for $\pi_{\mathrm{out}}$, the output policy of Algorithm 1, can be upper bounded, with probability at least $1 - \delta$, as*

$$
\mathrm{NG}(\pi_{\mathrm{out}}) \leq H \sum_{h=1}^H \max_{\substack{n \in \{1,2\} \\ \pi^{-n} \in \Pi^{-n}}} \left\| \varphi_h^{\pi_E^n, \pi^{-n}} \right\|_{(\Lambda_h^{-n,K})^{-1}}
$$
$$
\cdot \mathcal{O}\left( \sqrt{\Delta\mathrm{TV}_{n,h}^2(\pi_{\mathrm{out}}) + B^2 d \log(K/\delta)} \right)
$$

*where $\Delta\mathrm{TV}_{n,h}^2(\pi_{\mathrm{out}})$ is defined as follows*

$$
\sum_{k=1}^K \sum_{x \in \mathcal{X}} \nu_h^{\pi_E^n, \pi^{-n,k}}(x)\mathrm{TV}^2(\pi_{\mathrm{out},h}^n, \pi_{E,h}^n)(x),
$$

*and* TV *is the total variation (for a definition see Table A).*

*Proof.* Let us recall that we now focus on the case of two players zero sum and that the first player maximizes the value function. Then, we have that the Nash gap can be written as follows for any arbitrary choice of the best responding policies, i.e. $\pi_\star^2 \in \mathrm{br}(\pi_{\mathrm{out}}^1), \pi_\star^1 \in \mathrm{br}(\pi_{\mathrm{out}}^2)$

$$
\begin{aligned}
\mathrm{NG}(\pi_{\mathrm{out}}) &\overset{(i)}{\leq} \left\langle \nu_1, V^{\pi_\star^1, \pi_{\mathrm{out}}^2} - V^{\pi_{\mathrm{out}}^1, \pi_\star^2} \right\rangle \\
&= \left\langle \nu_1, V^{\pi_\star^1, \pi_{\mathrm{out}}^2} - V^{\pi_E} \right\rangle + \left\langle \nu_1, V^{\pi_E} - V^{\pi_{\mathrm{out}}^1, \pi_\star^2} \right\rangle \\
&\overset{(ii)}{\leq} \left\langle \nu_1, V^{\pi_\star^1, \pi_{\mathrm{out}}^2} - V^{\pi_\star^1, \pi_E^2} \right\rangle + \left\langle \nu_1, V^{\pi_E^1, \pi_\star^2} - V^{\pi_{\mathrm{out}}^1, \pi_\star^2} \right\rangle \\
&\overset{(iii)}{\leq} H\left( \sum_{h=1}^H \mathbb{E}_{X \sim \nu_h^{\pi_\star^1, \pi_E^2}} \left[ \mathrm{TV}\left( \pi_{\mathrm{out},h}^2, \pi_{E,h}^2 \right)(X) \right] + \sum_{h=1}^H \mathbb{E}_{X \sim \nu_h^{\pi_E^1, \pi_\star^2}} \left[ \mathrm{TV}\left( \pi_{\mathrm{out},h}^1, \pi_{E,h}^1 \right)(X) \right] \right) \\
&\leq H \max_{\pi^1 \in \Pi^1} \sum_{h=1}^H \mathbb{E}_{X \sim \nu_h^{\pi^1, \pi_E^2}} \left[ \mathrm{TV}\left( \pi_{\mathrm{out},h}^2, \pi_{E,h}^2 \right)(X) \right] \\
&\quad + H \max_{\pi^2 \in \Pi^2} \sum_{h=1}^H \mathbb{E}_{X \sim \nu_h^{\pi_E^1, \pi^2}} \left[ \mathrm{TV}\left( \pi_{\mathrm{out},h}^1, \pi_{E,h}^1 \right)(X) \right],
\end{aligned}
\tag{4}
$$

where in $(i)$ we used $\max\{a, b\} \leq a + b$ for $a, b > 0$, in $(ii)$ we used that the expert is a NE and therefore each policy is a best response to another and in $(iii)$ we used the performance difference lemma (Kakade & Langford, 2002) combined with Hölder's inequality and the fact that the value function is bounded by $H$. At this point, we need to bound

$$
\max_{\pi^{-n} \in \Pi^{-n}} \mathbb{E}_{X \sim \nu_h^{\pi_E^n, \pi^{-n}}} \left[ \mathrm{TV}\left( \pi_{\mathrm{out},h}^n, \pi_{E,h}^n \right)(X) \right]
$$

for $n \in \{1, 2\}$. The steps of this proof follow similar steps as done in the proof of Lemma D.6. However, instead of the matrix $\Lambda_{E,h}^n$, we introduce the covariance matrices $\Lambda_h^{-n,k}$ generated by Algorithm 2 invoked with active $= -\mathrm{n}$. Applying inequality (2) with the covariance matrix $\Lambda_h^{-n,k}$ and $W(X) = \mathrm{TV}\left( \pi_{\mathrm{out},h}^n, \pi_{E,h}^n \right)(X)$, we get that

$$
\begin{aligned}
&\max_{\pi^{-n} \in \Pi^{-n}} \mathbb{E}_{X \sim \nu_h^{\pi_E^n, \pi^{-n}}} \left[ \mathrm{TV}\left( \pi_{\mathrm{out},h}^n, \pi_{E,h}^n \right)(X) \right] \\
&\leq \max_{\pi^{-n} \in \Pi^{-n}} \sum_{h=1}^H \left\| \varphi_h^{\pi^{-n}, \pi_E^n} \right\|_{(\Lambda_{h-1}^{-n,K})^{-1}} \left\| \sum_{x \in \mathcal{X}} M_{\pi_E^n}(x)\mathrm{TV}(\pi_{\mathrm{out},h}^n, \pi_{E,h}^n)(x) \right\|_{\Lambda_h^{-n,K}}
\end{aligned}
$$

Again following the steps of the proof of Lemma D.6, we get with the replaced terms the analogous results as in (3), namely

$$\max_{\pi^{-n} \in \Pi^{-n}} \mathbb{E}_{X \sim \nu_h^{\pi_E^n, \pi^{-n}}} \left[ \mathrm{TV}\left(\pi_{\mathrm{out},h}^n, \pi_{E,h}^n\right)(X)\right] \leq \max_{\pi^{-n} \in \Pi^{-n}} \sum_{h=1}^{H} \left\|\varphi_{h-1}^{\pi^{-n}, \pi_E^n}\right\|_{(\Lambda_{h-1}^{-n,K})^{-1}}$$
$$\cdot \sqrt{\sum_{k=1}^{K} \sum_{x \in \mathcal{X}} P^{\pi_E^n}(x|X_{h-1}^{-n,k}, A_{h-1}^{-n,k})\mathrm{TV}^2\left(\pi_{\mathrm{out},h}^n, \pi_{E,h}^n\right)(x) + 4B^2},$$

where we also used $W_{\max}^2 \leq 4$ for our current choice of $W(x)$.

At this point, we would like to establish that with high probability $\sum_{x \in \mathcal{X}} P^{\pi_E^n}(x|X_{h-1}^{-n,k}, A_{h-1}^{-n,k})\mathrm{TV}^2(\pi_{\mathrm{out},h}^n, \pi_{E,h}^n)(x)$ concentrates around $\sum_{x \in \mathcal{X}} \nu^{\pi_E^n, \pi_k^{-n}}(x)\mathrm{TV}^2(\pi_{\mathrm{out},h}^n, \pi_{E,h}^n)(x)$. Towards this goal let us denote by $\mathcal{F}_k$ the sigma algebra induced by the random variables $\mathcal{D}_k = \{X_h^{n,s}, A_h^{n,s}\}_{s=1,h=1,n=1}^{k-1,H,N}$ where in the two player case we can set $N = 2$. Unfortunately $\mathrm{TV}^2(\pi_{\mathrm{out},h}^n, \pi_{E,h}^n)(x)$ is not an $\mathcal{F}_k$-adapted random variable unless $k = K$ because the policy $\pi_{\mathrm{out},h}^n$ is computed using the whole dataset $\mathcal{D}_K$. Therefore, we can not apply Lemma J.1 directly but we need to pass through a covering argument. Towards this goal, let us define the function class $\mathcal{W}_h := \left\{W : \mathcal{X} \to [0, 2] : W(x) = \mathrm{TV}(\pi, \pi_{E,h}^n)(x) | \pi \in \Pi_{\mathrm{softlin}}^n\right\}$. Clearly, $\mathrm{TV}(\pi_{\mathrm{out},h}^n, \pi_{E,h}^n)(x) \in \mathcal{W}_h$ because $\pi_{\mathrm{out},h}^n \in \Pi_{\mathrm{softlin}}^n$. Therefore, we can achieve our goal by establishing a uniform convergence argument over the class $\mathcal{W}_h$. As an intermediate step, we need to consider the covering set $\mathcal{C}_\epsilon(\mathcal{W}_h)$ defined such that

$$\forall W \in \mathcal{W}_h, \quad \exists \tilde{W} \in \mathcal{C}_\epsilon(\mathcal{W}_h) \quad \text{s.t.} \left\|W - \tilde{W}\right\|_\infty \leq \epsilon.$$

Then let us fix a particular $\tilde{W}$ and let us invoke Lemma J.1 for $X_k = \sum_{x \in \mathcal{X}} \frac{1}{4} P^{\pi_E^n}(x|X_{h-1}^{-n,k}, A_{h-1}^{-n,k})\tilde{W}^2(x)$ for all $h \in [H]$ to obtain that with probability $1 - \delta$,

$$\sum_{k=1}^{K} \sum_{x \in \mathcal{X}} P^{\pi_E^n}(x|X_{h-1}^{-n,k}, A_{h-1}^{-n,k})\tilde{W}^2(x) \leq 2 \sum_{k=1}^{K} \mathbb{E}\left[\sum_{x \in \mathcal{X}} P^{\pi_E^n}(x|X_{h-1}^{-n,k}, A_{h-1}^{-n,k})\tilde{W}^2(x) \Big| \mathcal{D}_k\right] + 4 \log\left(\frac{1}{\delta}\right).$$

Then, via a union bound over $\mathcal{C}_\epsilon(\mathcal{W}_h)$ it holds that for all $\tilde{W} \in \mathcal{C}_\epsilon(\mathcal{W}_h)$ simultaneously we have with probability $1 - \delta$ that

$$\sum_{k=1}^{K} \sum_{x \in \mathcal{X}} P^{\pi_E^n}(x|X_{h-1}^{-n,k}, A_{h-1}^{-n,k})\tilde{W}^2(x) \leq 2 \sum_{k=1}^{K} \mathbb{E}\left[\sum_{x \in \mathcal{X}} P^{\pi_E^n}(x|X_{h-1}^{-n,k}, A_{h-1}^{-n,k})\tilde{W}^2(x) \Big| \mathcal{D}_k\right] + 4 \log\left(\frac{|\mathcal{C}_\epsilon(\mathcal{W}_h)|}{\delta}\right).$$

Then, it follows that with probability at least $1 - \delta$ for all $W \in \mathcal{W}_h$ simultaneously that

$$
\begin{aligned}
\sum_{k=1}^{K} \sum_{x \in \mathcal{X}} P^{\pi_{\mathrm{E}}^n}(x | X_{h-1}^{-n,k}, A_{h-1}^{-n,k}) W^2(x) &\leq \sum_{k=1}^{K} \sum_{x \in \mathcal{X}} P^{\pi_{\mathrm{E}}^n}(x | X_{h-1}^{-n,k}, A_{h-1}^{-n,k})(W^2(x) - \tilde{W}^2(x)) \\
&+ 2 \sum_{k=1}^{K} \mathbb{E} \left[ \sum_{x \in \mathcal{X}} P^{\pi_{\mathrm{E}}^n}(x | X_{h-1}^{-n,k}, A_{h-1}^{-n,k}) \left( \tilde{W}^2(x) - W^2(x) \right) | \mathcal{D}_k \right] \\
&+ 2 \sum_{k=1}^{K} \mathbb{E} \left[ \sum_{x \in \mathcal{X}} P^{\pi_{\mathrm{E}}^n}(x | X_{h-1}^{-n,k}, A_{h-1}^{-n,k}) W^2(x) | \mathcal{D}_k \right] \\
&+ 4 \log \left( \frac{|\mathcal{C}_\epsilon(\mathcal{W}_h)|}{\delta} \right) \\
&\leq 2 W_{\max} \epsilon K + 2 \sum_{k=1}^{K} \mathbb{E} \left[ \sum_{x \in \mathcal{X}} P^{\pi_{\mathrm{E}}^n}(x | X_{h-1}^{-n,k}, A_{h-1}^{-n,k}) W^2(x) | \mathcal{D}_k \right] \\
&+ 4 \log \left( \frac{|\mathcal{C}_\epsilon(\mathcal{W}_h)|}{\delta} \right) \\
&= 2 W_{\max} \epsilon K + 2 \sum_{k=1}^{K} \sum_{x \in \mathcal{X}} \nu_h^{\pi_{\mathrm{E}}^n, \pi^{-n,k}}(x) W^2(x) \\
&+ 4 \log \left( \frac{|\mathcal{C}_\epsilon(\mathcal{W}_h)|}{\delta} \right)
\end{aligned}
$$

Then, using the bound on the covering number, i.e. $|\mathcal{C}_\epsilon(\mathcal{W}_h)|$ (see Lemma J.2), we conclude that for any $W \in \mathcal{W}_h$ with probability at least $1 - \delta$, it holds that

$$
\begin{aligned}
\sum_{k=1}^{K} \sum_{x \in \mathcal{X}} P^{\pi_{\mathrm{E}}^n}(x | X_{h-1}^{-n,k}, A_{h-1}^{-n,k}) W^2(x) &\leq 2 W_{\max} \epsilon K + 2 \sum_{k=1}^{K} \sum_{x \in \mathcal{X}} \nu_h^{\pi_{\mathrm{E}}^n, \pi^{-n,k}}(x) W^2(x) \\
&+ 4d \log \left( \frac{1 + \frac{4 \eta A_{\max} B_\theta}{\epsilon}}{\delta} \right).
\end{aligned}
$$

Then, putting all together, using $W_{\max} \leq 2$, choosing the size of the covering set as $\epsilon = 1/K$ and using that $\mathrm{TV}(\pi_{\mathrm{out},h}^n, \pi_{E,h}^n)(x) \in \mathcal{W}_h$, we obtain that

$$
\begin{aligned}
\max_{\pi^{-n} \in \Pi^{-n}} \mathbb{E}_{X \sim \nu_h^{\pi_{\mathrm{E}}^n, \pi^{-n}}} &\left[ \mathrm{TV} \left( \pi_{\mathrm{out},h}^n, \pi_{E,h}^n \right)(X) \right] \leq \max_{\pi^{-n} \in \Pi^{-n}} \sum_{h=1}^{H} \left\| \varphi_{h-1}^{\pi^{-n}, \pi_{\mathrm{E}}^n} \right\|_{(\Lambda_{h-1}^{-n,K})^{-1}} \\
&\cdot \sqrt{4 + 2 \sum_{k=1}^{K} \sum_{x \in \mathcal{X}} \nu_h^{\pi_{\mathrm{E}}^n, \pi^{-n,k}}(x) \mathrm{TV}^2(\pi_{\mathrm{out},h}^n, \pi_{E,h}^n)(x) + 4d \log \left( \frac{4K \log K A_{\max} B_\theta}{\delta} \right) + 4B^2} \\
&\leq \max_{\pi^{-n} \in \Pi^{-n}} \sum_{h=1}^{H} \left\| \varphi_{h-1}^{\pi^{-n}, \pi_{\mathrm{E}}^n} \right\|_{(\Lambda_{h-1}^{-n,K})^{-1}} \\
&\cdot \sqrt{2 \sum_{k=1}^{K} \sum_{x \in \mathcal{X}} \nu_h^{\pi_{\mathrm{E}}^n, \pi^{-n,k}}(x) \mathrm{TV}^2(\pi_{\mathrm{out},h}^n, \pi_{E,h}^n)(x) + 36dB^2 \log \left( \frac{4K \log K A_{\max} B_\theta}{\delta} \right)}.
\end{aligned}
$$

where the last steps uses that

$$
4d \log \left( \frac{4K \log K A_{\max} B_\theta}{\delta} \right) + 4B^2 + 4 \leq 36dB^2 \log \left( \frac{4K \log K A_{\max} B_\theta}{\delta} \right)
$$

since both terms are larger than one. Using the big-oh notation and plugging this into (4) concludes the proof. That is,

$$\mathrm{NG}(\pi_{\mathrm{out}}) \leq H \max_{n \in \{1,2\}} \max_{\pi^{-n} \in \Pi^{-n}} \sum_{h=1}^{H} \left\| \varphi_{h-1}^{\pi^{-n}, \pi_{\mathrm{E}}^{n}} \right\|_{(\Lambda_{h-1}^{-n,K})^{-1}}$$

$$\cdot \sqrt{2 \sum_{k=1}^{K} \sum_{x \in \mathcal{X}} \nu_h^{\pi_{\mathrm{E}}^n, \pi^{-n,k}}(x) \mathrm{TV}^2(\pi_{\mathrm{out},h}^n, \pi_{E,h}^n)(x) + 36 d B^2 \log\left( \frac{4K \log K A_{\max} B_\theta}{\delta} \right)}.$$

$\square$

### E.2. Proof of Lemma 4.4

The proof of this result follows directly from the $N$-players case proven next. Note that we have given the general-sum $N$-player version in Algorithm 3. The conceptual idea of both algorithms is identical. The most important change is a change of notation. While in the 2 player setting we set $\mathrm{active} = -n$ which is only one player, here we need to change to $\mathrm{active} = \mathrm{n}$ to ensure again that only one player is actively exploring. That being said, we can continue with the theoretical result for the $N$-players case.

**Lemma E.1.** *For each player $n \in [N]$, the output matrix $\Lambda^{n,K}$ generated by Algorithm 3 invoked with $\mathrm{active} = n$ satisfies with probability $1 - \delta$ that*

$$\max_{\pi^n \in \Pi^n} \sum_{h=1}^{H} \left\| \varphi^{\pi^n, \pi_{\mathrm{E}}^{-n}} \right\|_{(\Lambda_h^{n,K})^{-1}} \leq \sqrt{\frac{d^3 H^4 B^2 \log(dKH/\delta)}{K}}.$$

*Proof.* To see this let us first aim at bounding the following regret notion, defined for any comparator policy $\pi_\star^n$. For simplicity, we now drop the footnote $n$ because the exact same reasoning applies to all players.

$$\mathrm{Regret}(\pi_\star) := \sum_{k=1}^{K} \sum_{h=1}^{H} \left\langle \mu_h^{\pi_\star}, r_h^k \right\rangle - \sum_{k=1}^{K} \sum_{h=1}^{H} \left\langle \mu_h^{\pi^k}, r_h^k \right\rangle$$

Note, that the reward of the algorithm is given by

$$r_h^{n,k}(x, a) = \| \varphi(x, a) \|_{(\Lambda_{n,h}^k)^{-1}}.$$

It is important to emphasize, that this reward is *not* the reward of the underlying Markov game. Instead this reward is an artificially constructed reward to increase the exploration. Additionally, notice that the exploratory rewards $\{ r^k \}_{k=1}^{K}$ are bounded between 0 and 1 and it is revealed to the learner *before* the policy $\pi^k$ is updated. Therefore, invoking Viano et al. (2024, Theorem 6)[8] we obtain

$$\mathrm{Regret}(\pi_\star) \leq \mathcal{O}\left( H^2 d^{3/2} B \sqrt{K \log(K\delta^{-1})} \right).$$

Therefore, rearranging, we obtain that with probability $1 - \delta$

$$\sum_{k=1}^{K} \sum_{h=1}^{H} \left\langle \mu_h^{\pi_\star}, r_h^k \right\rangle \leq \sum_{k=1}^{K} \sum_{h=1}^{H} \left\langle \mu_h^{\pi^k}, r_h^k \right\rangle + \mathcal{O}\left( H^2 d^{3/2} B \sqrt{K \log(K\delta^{-1})} \right).$$

At this point, the first term on the right hand side can be upper bounded via a variant of the Freedman's inequality (Cohen et al., 2019, Lemma D.4) and the classical elliptical potential lemma (Abbasi-Yadkori et al., 2011). In particular, invoking Viano et al. (2024, Lemma 10) we obtain

$$\sum_{k=1}^{K} \sum_{h=1}^{H} \left\langle \mu_h^{\pi^k}, r_h^k \right\rangle \leq \mathcal{O}\left( d^{3/2} H^2 B \sqrt{K \log(2K\delta^{-1})} \right). \tag{5}$$

---

[8]This is up to a minimal variation to accommodate quadratic rewards instead of linear ones and transition weights norm bounded by $B$ instead than 1, i.e. $\left\| M^{\pi^{-n}} \right\| \leq B$ for all $n \in [N]$ and $\pi^n \in \Pi^n$.

Therefore, all in all with probability $1 - \delta$

$$\sum_{k=1}^{K} \sum_{h=1}^{H} \langle \mu_h^{\pi_\star}, r_h^k \rangle \leq \mathcal{O}\left(H^2 d^{3/2} B \sqrt{K \log(K\delta^{-1})}\right),$$

where a union bound is not needed since the event under which the regret is bounded contains the event under which (5) holds. Now, since for each state-action pair and episode index $k \in [K]$, it holds that $r^k(x,a) \leq r^{k-1}(x,a)$ therefore, we have that

$$K \sum_{h=1}^{H} \langle \mu_h^{\pi_\star}, r_h^K \rangle \leq \sum_{k=1}^{K} \sum_{h=1}^{H} \langle \mu_h^{\pi_\star}, r_h^k \rangle \leq \mathcal{O}\left(H^2 d^{3/2} B \sqrt{K \log(K\delta^{-1})}\right)$$

Now, rewriting the reward function more explicitly and using the Jensen's inequality

$$\langle \mu_h^{\pi_\star}, r_h^K \rangle = \sum_{x,a} \mu_h^{\pi_\star}(x,a) \|\varphi(x,a)\|_{(\Lambda_h^K)^{-1}}$$

$$\geq \left\| \sum_{x,a} \mu_h^{\pi_\star}(x,a) \varphi(x,a) \right\|_{(\Lambda_h^K)^{-1}}$$

$$:= \|\varphi_h^{\pi_\star}\|_{(\Lambda_h^K)^{-1}}$$

Therefore, choosing $\pi_\star$ to be the maximizer of the last quantity in the derivation above and dividing by $K$, we obtain that with probability $1 - \delta$ it holds that

$$\max_{\pi_\star \in \Pi} \sum_{h=1}^{H} \|\varphi_h^{\pi_\star}\|_{(\Lambda_h^K)^{-1}} \leq \mathcal{O}\left(H^2 d^{3/2} B \sqrt{\frac{\log(K\delta^{-1})}{K}}\right).$$

The full theorem statement considers that each player $n$ acts in the environment where all other players are fixed, playing according to the profile $\pi_E^{-n}$. Therefore, for each player index $n \in [N]$, $\max_{\pi_\star \in \Pi} \|\varphi_h^{\pi_\star}\|$ can be replaced by $\max_{\pi^n \in \Pi^n} \left\| \varphi_h^{\pi^n, \pi_E^{-n}} \right\|$. $\qquad \square$

Finally, we provide the guarantees for the mispecified MLE.

### E.3. Proof of Lemma 4.5

**Lemma 4.5.** *For any failure probability $\delta \in (0,1]$, let us consider $\eta = \log(K)/H$ in the definition of the policy class $\Pi_{\text{softlin}}$, then for any $n, h \in \{1,2\} \times [H]$ it holds that with probability at least $1 - \delta$, we have $\Delta \mathrm{TV}_{n,h}^2(\pi_{\text{out}}) \leq \widetilde{\mathcal{O}}\left(d \log\left(K A_{\max} B_\theta \delta^{-1}\right)\right)$.*

*Proof.* We start by reminding ourselves that we have defined

$$\Delta \mathrm{TV}_{n,h}^2(\pi_{\text{out}}) := \sum_{k=1}^{K} \sum_{x \in \mathcal{X}} \nu_h^{\pi_E^n, \pi^{-n,k}}(x) \mathrm{TV}^2(\pi_{\text{out},h}^n, \pi_{E,h}^n)(x).$$

We continue by upper bounding the total variation by the Hellinger distance, i.e.

$$2\sum_{k=1}^{K} \sum_{x \in \mathcal{X}} \nu_h^{\pi_E^n, \pi^{-n,k}}(x) \mathrm{TV}^2(\pi_{\text{out},h}^n, \pi_{E,h}^n)(x) \leq 8\sum_{k=1}^{K} \sum_{x \in \mathcal{X}} \nu_h^{\pi_E^n, \pi^{-n,k}}(x) D_{\text{Hel}}^2(\pi_{\text{out},h}^n, \pi_{E,h}^n)(x). \tag{6}$$

Then, applying Lemma J.4 to upper bound the term involving the squared Hellinger distance between the output policy and the expert, we obtain with probability $1 - 2\delta$

$$8\sum_{k=1}^{K} \sum_{x \in \mathcal{X}} \nu_h^{\pi_E^n, \pi^{-n,k}}(x) D_{\text{Hel}}^2(\pi_{\text{out},h}^n, \pi_{E,h}^n)(x) \leq 16\epsilon K + 16\log\frac{|\mathcal{C}_\epsilon(\log \Pi_{\text{softlin}})|}{\delta} + \frac{1152(\log A_{\max} + \eta H)}{\exp(\eta H)}K$$

$$+ 32(\log A_{\max} + \eta H)\log\frac{1}{\delta}.$$

Now, similarly to the proof of Theorem D.5 by applying Lemma J.2 and the choice of $\epsilon = K^{-1}$, we obtain

$$\sum_{k=1}^{K} \sum_{x \in \mathcal{X}} \nu_h^{\pi_{\mathrm{E}}^n, \pi^{-n,k}}(x) D_{\mathrm{Hel}}^2(\pi_{\mathrm{out},h}^n, \pi_{\mathrm{E},h}^n)(x) \leq 2 + 2d\eta H \log\left(\frac{2KA_{\max}^2 \eta B_\theta}{\delta}\right) + \frac{144(\log A_{\max} + \eta H)}{\exp(\eta H)} K$$
$$+ 4(\log A_{\max} + \eta H)\log \delta^{-1}.$$

Finally, choosing $\eta := \log(K)/H$, ensures that

$$\sum_{k=1}^{K} \sum_{x \in \mathcal{X}} \nu_h^{\pi_{\mathrm{E}}^n, \pi^{-n,k}}(x) D_{\mathrm{Hel}}^2(\pi_{\mathrm{out},h}^n, \pi_{\mathrm{E},h}^n)(x) \leq \widetilde{\mathcal{O}}\left(d\log\left(\frac{KA_{\max}B_\theta}{\delta}\right)\right).$$

The proof is concluded plugging the last result back into (6). □

Finally, we are ready to prove our main result for interactive MAIL in Linear MGs.

### E.4. Proof of Theorem 4.1

By Lemma 4.3, we have that with probability at least $1 - \delta$, it holds that

$$\mathrm{NG}(\pi_{\mathrm{out}}) \leq H \max_{n \in \{1,2\}} \sum_{h=1}^{H} \max_{\pi^{-n} \in \Pi^{-n}} \left\|\varphi_h^{\pi_{\mathrm{E}}^n, \pi^{-n}}\right\|_{(\Lambda_{h-1}^{-n,K})^{-1}} \mathcal{O}\left(\sqrt{\Delta \mathrm{TV}_{n,h}^2(\pi_{\mathrm{out}}) + B^2 d\log(K/\delta)}\right).$$

Then, by invoking Lemma 4.4 and a union bound we have that with probability at least $1 - 2\delta$,

$$\mathrm{NG}(\pi_{\mathrm{out}}) \leq H\widetilde{\mathcal{O}}\left(\sqrt{\frac{d^3 H^4 B^2 \log(dKH/\delta)}{K}}\right) \mathcal{O}\left(\sqrt{\Delta \mathrm{TV}_{n,h}^2(\pi_{\mathrm{out}}) + B^2 d\log(K/\delta)}\right)$$
$$= \widetilde{\mathcal{O}}\left(\sqrt{\frac{d^3 H^6 B^2 \log(dKH/\delta)\left(\Delta \mathrm{TV}_{n,h}^2(\pi_{\mathrm{out}}) + B^2 d\log(K/\delta)\right)}{K}}\right).$$

Finally, invoking Lemma 4.5 and a final union bound we obtain that with probability at least $1 - 3\delta$ it holds that

$$\mathrm{NG}(\pi_{\mathrm{out}}) \leq \widetilde{\mathcal{O}}\left(\sqrt{\frac{d^3 H^6 B^2 \log(dKH/\delta)\left(d\log(KA_{\max}B_\theta/\delta) + B^2 d\log(K/\delta)\right)}{K}}\right)$$
$$= \widetilde{\mathcal{O}}\left(\sqrt{\frac{d^4 H^6 B^4 \log(dKH/\delta)\log(KA_{\max}B_\theta/\delta)}{K}}\right).$$

Therefore, choosing $K = \widetilde{\mathcal{O}}\left(\frac{d^4 H^6 B^4 \log(A_{\max}B_\theta/\delta)}{\varepsilon^2}\right)$ guarantees that $\mathrm{NG}(\pi_{\mathrm{out}}) = \mathcal{O}(\varepsilon)$ with probability at least $1 - \delta$.

Moving forward, we generalize the analysis of LSVI-UCB-ZERO-BC in two directions: (i) allowing arbitrary number of players $N$ and (ii) studying infinite horizon discounted linear Markov games.

## F. Interactive MAIL in $N$-players Linear Markov games

We start by extending our interactive result to the $N$ players setting, at first, in the final horizon setting. Here, the idea is to pre-collect $K$ trajectories in the game interacting with the expert as explained in the first part of Algorithm 3 which serves as $N$-players extension of Algorithm 2.

Collecting the data in this way, guarantees that the covariance matrix output by the above algorithm guarantees that

$$\sum_{h=1}^{H} \max_{n' \in [N]} \max_{\pi^{n'} \in \Pi^{n'}} \left\|\varphi_{h-1}^{\pi^{n'}, \pi_{\mathrm{E}}^{-n'}}\right\|_{(\Lambda_{h-1}^{n',K})^{-1}},$$

---

**Algorithm 3** $N$-players LSVI-UCB-ZERO-BC

---

**for** $n \in [N]$ **do**

  **% Exploration phase**, i.e. LSVI-UCB-ZERO($\mathcal{M}^{-n}$, active $= n$)

  Initialize the MDP with dynamics $P^{\pi_{\mathrm{E}}^{-n}}$.

  Initialize $\Lambda_h^{n,1} = I$.

  Initialize policy $\pi_h^{n,1}(\cdot|x) = \mathrm{Unif}(\mathcal{A})$ for all $x, h \in \mathcal{X} \times [H]$.

  **for** $k \in \{1, \ldots, K\}$ **do**

    Collect a trajectory $\tau_n^k = \left\{ (X_h^{n,k}, A_h^{n,k}) \right\}_{h=1}^H$ with $\pi_k$ in the MDP with dynamics $P^{\pi_{\mathrm{E}}^{-n}}$.

    **for** $n' \in [N]$ **do**

      Sample $A_{n,h}^{k,\mathrm{E}(n')} \sim \pi_{\mathrm{E},h}^{n'}(\cdot|X_h^{n,k})$

    **end for**

    Update covariance matrix $\Lambda_h^{n,k+1} = \Lambda_h^{n,k} + \varphi(X_h^{n,k}, A_h^{n,k})\varphi(X_h^{n,k}, A_h^{n,k})^\top$.

    Define the reward $r_h^{n,k+1}(x,a) = \|\varphi(x,a)\|_{(\Lambda_{n,h}^{k+1})^{-1}}$.

    Update policy to $\pi^{n,k+1}$ with one step of LSVI-UCB with reward $r^{n,k+1}$.

  **end for**

**end for**

Perform BC on the collected dataset

$$\pi_{\mathrm{out},h}^n = \arg\max_{\pi \in \Pi_{\mathrm{softlin}}} \sum_{k=1}^K \sum_{n'=1}^N \log \pi(A_{n',h}^{k,\mathrm{E}(n)}|X_h^{n',k}) \quad \forall n \in [N]$$

**Output** $\pi_{\mathrm{out}} = \left\{ \pi_{\mathrm{out},h}^n \right\}_{h=1,n=1}^{H,N}$.

---

which is the equivalent of $\mathcal{C}_{\varphi,\max}$ where $\Lambda_h^{n,K}$ replaces $\Lambda_{\mathrm{E},h}^n$, decreases at a $\mathcal{O}\left(K^{-1/2}\right)$ rate. We recall that this result is proven in Lemma E.1. At an intuitive level we recall, that the first part of Algorithm 3 aims at collecting a dataset which guarantees good coverage of all possible features direction.

Therefore, it remains to show a change of measure similar to the one presented in Lemma D.6 for the more general case of $N$ players. We present such result in Lemma F.1.

**Lemma F.1.** *It holds that*

$$\sum_{n=1}^N \max_{\pi_\star^n \in \Pi^n} \left\langle \nu_1, V_n^{\pi_\star^n, \pi_{\mathrm{out}}^{-n}} - V_n^{\pi_{\mathrm{out}}} \right\rangle$$

$$\leq NH \sum_{h=1}^H \max_{n' \in [N]} \max_{\pi^{n'} \in \Pi^{n'}} \left\| \varphi_{h-1}^{\pi^{n'}, \pi_{\mathrm{E}}^{-n'}} \right\|_{(\Lambda_{h-1}^{n',K})^{-1}}$$

$$\cdot \sum_{n'=1}^N \sqrt{\sum_{k=1}^K \sum_{x \in \mathcal{X}} P^{\pi_{\mathrm{E}}^{-n'}}(x|X_{h-1}^{n',k}, A_{h-1}^{n',k}) \mathrm{TV}^2(\pi_{\mathrm{out},h}^n, \pi_{\mathrm{E},h}^n)(x) + 4B^2}.$$

*Proof.* For any state dependent random variable $W : \mathcal{X} \to \mathbb{R}$ with steps similar to the proof of Lemma D.6 but using $\Lambda_h^{n,k}$ instead of $\Lambda_{\mathrm{E},h}^n$ we obtain for any policy $\pi^n$

$$\sum_{h=1}^H \mathbb{E}_{X \sim \nu_h^{\pi^n, \pi_{\mathrm{E}}^{-n}}} [W(X)] \leq \sum_{h=1}^H \left\| \varphi_{h-1}^{\pi^n, \pi_{\mathrm{E}}^{-n}} \right\|_{(\Lambda_{h-1}^{n,K})^{-1}} \left\| \sum_{x \in \mathcal{X}} M_{\pi_{\mathrm{E}}^{-n}}(x) W(x) \right\|_{\Lambda_{h-1}^{n,K}},$$

Then, we rewrite the second term as

$$
\left\| \sum_{x \in \mathcal{X}} M_{\pi_{\mathrm{E}}^{-n}}(x) W(x) \right\|_{\Lambda_{h-1}^{n,K}}^2 = \sum_{k=1}^{K} \sum_{x',a^n} \left( \sum_{x \in \mathcal{X}} P(x|x',a^n,\pi_{\mathrm{E}}^{-n}) W(x) \right)^2 \mathbb{1}_{\left\{ x',a^n = X_{n,h-1}^k, A_{n,h-1}^k \right\}}
$$

$$
+ \left\| \sum_{x \in \mathcal{X}} M_{\pi_{\mathrm{E}}^{-n}}(x) W(x) \right\|^2
$$

$$
\leq \sum_{k=1}^{K} \sum_{x',a^n} \sum_{x \in \mathcal{X}} P(x|x',a^n,\pi_{\mathrm{E}}^{-n}) W^2(x) \mathbb{1}_{\left\{ x',a^n = X_{n,h-1}^k, A_{n,h-1}^k \right\}}
$$

$$
+ B^2 W_{\max}^2
$$

$$
= \sum_{k=1}^{K} \sum_{x \in \mathcal{X}} P(x|X_{n,h-1}^k, A_{n,h-1}^k, \pi_{\mathrm{E}}^{-n}) W^2(x) + B^2 W_{\max}^2.
$$

At this point setting $W(x) = \mathrm{TV}^2(\pi_h^n, \pi_{\mathrm{E},h}^n)(x)$, we reuse the decomposition in Equation(1) and the same steps in the proof of Theorem D.5 to obtain that

$$
\sum_{n=1}^{N} \max_{\pi_\star^n \in \Pi^n} \left\langle \nu_1, V_n^{\pi_\star^n, \pi_{\mathrm{out}}^{-n}} - V_n^{\pi_{\mathrm{out}}} \right\rangle
$$

$$
\leq \sum_{n=1}^{N} \max_{\pi_\star^n \in \Pi^n} H \sum_{h=1}^{H} \left( \mathbb{E}_{X \sim \nu_h^{\pi_\star^n, \pi_{\mathrm{E}}^{-n}}} \left[ \sum_{n' \neq n} \mathrm{TV}(\pi_{\mathrm{out},h}^{n'}, \pi_{\mathrm{E},h}^{n'})(X) \right] + \mathbb{E}_{X \sim \nu_h^{\pi_{\mathrm{E}}}} \left[ \mathrm{TV}(\pi_{\mathrm{out},h}^n, \pi_{\mathrm{E},h}^n)(X) \right] \right)
$$

$$
= \sum_{n=1}^{N} \max_{\pi_\star^n \in \Pi^n} H \sum_{h=1}^{H} \left( \sum_{n' \neq n} \mathbb{E}_{X \sim \nu_h^{\pi_\star^{n'}, \pi_{\mathrm{E}}^{-n'}}} \left[ \mathrm{TV}(\pi_{\mathrm{out},h}^n, \pi_{\mathrm{E},h}^n)(X) \right] + \mathbb{E}_{X \sim \nu_h^{\pi_{\mathrm{E}}}} \left[ \mathrm{TV}(\pi_{\mathrm{out},h}^n, \pi_{\mathrm{E},h}^n)(X) \right] \right)
$$

$$
\leq \sum_{n=1}^{N} H \sum_{h=1}^{H} \max_{n' \in [N]} \max_{\pi^{n'} \in \Pi^{n'}} \left\| \varphi_{h-1}^{\pi^{n'}, \pi_{\mathrm{E}}^{-n'}} \right\|_{(\Lambda_{h-1}^{n',K})^{-1}}
$$

$$
\cdot \sum_{n'=1}^{N} \sqrt{ \sum_{k=1}^{K} \sum_{x \in \mathcal{X}} P^{\pi_{\mathrm{E}}^{-n'}}(x|X_{h-1}^{n',k}, A_{h-1}^{n',k}) \mathrm{TV}^2(\pi_{\mathrm{out},h}^n, \pi_{\mathrm{E},h}^n)(x) + B^2 W_{\max}^2 } \tag{7}
$$

The final statement follows from $W_{\max} = 2$. $\qquad\square$

Finally, we can prove our main result leveraging the two previous lemmas.

**Theorem F.2.** *Let $\pi_{\mathrm{out}} = (\pi_{\mathrm{out}}^1, \dots \pi_{\mathrm{out}}^N)$ be the output of Algorithm 3. Then, it satisfies that with probability $1 - \delta$*

$$
\sum_{n=1}^{N} \max_{\pi_\star^n \in \Pi^n} \left\langle \nu_1, V_n^{\pi_\star^n, \pi_{\mathrm{out}}^{-n}} - V_n^{\pi_{\mathrm{out}}} \right\rangle \leq \tilde{\mathcal{O}} \left( \frac{N^2 H^3 d^2 B^2 \log\left(A_{\max} B_\theta K \delta^{-1}\right)}{\sqrt{K}} \right).
$$

*Therefore, setting $K = \tilde{\mathcal{O}} \left( \frac{N^4 H^6 d^4 B^4 \log(A_{\max} B_\theta K \delta^{-1})}{\varepsilon^2} \right)$ ensures that $\pi_{\mathrm{out}}$ is a $\varepsilon$-approximate Nash equilibrium with probability $1 - \delta$. The total number of queries is therefore $K N^2 H = \tilde{\mathcal{O}} \left( \frac{N^6 H^7 d^4 B^4 \log(A_{\max} B_\theta K \delta^{-1})}{\varepsilon^2} \right)$.*

*Proof.* The missing step to prove this result is to find a high probability upper bound on (7) which can be shown to grow only logarithmically in $K$. To this end, we look for an upper bound of $\sum_{x \in \mathcal{X}} P^{\pi_{\mathrm{E}}^{-n'}}(x|X_{h-1}^{n',k}, A_{h-1}^{n',k}) \mathrm{TV}^2(\pi_{\mathrm{out},h}^n, \pi_{\mathrm{E},h}^n)(x)$ for any $n, n' \in [N] \times [N]$ in terms of its expected value $\sum_{k=1}^{K} \sum_{x \in \mathcal{X}} \nu_h^{\pi^{n',k}, \pi_{\mathrm{E}}^{-n'}}(x) \mathrm{TV}^2(\pi_{\mathrm{out},h}^n, \pi_{\mathrm{E},h}^n)(x)$ up to numerical multiplicative constant and additive logarithmic factors. Analogously to the two player case, we introduce the notation $\mathcal{D}_k = \left\{ X_{n,h}^s, A_{n,h}^s \right\}_{s=1,h=1,n=1}^{k-1,H,N}$ for all $h \in [H]$ and the filtration $\mathcal{F}_k$ which is the sigma algebra induced by $\mathcal{D}_k$. For any

fixed $W \in \mathcal{W}_h := \left\{ W : \mathcal{X} \to [0, 2] : W(x) = \mathrm{TV}(\pi, \pi_{\mathrm{E},h}^n)(x) | \pi \in \Pi_{\mathrm{softlin}}^n \right\}$. As aforementioned, $\mathrm{TV}^2(\pi_{\mathrm{out},h}^n, \pi_{\mathrm{E},h}^n)(x)$ is sadly not a $\mathcal{F}_k$-adapted random variable unless $k = K$ because the policy $\pi_{\mathrm{out},h}^n$ is computed using the whole dataset $\mathcal{D}_K$. Therefore, we can not apply Lemma J.1 directly but we need to pass through a covering argument, again.

At this point, consider the covering set $\mathcal{C}_\epsilon(\mathcal{W}_h)$ defined such that

$$\forall W \in \mathcal{W}_h, \quad \exists \tilde{W} \in \mathcal{C}_\epsilon(\mathcal{W}_h) \quad \text{s.t.} \left\| W - \tilde{W} \right\|_\infty \leq \epsilon.$$

Then, let us fix a particular $\tilde{W}$ and let us invoke Lemma J.1 for $X_k = \sum_{x \in \mathcal{X}} \frac{1}{4} P^{\pi_{\mathrm{E}}^{-n'}}(x | X_{h-1}^{n',k}, A_{h-1}^{n',k}) \tilde{W}^2(x)$ for all $n', h \in [N] \times [H]$ to obtain that with probability $1 - \delta$,

$$\sum_{k=1}^{K} \sum_{x \in \mathcal{X}} P^{\pi_{\mathrm{E}}^{-n'}}(x | X_{h-1}^{n',k}, A_{h-1}^{n',k}) \tilde{W}^2(x) \leq 2 \sum_{k=1}^{K} \mathbb{E} \left[ \sum_{x \in \mathcal{X}} P^{\pi_{\mathrm{E}}^{-n'}}(x | X_{h-1}^{n',k}, A_{h-1}^{n',k}) \tilde{W}^2(x) | \mathcal{D}_k \right] + 4 \log \left( \frac{1}{\delta} \right).$$

Then, via a union bound over $\mathcal{C}_\epsilon(\mathcal{W}_h)$ it holds that for all $\tilde{W} \in \mathcal{C}_\epsilon(\mathcal{W}_h)$ simultaneously it holds with probability $1 - \delta$ that

$$\sum_{k=1}^{K} \sum_{x \in \mathcal{X}} P^{\pi_{\mathrm{E}}^{-n'}}(x | X_{h-1}^{n',k}, A_{h-1}^{n',k}) \tilde{W}^2(x) \leq 2 \sum_{k=1}^{K} \mathbb{E} \left[ \sum_{x \in \mathcal{X}} P^{\pi_{\mathrm{E}}^{-n'}}(x | X_{h-1}^{n',k}, A_{h-1}^{n',k}) \tilde{W}^2(x) | \mathcal{D}_k \right]$$
$$+ 4 \log \left( \frac{|\mathcal{C}_\epsilon(\mathcal{W}_h)|}{\delta} \right).$$

Then, it follows that with probability at least $1 - \delta$ for all $W \in \mathcal{W}_h$ simultaneously that

$$\sum_{k=1}^{K} \sum_{x \in \mathcal{X}} P^{\pi_{\mathrm{E}}^{-n'}}(x | X_{h-1}^{n',k}, A_{h-1}^{n',k}) W^2(x) \leq \sum_{k=1}^{K} \sum_{x \in \mathcal{X}} P^{\pi_{\mathrm{E}}^{-n'}}(x | X_{h-1}^{n',k}, A_{h-1}^{n',k}) (W^2(x) - \tilde{W}^2(x))$$
$$+ 2 \sum_{k=1}^{K} \mathbb{E} \left[ \sum_{x \in \mathcal{X}} P^{\pi_{\mathrm{E}}^{-n'}}(x | X_{h-1}^{n',k}, A_{h-1}^{n',k}) \left( \tilde{W}^2(x) - W^2(x) \right) | \mathcal{D}_k \right]$$
$$+ 2 \sum_{k=1}^{K} \mathbb{E} \left[ \sum_{x \in \mathcal{X}} P^{\pi_{\mathrm{E}}^{-n'}}(x | X_{h-1}^{n',k}, A_{h-1}^{n',k}) W^2(x) | \mathcal{D}_k \right]$$
$$+ 4 \log \left( \frac{|\mathcal{C}_\epsilon(\mathcal{W}_h)|}{\delta} \right)$$
$$\leq 2 W_{\max} \epsilon K + 2 \sum_{k=1}^{K} \mathbb{E} \left[ \sum_{x \in \mathcal{X}} P^{\pi_{\mathrm{E}}^{-n'}}(x | X_{h-1}^{n',k}, A_{h-1}^{n',k}) W^2(x) | \mathcal{D}_k \right]$$
$$+ 4 \log \left( \frac{|\mathcal{C}_\epsilon(\mathcal{W}_h)|}{\delta} \right).$$

Then using Lemma J.2 to bound the covering number $|\mathcal{C}_\epsilon(\mathcal{W}_h)|$, we conclude that for any $W \in \mathcal{W}_h$ with probability at least $1 - \delta$, it holds that

$$\sum_{k=1}^{K} \sum_{x \in \mathcal{X}} P^{\pi_{\mathrm{E}}^{-n'}}(x | X_{h-1}^{n',k}, A_{h-1}^{n',k}) W^2(x) \leq 2 W_{\max} \epsilon K + 2 \sum_{k=1}^{K} \mathbb{E} \left[ \sum_{x \in \mathcal{X}} P^{\pi_{\mathrm{E}}^{-n'}}(x | X_{h-1}^{n',k}, A_{h-1}^{n',k}) W^2(x) | \mathcal{D}_k \right]$$
$$+ 4d \log \left( \frac{1 + \frac{4 \eta A_{\max} B_\theta}{\epsilon}}{\delta} \right).$$

Now, setting $\epsilon = B^2/(W_{\max}K)$ and choosing $W = \mathrm{TV}(\pi_{\mathrm{out},h}^n, \pi_{\mathrm{E},h}^n)$ we obtain that with probability at least $1 - \delta$

$$
\begin{aligned}
&\sum_{n=1}^N \max_{\pi_\star^n \in \Pi^n} \left\langle \nu_1, V_n^{\pi_\star^n, \pi_{\mathrm{out}}^{-n}} - V_n^{\pi_{\mathrm{out}}} \right\rangle \\
&\leq \sum_{n=1}^N H \sum_{h=1}^H \max_{n' \in [N]} \max_{\pi^{n'} \in \Pi^{n'}} \left\| \varphi_{h-1}^{\pi^{n'}, \pi_{\mathrm{E}}^{-n'}} \right\|_{(\Lambda_{h-1}^{n',K})^{-1}} \\
&\quad \cdot \sum_{n'=1}^N \sqrt{\sum_{k=1}^K \sum_{x \in \mathcal{X}} P^{\pi_{\mathrm{E}}^{-n'}}(x | X_{h-1}^{n',k}, A_{h-1}^{n',k}) \mathrm{TV}^2(\pi_{\mathrm{out},h}^n, \pi_{\mathrm{E},h}^n)(x) + 4B^2} \\
&\leq \sum_{n=1}^N H \sum_{h=1}^H \max_{n' \in [N]} \max_{\pi^{n'} \in \Pi^{n'}} \left\| \varphi_{h-1}^{\pi^{n'}, \pi_{\mathrm{E}}^{-n'}} \right\|_{(\Lambda_{h-1}^{n',K})^{-1}} \\
&\quad \cdot \sum_{n'=1}^N \sqrt{2 \sum_{k=1}^K \mathbb{E}\left[ \sum_{x \in \mathcal{X}} P^{\pi_{\mathrm{E}}^{-n'}}(x | X_{h-1}^{n',k}, A_{h-1}^{n',k}) \mathrm{TV}^2(\pi_{\mathrm{out},h}^n, \pi_{\mathrm{E},h}^n)(x) | \mathcal{D}_k \right] + 8B^2 + 4d \log\left( \frac{1 + 4\eta A_{\max} B_\theta K}{\delta} \right)} \\
&\leq \sum_{n=1}^N H \sum_{h=1}^H \max_{n' \in [N]} \max_{\pi^{n'} \in \Pi^{n'}} \left\| \varphi_{h-1}^{\pi^{n'}, \pi_{\mathrm{E}}^{-n'}} \right\|_{(\Lambda_{h-1}^{n',K})^{-1}} \\
&\quad \cdot \sum_{n'=1}^N \sqrt{2 \sum_{k=1}^K \sum_{x \in \mathcal{X}} \nu_h^{\pi^{n',k}, \pi_{\mathrm{E}}^{-n'}}(x) \mathrm{TV}^2(\pi_{\mathrm{out},h}^n, \pi_{\mathrm{E},h}^n)(x) + 8B^2 + 4d \log\left( \frac{1 + 4\eta A_{\max} B_\theta K}{\delta} \right)}. \quad (8)
\end{aligned}
$$

Now, let us recall that the output policy of Algorithm 3 is defined as

$$
\pi_{\mathrm{out},h}^n = \arg\max_{\pi \in \Pi_{\mathrm{softlin}}} \sum_{k=1}^K \sum_{n'=1}^N \log \pi(A_{n',h}^{k,\mathrm{E}(n)} | X_h^{n',k}).
$$

Therefore, we can bound $\sum_{k=1}^K \sum_{x \in \mathcal{X}} \nu_h^{\pi^{n',k}, \pi_{\mathrm{E}}^{-n'}}(x) \mathrm{TV}^2(\pi_{\mathrm{out},h}^n, \pi_{\mathrm{E},h}^n)(x)$ similarly to the proof of Theorem D.5 adapted to account for the fact that now the states $X_h^{n',k}$ are not i.i.d. distributed. However, upper-bounding the total variation by the Hellinger distance, i.e.

$$
2 \sum_{k=1}^K \sum_{x \in \mathcal{X}} \nu_h^{\pi^{n',k}, \pi_{\mathrm{E}}^{-n'}}(x) \mathrm{TV}^2(\pi_{\mathrm{out},h}^n, \pi_{\mathrm{E},h}^n)(x) \leq 8 \sum_{k=1}^K \sum_{x \in \mathcal{X}} \nu_h^{\pi^{n',k}, \pi_{\mathrm{E}}^{-n'}}(x) D_{\mathrm{Hel}}^2(\pi_{\mathrm{out},h}^n, \pi_{\mathrm{E},h}^n)(x),
$$

and applying Lemma J.4, we obtain with probability $1 - 2\delta$

$$
\begin{aligned}
8 \sum_{k=1}^K \sum_{x \in \mathcal{X}} \nu_h^{\pi^{n',k}, \pi_{\mathrm{E}}^{-n'}}(x) D_{\mathrm{Hel}}^2(\pi_{\mathrm{out},h}^n, \pi_{\mathrm{E},h}^n)(x) &\leq 16\epsilon K + 16 \log \frac{|\mathcal{C}_\epsilon(\log \Pi_{\mathrm{softlin}})|}{\delta} + \frac{1152(\log A_{\max} + \eta H)}{\exp(\eta H)} K \\
&\quad + 32(\log A_{\max} + \eta H) \log \frac{1}{\delta}. \quad (9)
\end{aligned}
$$

Now, again similarly to the proof of Theorem D.5 by applying Lemma J.3 to bound the covering number of the log policy class and the choice of $\epsilon = K^{-1}$, we can bound it with

$$
(9) \leq 16 + 16 d\eta H \log\left( \frac{2K A_{\max}^2 \eta B_\theta}{\delta} \right) + \frac{1152(\log A_{\max} + \eta H)}{\exp(\eta H)} K + 32(\log A_{\max} + \eta H) \log \frac{1}{\delta}. \quad (10)
$$

Next, we choose $\eta := \log(K)/H$. Plugging this into (10) gives

$$(10) \leq 16 + 16d\log(K)\log\left(\frac{2KA_{\max}^2\log(K)B_\theta}{H\delta}\right) + 1152(\log A_{\max} + \log(K))$$
$$+ 32(\log A_{\max} + \log(K))\log\frac{1}{\delta}$$
$$\leq \widetilde{\mathcal{O}}\left(d\log\left(\frac{KA_{\max}B_\theta}{\delta}\right)\right).$$

Therefore, plugging into (8), we obtain

$$\sum_{n=1}^{N} \max_{\pi_\star^n \in \Pi^n} \left\langle \nu_1, V_n^{\pi_\star^n, \pi_{\mathrm{out}}^{-n}} - V_n^{\pi_{\mathrm{out}}} \right\rangle$$
$$\leq \tilde{\mathcal{O}}\left( N^2 H \sum_{h=1}^{H} \max_{n' \in [N]} \max_{\pi^{n'} \in \Pi^{n'}} \left\| \varphi_{h-1}^{\pi^{n'}, \pi_{\mathrm{E}}^{-n'}} \right\|_{(\Lambda_{h-1}^{n',K})^{-1}} \sqrt{dB^2\log(A_{\max}B_\theta K\delta^{-1})} \right)$$

Then, using Lemma E.1 to obtain

$$\sum_{n=1}^{N} \max_{\pi_\star^n \in \Pi^n} \left\langle \nu_1, V_n^{\pi_\star^n, \pi_{\mathrm{out}}^{-n}} - V_n^{\pi_{\mathrm{out}}} \right\rangle \leq \tilde{\mathcal{O}}\left( N^2 H \mathcal{O}\left( H^2 d^{3/2} B \sqrt{\frac{\log(K\delta^{-1})}{K}} \right) \sqrt{dB^2\log(A_{\max}B_\theta K\delta^{-1})} \right)$$
$$= \tilde{\mathcal{O}}\left( \frac{N^2 H^3 d^2 B^2 \log\left(A_{\max}B_\theta K\delta^{-1}\right)}{\sqrt{K}} \right).$$

Therefore, choosing $K = \widetilde{\mathcal{O}}\left(\frac{N^4 H^6 d^4 B^4 \log(A_{\max}B_\theta\delta^{-1})}{\varepsilon^2}\right)$ ensures that the output of Algorithm 3 is an $\varepsilon$-approximate Nash equilibrium. $\qquad\square$

## G. Extension to infinite horizon games

When the horizon $H$ grows large, learning a non stationary policy becomes more and more costly. In those situation, it is more attractive to learn a stationary policy in discounted Markov Games defined as follows.

**Infinite Horizon Discounted $N$-players Markov games.** In general, a Infinite Horizon Discounted $N$-players Markov game is defined as the tuple $\mathcal{G} = (\mathcal{X}, \{\mathcal{A}^n\}_{n=1}^N, \gamma, r, P, \nu_1)$, where $\mathcal{X}$ is the state space, $\mathcal{A}^n$ is the action space of player $n$, and $\mathcal{A}$ is the joint action space defined as the product space of the individual action spaces $\mathcal{A} := \times_{n=1}^N \mathcal{A}^n$, $\gamma$ is the discount factor, and $P : \mathcal{X} \times \mathcal{A} \to \Delta_{\mathcal{X}}$ is a transition kernel. In other words, $P(x' \mid x, \{a^n\}_{n=1}^N)$ denotes the probability of landing in $x'$ from $x$ under the joint action $a = [a^1, \ldots, a^N]$ and a reward $r^n : \mathcal{X} \times \mathcal{A} \to \mathbb{R}$ for $n \in [N]$ with $r^n(x,a) \in [-1,1]$. Additionally, we denote the initial state distribution $\nu_1 \in \Delta_{\mathcal{X}}$.

**Proof technique for the infinite horizon setting.** The finite horizon proof technique of creating an exploratory dataset via online learning to maximize the reward sequence $\|\varphi(\cdot, \cdot)\|_{\Lambda_{k,n}^{-1}}$ generalizes to the infinite horizon setting changing the regret minimization algorithm from LSVI-UCB (Jin et al., 2019) to RMAX-RAVI-LSVI-UCB (Moulin et al., 2025b). However, RMAX-RAVI-LSVI-UCB can not naively tolerate quadratic reward. Indeed, RMAX-RAVI-LSVI-UCB uses softmax updates instead of greedy updates and needs to use slow-changing bonuses function to ensure that the policy class is parameterized by at most $d^2$ parameters. Analogously, we can not update the reward at each step, otherwise the policies played by the algorithm would be parameterized by $Kd^2$ parameters. This fact would invalidate the proof technique of Moulin et al. (2025b), leading to linear regret. As a remedy, we use slow-changing reward functions as specified in the following algorithm. A careful inspection of the RMAX-RAVI-LSVI-UCB analysis reveals that the reward sequence used in Algorithm 4 matches the bonus sequence in the (non-optimistically augmented) MDPs in (Moulin et al., 2025b). As a consequence, even if the rewards are non-linear, they do not change the complexity of the class realizing the policies which can be potentially produced by RMAX-RAVI-LSVI-UCB. These observation leads to the following result.

---

**Algorithm 4** $N$-players LSVI-UCB-ZERO-BC for Infinite Horizon Discounted Linear Games

---

**for** $n \in [N]$ **do**

    Initialize the MDP with dynamics $P^{\pi_{\mathrm{E}}^{-n}}$.

    Initialize $\Lambda_n^1 = I$.

    $\mathcal{D} = \emptyset$

    Set reward update index $e = 1$ and slow changing variance matrix $\bar{\Lambda}_n^e = I$.

    Initialize policy $\pi_n^1(\cdot|x) = \mathrm{Unif}(\mathcal{A})$ for all $x, h \in \mathcal{X} \times [H]$.

    **for** $k \in \{1, \ldots, K\}$ **do**

        Sample horizon $H \sim \mathrm{Geom}(1-\gamma)$

        Collect a trajectory $\tau_n^k = \left\{ (X_h^{n,k}, A_h^{n,k}) \right\}_{h=1}^H$ rolling out $\pi_k$ in the MDP with dynamics $P^{\pi_{\mathrm{E}}^{-n}}$.

        **for** $n' \in [N]$ **do**

            Sample $A_{n,h}^{k,\mathrm{E}(n')} \sim \pi_{\mathrm{E},h}^{n'}(\cdot|X_h^{n,k})$

            $\mathcal{D} \leftarrow \mathcal{D} \cup \left\{ X_h^{n,k}, A_{n,h}^{k,\mathrm{E}(n')} \right\}$

        **end for**

        Update covariance matrix $\Lambda^{n,k+1} = \Lambda^{n,k} + \sum_{X,A \in \tau_n^k} \varphi(X,A)\varphi(X,A)^\top$.

        **if** $\det(\Lambda^{n,k+1}) \geq 2\det(\bar{\Lambda}_n^e)$ **then**

            $r^{n,k+1}(x,a) = \|\varphi(x,a)\|_{(\Lambda^{n,k+1})^{-1}}$.

            $\bar{\Lambda}_n^{e+1} = \Lambda^{n,k+1}$

            $e \leftarrow e + 1$

        **else**

            $r^{n,k+1}(x,a) = \|\varphi(x,a)\|_{(\bar{\Lambda}_n^e)^{-1}}$

        **end if**

        Update $\pi^{n,k+1}$ with one step of RMAX-RAVI-LSVI-UCB (Moulin et al., 2025b) with reward $r^{n,k+1}$ using $\mathcal{D}$.

    **end for**

**end for**

Perform BC on the collected dataset ($\mathcal{D}$)

$$\pi_{\mathrm{out}}^n = \arg\max_{\pi \in \Pi_{\mathrm{softlin}}^n} \sum_{k=1}^K \sum_{n'=1}^N \sum_{h=1}^H \log \pi(A_{n',h}^{k,\mathrm{E}(n)}|X_h^{n',k})$$

**Output** $\{\pi_{\mathrm{out}}^n\}_{n=1}^N$.

---

**Lemma G.1.** *It holds with probability $1 - \delta$ that*

$$\frac{\max_{n' \in [N], \pi^{n'} \in \Pi^{n'}} \left\| \varphi^{\pi^{n'}, \pi_{\mathrm{E}}^{-n'}} \right\|_{(\Lambda^{n',K})^{-1}}}{1 - \gamma} \leq \mathcal{O}\left( \sqrt{\frac{d^3 B^2 \log(A_{\max}\delta^{-1})}{(1-\gamma)^{9/2} K}} \right).$$

*Proof.* Since the argument holds equally for all players, we omit the index $n$ from the proof. Invoking Moulin et al. (2025b, Theorem 1), we obtain

$$\mathrm{Regret}(\pi_\star) := (1-\gamma)^{-1} \sum_{k=1}^K \left\langle \mu^{\pi_\star}, r^k \right\rangle - \left\langle \mu^{\pi^k}, r^k \right\rangle \leq \mathcal{O}\left( \sqrt{d^3 (1-\gamma)^{-9/2} B^2 K \log(A_{\max}\delta^{-1})} \right),$$

where $H = (1-\gamma)^{-1}$ in this context. Then, rearranging yields

$$(1-\gamma)^{-1} \sum_{k=1}^K \left\langle \mu^{\pi_\star}, r^k \right\rangle \leq \mathcal{O}\left( \sqrt{d^3 (1-\gamma)^{-9/2} B^2 K \log(A_{\max}\delta^{-1})} \right) + (1-\gamma)^{-1} \sum_{k=1}^K \left\langle \mu^{\pi^k}, r^k \right\rangle.$$

It remains now to bound the sum of the rewards on the right hand side. Recalling again the analogy between rewards and

bonuses under this setting we can invoke Moulin et al. (2025b, Lemma 7) with $\lambda = 1$ to obtain

$$\sum_{k=1}^{K} \left\langle \mu^{\pi^k}, r^k \right\rangle \leq 4(1-\gamma)B\sqrt{dKH \log\left(1 + \frac{B^2 KH}{d}\right)} + 4B \log\left(\frac{2K}{\delta}\right)^2.$$

Therefore, this implies

$$(1-\gamma)^{-1} \sum_{k=1}^{K} \left\langle \mu^{\pi_\star}, r^k \right\rangle \leq \mathcal{O}\left(\sqrt{d^3(1-\gamma)^{-9/2} B^2 K \log(A_{\max}\delta^{-1})}\right).$$

Finally, using again the decreasing property of the rewards, it holds that

$$(1-\gamma)^{-1} \left\langle \mu^{\pi_\star}, r^K \right\rangle \leq \mathcal{O}\left(\sqrt{\frac{d^3 B^2 \log(A_{\max}\delta^{-1})}{(1-\gamma)^{9/2} K}}\right).$$

Finally, by Jensen's inequality and the reward definition it holds that

$$\frac{\max_\pi \|\varphi^\pi\|_{(\Lambda^K)^{-1}}}{1-\gamma} \leq \mathcal{O}\left(\sqrt{\frac{d^3 B^2 \log(A_{\max}\delta^{-1})}{(1-\gamma)^{9/2} K}}\right).$$

The proof is then concluded stating the result considering the player index and taking the maximum over players. □

The final bound follows the same step as in the finite horizon case which leads to the following bound

**Theorem G.2.** *The output of Algorithm 4 satisfies that with probability $1 - \delta$*

$$\sum_{n=1}^{N} \max_{\pi_\star^n \in \Pi^n} \left\langle \nu_1, V_n^{\pi_\star^n, \pi_{\text{out}}^{-n}} - V_n^{\pi_{\text{out}}} \right\rangle \leq \widetilde{\mathcal{O}}\left(\sqrt{\frac{N^4 d^4 B^4 \log(A_{\max} B_\theta \delta^{-1})}{(1-\gamma)^{6.5} K}}\right).$$

*Therefore, setting $K = \widetilde{\mathcal{O}}\left(\frac{N^4 d^4 B^4 \log(A_{\max} B_\theta \delta^{-1})}{(1-\gamma)^{6.5} \varepsilon^2}\right)$ ensures that $\pi$ is an $\varepsilon$-Nash equilibrium. Moreover, since the total number of expert queries is $KN(1-\gamma)^{-1}$, the total queries amounts to $\widetilde{\mathcal{O}}\left(\frac{N^5 d^4 B^4 \log(A_{\max} B_\theta \delta^{-1})}{(1-\gamma)^{7.5} \varepsilon^2}\right)$*

*Proof.* The result follows again by Lemma G.1 and the change of measure argument in Lemma D.6 that extends naturally to the infinite horizon. Indeed, we have that the state occupancy measure in the discounted setting can be written as the following affine function in the features space

$$\nu^{\pi_\star^n, \pi_{\text{E}}^{-n}}(x) = \left\langle \varphi^{\pi_\star^n, \pi_{\text{E}}^{-n}}, \underbrace{\gamma M_{\pi_{\text{E}}^{-n}}(x) + (1-\gamma)\nu_1(x)\mathbf{1}}_{:=M_{\pi_{\text{E}}^{-n}}^\gamma} \right\rangle,$$

therefore using again Lemma J.1 and Lemma J.2, it holds that

$$\mathbb{E}_{X \sim \nu^{\pi_\star^n, \pi_{\text{E}}^{-n}}}[W(X)] \leq \frac{1}{(1-\gamma)^2} \max_{n' \in [N]} \max_{\pi^{n'} \in \Pi^{n'}} \left\| \varphi^{\pi^{n'}, \pi_{\text{E}}^{-n'}} \right\|_{(\Lambda^{n', K})^{-1}}$$

$$\cdot \sum_{n'=1}^{N} \sqrt{4 \sum_{k=1}^{K} \sum_{x \in \mathcal{X}} \nu^{\pi^{n', k}, \pi_{\text{E}}^{-n'}}(x) \text{TV}^2(\pi_{\text{out}}^n, \pi_{\text{E}}^n)(x) + 8B^2 + 4d \log(A_{\max} B_\theta K \log(HK)\delta^{-1})}.$$

The proof follows from steps analogous to Lemma D.6 with $M_{\pi_{\text{E}}^{-n}}^\gamma$ replacing $M_{\pi_{\text{E}}^{-n}}$. Then, summing over $n$ over the set $\{1, \dots, N\}$, upper-bounding the squared total variation via the squared Hellinger divergence and applying our MLE

concentration for martingale distributed data under misspecification Lemma J.4, we obtain that with probability $1 - \delta$, it holds that

$$
\sum_{n=1}^{N} \max_{\pi_\star^n \in \Pi^n} \left\langle \nu_1, V_n^{\pi_\star^n, \pi_{\text{out}}^{-n}} - V_n^{\pi_{\text{out}}} \right\rangle \leq \frac{N^2}{1 - \gamma} \frac{\max_{n' \in [N], \pi^{n'} \in \Pi^{n'}} \left\| \varphi^{\pi^{n'}, \pi_{\text{E}}^{-n'}} \right\|_{(\Lambda^{n'}, K)^{-1}}}{1 - \gamma} \sqrt{32 d B^2 \log \left( \frac{A_{\max} B_\theta K \log(KH)}{\delta} \right)}
$$

$$
\leq \frac{N^2}{1 - \gamma} \mathcal{O} \left( \sqrt{\frac{d^3 B^2 \log(A_{\max} \delta^{-1})}{(1 - \gamma)^{9/2} K}} \right) \sqrt{32 d B^2 \log(A_{\max} B_\theta K \log(KH) \delta^{-1})}
$$

$$
\leq \mathcal{O} \left( \sqrt{\frac{N^4 d^4 B^4 \log(A_{\max} B_\theta \delta^{-1})}{(1 - \gamma)^{6.5} K}} \right).
$$

$\square$

## H. Omitted pseudocode for Deep multi-agent Imitation Learning

In this section, we provide the complete pseudo-code for the deep MAIL algorithm. In particular, we consider the following algorithm 5. For simplicity we present the simpler two player case.

---

**Algorithm 5** DQN-EXPLORE-BC for 2-players game

---

**Require** Number of warm up iterations $K$.
% Exploration Phase
% Loop over the players
**for** $n \in [1, 2]$ **do**
    Initialize the DQN Network $Q_n(\cdot) = \text{LastLayer}_n(\text{InitialLayers}_n(\cdot))$ and the corresponding target network.
    Create a copy of the initial layers for the reward function $\text{OldInitialLayers}_n(\cdot)) = \text{InitialLayers}_n(\cdot)$
    Replay Buffer $\mathcal{D}^n$ with fixed capacity S
    Initialize $\Lambda = \lambda I$
    **for** $k \in [K]$ **do**
        % Update the exploratory reward function
        Set the reward function $r_{\text{expl}}(x, a) = \|\text{OldInitialLayers}_n(x, a)\|_{\Lambda^{-1}}$
        $\Lambda_{\text{new}} = \Lambda$
        **while** $\text{logdet}(\Lambda_{\text{new}}) \leq \text{logdet}(\Lambda) + \log(2)$ **do**
            Collect a trajectory $\tau$ rolling out $\pi^n$ (and the opponent playing according to a Nash) and add it to $\mathcal{D}^n$.
            Using $\mathcal{D}^n$ and use DQN to update $\text{LastLayer}_n$ and $\text{InitialLayers}_n$ by solving approximately

$$
\underset{\text{Layers}}{\arg\min} \sum_{x, a, x' \in \mathcal{D}_n} \left( r_{\text{expl}}(x, a) + \gamma \max_{a' \in \mathcal{A}} \text{OldLayers}(x', a') - \text{Layers}(x, a) \right)^2
$$

            Set $Q_n(x, a) = \text{LastLayer}_n(\text{OldInitialLayers}_n(x, a))$
            $\pi^n(\cdot | x) = \text{softmax}(Q_n(x, \cdot))$ for all $x \in \mathcal{X}$.
            $\Lambda_{\text{new}} = \Lambda_{\text{new}} + \sum_{x, a \in \mathcal{D}_n} \text{OldInitialLayers}_n(x, a) \text{OldInitialLayers}_n(x, a)^T$
        **end while**
        % Update the initial layers that will be used to update the exploratory reward function
        $\text{OldInitialLayers} \leftarrow \text{InitialLayers}$
        Compute $\Lambda = \sum_{x, a \in \mathcal{D}_n} \text{OldInitialLayers}(x, a) \text{OldInitialLayers}(x, a)^\top + \lambda I$.
    **end for**
**end for**
% Imitation Phase
Initialize BC networks.
Output $\pi_{\text{out}}^1 = \text{BC}\left( \mathcal{D}^2 \right)$ and $\pi_{\text{out}}^2 = \text{BC}\left( \mathcal{D}^1 \right)$

---

# I. Experiments

## I.1. Linear experiments

We next describe the setup of the considered zero-sum Gridworld environment. The environment is the same as the one used in Freihaut et al. (2025b). In particular, the state space is given by the joint positions of the two agents on a $3 \times 3$ grid. The agents can not take the same position at the same time. Formally,

$$\mathcal{X} = \{((i,j),(k,l)) \mid (i,j) \neq (k,l), \, i,j,k,l \in \{0,1,\ldots,2\}\},$$

which yields 72 states in total. The action space is identical for both agents and defined as $\mathcal{A}^1 = \mathcal{A}^2 = \{\text{left}, \text{right}, \text{up}, \text{down}\}$. The transition dynamics are deterministic. If an action leads an agent to run in a wall or the other agent, the position remains unchanged and otherwise the action transfers the agent to the respective neighboring cell.

The starting positions that can be considered are conditioned on the fact that both agents should have the same distance to the goal, which makes the game "fair" and the resulting Nash value is $0$. In particular, we fix the deterministic starting state $((1,0),(2,1))$, from which both players require exactly five steps to reach the goal. Note that multiple Nash equilibria exist. All of the existing Nash equilibria have in common that no path lets the other agent reach the goal first. For Figure 1 (a) we take a convex combination of Nash equilibrium strategies. To obtain different Nash equilibrium strategies, we run different initializations of zero-sum value iteration (Perolat et al., 2015). Note that the set of Nash equilibria is convex in zero-sum games, the resulting strategy is again a Nash equilibrium. For 1 (b) we instead used a deterministic Nash equilibrium strategy as the performance of BC remained unchanged.

**Linear feature maps.** We consider three different linear feature maps. In particular, to compare against the tabular version considered in Freihaut et al. (2025b), we use one hot encoded features for states and actions as described in Example 2.2, therefore we have features of dimension $d = 72 \times 4 = 288$. Additionally, we introduce a *relational feature map* of dimension $d = 80$. These features are designed to generalize across states by capturing relative geometric relationships rather than absolute positions. Specifically, for any state-action pair $(x, a)$, we construct a sparse binary vector $\varphi(x, a)$ based on the following relational concepts:

- Relative Direction (16 dims): The direction from the agent to the opponent (8 dims) and to the goal (8 dims), discretized into 8 sectors (N, NE, E, SE, S, SW, W, NW).

- Proximity (2 dims): Indicators for whether the agent is adjacent (within one step in any direction) to the opponent or the goal.

- Positional Status (2 dims): Indicators for whether the agent is currently located at the goal or in a corner of the grid.

These 20 binary concepts are computed for the current state and then embedded into the specific action slot, resulting in a total dimension of $d = 20 \times 4 = 80$. Finally, we evaluate a deep Linear Policy as the BC policy. This model is parameterized as a multi-layer feedforward network designed to improve representation learning while maintaining linear transformations. The architecture consists of a sequence of linear layers with decreasing hidden widths. Specifically, it maps the flattened state (9) to a hidden dimension of 12, followed by a Layer Normalization step. This is succeeded by a projection to a second hidden layer of dimension 6 (halved width), another Layer Normalization, and a final linear transformation to the 4 action logits. This results in a compact model with a total of 262 parameters.

**Parameters.** To run the experiments, we run them for three seeds, $[42, 123, 456, 789]$ and for the dataset sizes $10, 50, 100, 200, 500$, with 1000 BC epochs.

## I.2. Deep experiments.

**Environments.** We utilize the PettingZoo (Terry et al., 2021) implementations of Tic-Tac-Toe and Connect4. In both games, the observation state is represented as a stack of two binary planes ($C = 2$), where the first plane indicates the positions of Player 1's stones and the second plane indicates those of Player 2. Consequently, the input dimensions are $2 \times 3 \times 3$ for Tic-Tac-Toe and $2 \times 6 \times 7$ for Connect4. The action spaces are discrete, consisting of 9 actions for Tic-Tac-Toe (corresponding to the grid cells) and 7 actions for Connect4 (corresponding to the columns). Both environments are treated as zero-sum games with sparse rewards: agents receive a reward of $+1$ for winning, $-1$ for losing, and $0$ for a draw or intermediate steps. The horizon of Tic-Tac-Toe is $H = 9$ and $H = 42$ for Connect4.

**Experts.**    To derive the Nash equilibrium expert policies in Tic-Tac-Toe, we implement a computationally optimized minimax algorithm that exploits the symmetries of the Tic-Tac-Toe board. While the full state space consists of $3^9 = 19683$ configurations, the game is invariant under the dihedral group $D_4$ of the square, which includes four rotations and four reflections. We define a canonicalization mapping $\Phi : \mathcal{X} \to \mathcal{X}_{\text{canonical}}$ such that $\Phi(x) = \min_{g \in G} g(x)$, where $G$ is the group of valid transformations. By solving only for these canonical representatives, we reduce the decision space to exactly 765 unique legal states for the expert policy, making the storage as an array possible. We employ a recursive minimax search with memorization to compute the optimal policy $\pi_{\text{E}}$. Additionally to ensure the fastest victory for player 1 we weight the strategies by the length.

We facilitate optimal play in Connect4 by integrating BitBully, a high-performance solver that provides Nash equilibrium strategies. Implemented in C++ to leverage highly optimized bitboard operations and advanced tree-search pruning techniques, the library exposes efficient Python bindings. This allows us to query the optimal policy directly within our algorithm with minimal computational overhead.

**Parameters and Network architectures.**    As the number of warm up iterations, we choose $K = 30000$ for Connect4 and $K = 1000$. As the regularization parameter $\lambda$ that ensures that the matrix $\Lambda$ is invertible we choose $\lambda = 1.5$ for both experiments as this makes the inverse operation scalable. We repeat the experiments for three seeds: $[42, 123, 1]$.

For the exploration phase of `DQN-Explore` (Algorithm 5), we employ a Convolutional Neural Network architecture designed for state-action value estimation. The network structure is consistent across environments, with the exception of the fully connected hidden dimension size, which is set to 128 for Connect4 and 64 for Tic-Tac-Toe to account for the differing state space complexities. The fixed capacity of the replay buffer was set to $S = 10000$ for Connect4 and $S = 1000$ for Tic-Tac-Toe.

The architecture consists of three convolutional layers with $3 \times 3$ kernels, stride 1, and padding 1, ensuring the spatial dimensions ($H \times W$) remain preserved throughout the feature extraction. The channel depth increases sequentially from the input to 32, 64, and finally 128 channels. Consequently, the extracted state features are flattened into a vector of size $H \times W \times 128$.

For Q-value estimation, the flattened state features are first projected via a linear layer. This latent state representation is then concatenated with the action vector before being passed through two final fully connected layers. All hidden layers utilize ReLU activations. Optimization is performed using Adam with a learning rate of $\alpha = 0.01$. As a loss function we apply the Huber loss which is less sensitive to outliers (Raffin et al., 2021). When calling the network to receive the Q-values, we apply a clipping to avoid an overflow in the resulting softmax distribution common for DQN implementations (Raffin et al., 2021).

For the Behavioral cloning part of the Tic-Tac-Toe environments, we are using a small CNN network. In particular, we have the following network architecture: The model consists of three convolutional layers with $3 \times 3$ kernels, stride 1, and padding 1, with channel dimensions scaling as $2 \to 64 \to 128 \to 128$. Each convolutional layer is followed by a ReLU activation. The resulting feature map is flattened and passed to a fully connected layer with 256 units, followed by a ReLU activation and a Dropout layer ($p = 0.2$). The final linear layer projects the features to the 9 action logits. The learning rate is set to $0.01$, the loss function is a a Crossentropy function and we use an Adam optimizer. The batch size is set to 64 and the epochs to 100.

In Connect4, the state space is significantly more complex than Tic-Tac-Toe, comprising over $4 \times 10^{12}$ feasible board positions. To address this, we employ an advanced neural architecture inspired by AlphaGo (Silver et al., 2017). Specifically, we utilize a Residual Network (ResNet) adapted for the $6 \times 7$ board dimensions. The network begins with an initial convolutional block (128 filters, $3 \times 3$ kernel), followed by a backbone of 5 Residual Blocks. Each residual block consists of two $3 \times 3$ convolutional layers with Batch Normalization and ReLU activations, augmented by Squeeze-and-Excitation (SE) (Hu et al., 2019) modules to capture global channel-wise dependencies. The output is processed by a policy head which first reduces dimensionality via a $1 \times 1$ convolution (32 filters), followed by a fully connected hidden layer (512 units) that projects to the 7 action logits.

For training, we dynamically adjust the number of epochs based on the dataset size: 20 epochs for datasets with fewer than $50,000$ trajectories; 5 epochs for datasets between $50,000$ and $500,000$ trajectories; and 2 epochs for larger datasets. We use the Adam optimizer with a learning rate of $0.05$ and a batch size of $4096$, minimizing the Cross-Entropy loss.

Overall, note that the hyperparameters and network architectures could potentially be fine-tuned to increase the performance

of the algorithms.

**Opponents in Connect4.**   In this section, we detail the opponent configurations used to evaluate the robustness of the learned policy in Connect4, as summarized in Table 1.

Unlike the Grid-world experiments, computing an exact best response for the learned strategy is intractable due to the prohibitively large state space ($\approx 4 \times 10^{12}$ states). Furthermore, simply evaluating against the solver directly is insufficient to proxy true exploitability. As highlighted by Freihaut et al. (2025b), a policy trained via Behavior Cloning (BC) may perfectly memorize expert trajectories but remain fragile to distribution shifts. Instead the learned policy is potentially exploitable by first forcing it outside distribution, leading to potential mistakes and then playing the winning moves.

To address this, we construct an Approximate Best Response opponent designed to actively exploit the learner's generalization errors. This opponent leverages the action-value scores provided by the solver to switch between two modes. When the solver indicates the current state is a theoretical loss for the opponent (negative score), standard optimal play would simply prolong the game. Instead, our opponent randomizes over sub-optimal actions (excluding those that allow an immediate loss-in-one) to force the learner into unvisited states. The moment the learner commits an error making the solver scores positive, the opponent switches to deterministic perfect play to secure the win.

Additionally, to evaluate the policy against a spectrum of competence levels, we implement softmax opponents. These agents sample actions stochastically based on the solver's scores using a Boltzmann distribution. We test across temperature values $\{1, 2, 3, 4, 5\}$, where higher temperatures introduce greater stochasticity, simulating increasingly noisy and therefore potentially less optimal agents. To get a feeling of the optimality, we approximate the entropy of the opponent policies and report them in Table 2. Note that an agent has 7 actions and therefore a uniform policy would have an entropy of roughly 1.95.

*Table 2.* Estimated entropy of different opponents.

| OPPONENT | ENTROPY ESTIMATION |
|---|---|
| APPROX. BR | - |
| NOISE LVL 1 | $0.96 \pm 0.015$ |
| NOISE LVL 2 | $1.25 \pm 0.01$ |
| NOISE LVL 3 | $1.45 \pm 0.01$ |
| NOISE LVL 4 | $1.59 \pm 0.01$ |
| NOISE LVL 5 | $1.69 \pm 0.01$ |

**Notes on computational efficiency.**   In our proposed Algorithm 5, we calculate the inverse of the matrix $\Lambda$, which is derived from the frozen initial layers of the neural network. Crucially, this inversion is performed only within the outer loop, avoiding the frequent re-computation required by the inner while loop. Although matrix inversion is generally computationally expensive, the cost in our framework depends solely on the hidden dimension of the DQN network and does not scale directly with the number of samples or the size of the state space. Empirically, we observed that this operation was not a bottleneck. We investigated replacing the direct inversion with the Sherman-Morrison formula (Hager, 1989) to leverage the iterative updates. However, this yielded negligible performance gains. Consequently, we employ direct matrix inversion. For architectures with significantly larger hidden dimensions, the trade-offs between the stability of direct inversion and the efficiency of iterative schemes should be re-evaluated.

**Computational Resources.**   All numerical experiments and deep learning training phases were conducted on a compute node equipped with an NVIDIA A100 GPU (80GB VRAM) and an AMD chipset.

## I.3. Additional experiments

In this section, we provide additional experimental providing further insights into our methods. To start, we investigate if the exploration routine implemented with Explore-DQN can be replaced by simpler heuristics such as selecting an action uniformly at random in each state.

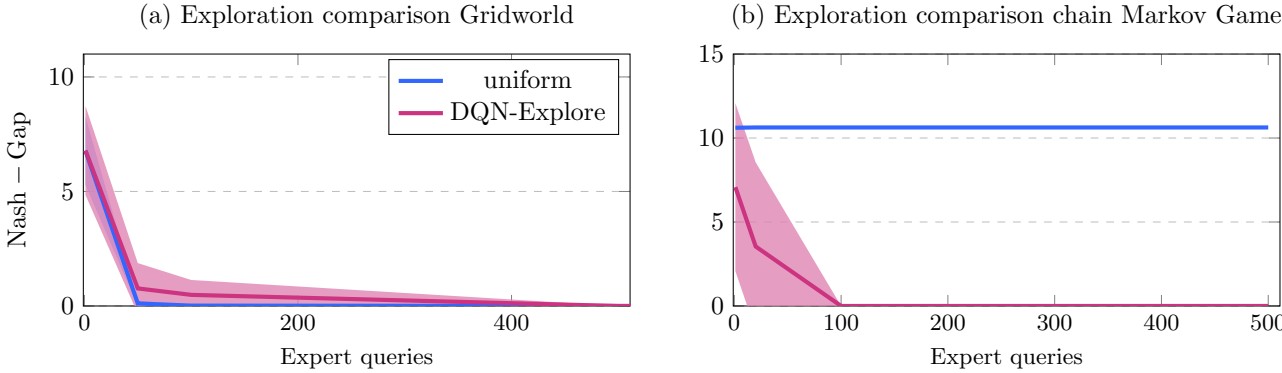

*Figure 2.* Comparison of exploration algorithms

### I.3.1. CAN THE EXPLORATION PHASE BE SIMPLIFIED?

To address this question, we compare the Explore-DQN-based exploration phase, which maximizes the exploratory reward in Algorithm 5, with a simpler heuristic that selects one action uniformly at random at each state. These two strategies differ in the sense that the uniform exploration strategy is memoryless: it plays all actions with equal probability without taking into account the information collected in previous rounds. On the contrary, the exploratory reward of actions for which a relevant amount of information has been previously collected decreases. This makes other actions more likely to be chosen under the DQN-Explore mechanism.

Intuitively, we expect significant differences to arise only in a hard environment where important states can be reached only when choosing one particular sequence of actions with length $H$. In this case, it is easy to see that the uniform exploration strategy requires in expectation $|\mathcal{A}|^H$ episodes before playing the informative sequence of actions. On the contrary, we expect the uniform heuristic to perform well on simpler, shorter-horizon tasks.

In order to confirm this insight, we compare the two exploration strategies in two different environments (Gridworld and Chain Markov Game) and report the results in Figure 2.

**The Gridworld experiment.** The Gridworld environment is based on a small 5x5 grid. Therefore, the horizon is short and it is easy for both exploration algorithms to collect a dataset that covers well any possible best response strategy for the opponent. This is well shown in Figure 2 (a) which shows that both algorithms converge quickly to zero exploitability. We can notice that in this setting the uniform exploration strategy is even slightly better than DQN-Explore. Therefore, in this case, we can conclude that simple heuristic strategies can be a valid practical replacement for the theory-based strategy implemented by DQN-Explore.

**The chain experiment** The same conclusion does not carry over to the experiment in the chain environment reported in Figure 2 (b). This environment is inspired by the lower bound construction in (Freihaut et al., 2025b) with the change that in order to reach the state in which a best responding agent can exploit the algorithms a particular sequence of length 8 should be chosen. The action space has size 2, so the expected number of trials that the uniform exploration needs to reach such state is $2^7 = 128$. The results in Figure 2 (b) confirm that this is a hard task for the uniform exploration strategy, while they show that DQN-Explore manages to successfully cover the best response. This implies that the policy trained via BC over the dataset collected via DQN-Explore successfully reaches 0 exploitability.

## J. Technical Lemmas

**Lemma J.1.** *Let $\{X_k\}_{k=1}^K$ be a sequence of positive random variables such that $\{X_k\}_{s=1}^{k-1}$ are $\mathcal{F}_k$-measurable and that $\forall k, X_k \leq 1$ almost surely, then it holds that,*

$$\mathbb{P}\left[\forall k \in [K] \quad \sum_{s=1}^k X_s \leq 2 \sum_{s=1}^k \mathbb{E}\left[X_s | \mathcal{F}_s\right] + \log\left(\frac{1}{\delta}\right)\right] \geq 1 - \delta.$$

*Proof.* Let us consider the sequence $\{Y_k\}_{k=1}^K$ such that $Y_k = \exp\left(\sum_{s=1}^k X_s - 2\mathbb{E}\left[X_s|\mathcal{F}_s\right]\right)$. We will now show that the sequence $Y_k$ is a supermartingale, that is $\mathbb{E}\left[Y_k|\mathcal{F}_k\right] \leq Y_{k-1}$. Towards this end, notice that

$$Y_k = Y_{k-1}\exp\left(X_k - 2\mathbb{E}\left[X_k|\mathcal{F}_k\right]\right).$$

Now, since $\{X_k\}_{s=1}^{k-1}$ are $\mathcal{F}_k$ measurable it follows that $Y_{k-1}$ is $\mathcal{F}_k$ measurable, therefore

$$\mathbb{E}\left[Y_k|\mathcal{F}_k\right] = Y_{k-1}\mathbb{E}\left[\exp\left(X_k - 2\mathbb{E}\left[X_k|\mathcal{F}_k\right]\right)|\mathcal{F}_k\right].$$

At this point, we need to establish that $\mathbb{E}\left[\exp\left(X_k - 2\mathbb{E}\left[X_k|\mathcal{F}_k\right]\right)|\mathcal{F}_k\right] \leq 1$.

$$\mathbb{E}\left[\exp\left(X_k - 2\mathbb{E}\left[X_k|\mathcal{F}_k\right]\right)|\mathcal{F}_k\right] \leq \mathbb{E}\left[1 + X_k - 2\mathbb{E}\left[X_k|\mathcal{F}_k\right] + (X_k - 2\mathbb{E}\left[X_k|\mathcal{F}_k\right])^2|\mathcal{F}_k\right]$$
$$= 1 - \mathbb{E}\left[X_k|\mathcal{F}_k\right] + \mathbb{E}\left[X_k^2|\mathcal{F}_k\right] - 4\mathbb{E}\left[X_k|\mathcal{F}_k\right]^2 + 4\mathbb{E}\left[X_k|\mathcal{F}_k\right]^2$$
$$= 1 - \mathbb{E}\left[X_k|\mathcal{F}_k\right] + \mathbb{E}\left[X_k^2|\mathcal{F}_k\right]$$
$$\leq 1.$$

The first and the last inequality hold because $X_k \leq 1$. In particular, for the first one we have that for all $x \leq 1$, $\exp(x) \leq 1 + x + x^2$ and in the last one $x^2 \leq x$ for all $x \in [0,1]$. All in all, this implies that $\mathbb{E}\left[Y_k|\mathcal{F}_k\right] \leq Y_{k-1}$ and therefore that $\{Y_k\}_{k=1}^K$ is a supermartingale. At this point, using the Ville's inequality we obtain that

$$\mathbb{P}\left[\exists k \ \text{ s.t. } \ Y_k \geq \frac{\mathbb{E}[Y_1]}{\delta}\right] \leq \delta.$$

Therefore, using that $\mathbb{E}[Y_1] = 1$ and the definition of $Y_k$

$$\mathbb{P}\left[\exists k \ \text{ s.t. } \ \exp\left(\sum_{s=1}^k X_s - 2\mathbb{E}\left[X_s|\mathcal{F}_s\right]\right) \geq \frac{1}{\delta}\right] \leq \delta.$$

Finally, rearranging yields

$$\mathbb{P}\left[\forall k \ \ \sum_{s=1}^k X_s \leq 2\sum_{s=1}^k \mathbb{E}\left[X_s|\mathcal{F}_s\right] + \log\frac{1}{\delta}\right] \geq 1 - \delta.$$

$\square$

**Lemma J.2.** *Covering number It holds that*

$$\mathcal{C}_\epsilon(\mathcal{W}_h) \leq \left(1 + \frac{4\eta A_{\max}B_\theta}{\epsilon}\right)^d.$$

*Proof.* Recall that
$$\mathcal{W}_h := \left\{W : \mathcal{X} \to [0,2] : W(x) = \mathrm{TV}(\pi, \pi_{\mathrm{E},h}^n)(x)|\pi \in \Pi_{\mathrm{softlin}}^n\right\}$$
Then, let us fix two arbitrary $\pi, \pi' \in \Pi_{\mathrm{softlin}}^n$ and notice that

$$\mathrm{TV}(\pi, \pi_{\mathrm{E},h}^n)(x) - \mathrm{TV}(\pi', \pi_{\mathrm{E},h}^n)(x) \leq \mathrm{TV}(\pi, \pi')(x)$$

At this point, if $\pi' \in \mathcal{C}_\epsilon(\Pi_{\mathrm{softlin}}^n)$ it holds that $\mathrm{TV}(\pi, \pi')(x) \leq \epsilon$ and therefore that

$$\mathrm{TV}(\pi, \pi_{\mathrm{E},h}^n)(x) - \mathrm{TV}(\pi', \pi_{\mathrm{E},h}^n)(x) \leq \epsilon.$$

This implies that $\mathcal{C}_\epsilon(\mathcal{W}_h) \leq \mathcal{C}_\epsilon(\Pi_{\mathrm{softlin}}^n)$. At this point, we can invoke Moulin et al. (2025a, Lemma 6) which gives

$$|\mathcal{C}_\epsilon(\Pi_{\mathrm{softlin}})| \leq \left(1 + \frac{4\eta A_{\max}B_\theta}{\epsilon}\right)^d.$$

and therefore $\mathcal{C}_\epsilon(\mathcal{W}_h) \leq \left(1 + \frac{4\eta A_{\max}B_\theta}{\epsilon}\right)^d.$

$\square$

**Lemma J.3.** *Covering number of the log policy class. It holds that*

$$|\mathcal{C}_\epsilon(\log \Pi_{\text{softlin}})| \leq \left(\frac{2A_{\max}^2 \exp(\eta H)\eta B_\theta}{\epsilon}\right)^d.$$

*Proof.* Notice that for any pair of policies $\pi, \pi' \in \Pi_{\text{softlin}}$ it holds that

$$\|\log \pi(\cdot|x) - \log \pi'(\cdot|x)\|_1 \leq \pi_{\min}^{-1} \|\pi(\cdot|x) - \pi'(\cdot|x)\|_1$$

$\square$

Therefore, an $\epsilon\pi_{\min}$ covering number for the policy class implies an $\epsilon$ covering number for the log space of the policy class. Therefore, recalling that $\pi_{\min} = \frac{1}{1+(A_{\max}-1)\exp(\eta H)}$ we need to compute the $\epsilon/(1+(A_{\max}-1)\exp(\eta H))$ covering number of the class $\Pi_{\text{softlin}}$ which using (Moulin et al., 2025a, Lemma 6) can be bounded as follows

$$|\mathcal{C}_\epsilon(\log \Pi_{\text{softlin}})| \leq |\mathcal{C}_{(1+(A_{\max}-1)\exp(\eta H))^{-1}\epsilon}(\Pi_{\text{softlin}})| \leq \left(1 + \frac{4(1+(A_{\max}-1)\exp(\eta H))\eta A_{\max}B_\theta}{\epsilon}\right)^d$$

$$\leq \left(\frac{2A_{\max}^2 \exp(\eta H)\eta B_\theta}{\epsilon}\right)^d.$$

**Lemma J.4.** *Let us consider a stochastic process $X_1, \ldots X_K$ and expert actions $A_i^E \sim \pi_{\text{E}}(X_i)$ such that $X_i$ is $\mathcal{F}_i^E = \sigma(X_1, A_1^E, \ldots, X_{i-1}, A_{i-1}^E)$ with $X_i|\mathcal{F}_i^E \sim \nu_i$, then the maximum likelihood estimator policy*

$$\pi_{\text{MLE}} = \text{argmax}_{\pi \in \Pi_{\text{softlin}}} \sum_{i=1}^K \log \pi(A_i^E|X_i)$$

*satisfies, with probability at least $1 - 2\delta$,*

$$\sum_{i=1}^K \mathbb{E}\left[D_{\text{Hel}}^2(\pi_{\text{E}}(\cdot|X_i), \pi_{\text{MLE}}(\cdot|X_i))|\mathcal{F}_i^E\right] \leq 2\epsilon K + 2\log\frac{|\mathcal{C}_\epsilon(\log \Pi_{\text{softlin}})|}{\delta} + \frac{144(\log A_{\max} + \eta H)}{\exp(\eta H)}K$$

$$+ 4(\log A_{\max} + \eta H)\log\frac{1}{\delta}.$$

*Proof.* Let $\Pi$ be a finite policy class, let us consider a distribution over $\Pi$ given by $\mathfrak{p}(\pi) = \frac{e^{g(\pi)}}{\sum_{\pi' \in \Pi} e^{g(\pi')}}$ where $g : \Pi \to \mathbb{R}$ is a function to be specified later. Then, for any other distribution over the policy class $\mathfrak{p}' \in \Delta_\Pi$ it holds that

$$0 \leq KL(\mathfrak{p}', \mathfrak{p})$$

$$= -\sum_{\pi \in \Pi} \mathfrak{p}'(\pi)g(\pi) + \log \sum_{\pi' \in \Pi} e^{g(\pi')} + \sum_{\pi \in \Pi} \mathfrak{p}'(\pi)\log\mathfrak{p}'(\pi)$$

$$= -\sum_{\pi \in \Pi} \mathfrak{p}'(\pi)g(\pi) + \log \sum_{\pi' \in \Pi} \frac{e^{g(\pi')}}{|\Pi|} + \sum_{\pi \in \Pi} \mathfrak{p}'(\pi)\log\mathfrak{p}'(\pi) + \log|\Pi|$$

$$\leq -\sum_{\pi \in \Pi} \mathfrak{p}'(\pi)g(\pi) + \log \sum_{\pi' \in \Pi} \frac{e^{g(\pi')}}{|\Pi|} + \log|\Pi|.$$

At this point, let us choose as $\Pi$ the $\epsilon$ covering set of the class $\log \Pi_{\text{softlin}}$, i.e. $\mathcal{C}_\epsilon(\log \Pi_{\text{softlin}})$ and let us define

$$\mathfrak{p}'(\pi) = \mathbb{1}_{\{\pi = \hat\pi^\epsilon\}} \quad \pi_{\text{MLE}} = \arg\max_{\pi \in \Pi_{\text{softlin}}} \sum_{i=1}^K \pi(A_i^E|X_i)$$

and $\hat\pi^\epsilon \in \mathcal{C}_\epsilon(\log \Pi_{\text{softlin}})$ such that $\max_{x \in \mathcal{X}} \|\log \pi_{\text{MLE}}(\cdot|x) - \log \hat\pi^\epsilon(\cdot|x)\|_1 \leq \epsilon$. At this point, let us specify $g(\pi) = -\sum_{i=1}^K \frac{1}{2}\log\frac{\pi_{\text{E}}(A_i^E|X_i)}{\pi(A_i^E|X_i)} - \sum_{i=1}^K \log\mathbb{E}\left[e^{-\frac{1}{2}\log\frac{\pi_{\text{E}}(\bar{A}_i^E|\bar{X}_i)}{\pi(\bar{A}_i^E|\bar{X}_i)}}\bigg|\mathcal{F}_i^E\right]$, where we defined the tangent sequence

$\bar{X}_1, \bar{A}_1^E, \ldots, \bar{X}_K, \bar{A}_K^E$ such that $\bar{X}_i \sim \mathbb{P}(\cdot | X_1, A_1^E, \ldots, X_{i-1}, A_{i-1}^E)$ where $\mathbb{P}(\cdot | X_1, A_1^E, \ldots, X_{i-1}, A_{i-1}^E)$ is the conditional distribution of the martingale sequence

$$X_1, A_1^E, \ldots, X_K, A_{\tau_E}^E.$$

Moreover, we have that $A_i^E \sim \pi_{\mathrm{E}}(\cdot | X_i)$ and $\bar{A}_i^E \sim \pi_{\mathrm{E}}(\cdot | \bar{X}_i)$. Therefore, we can notice that conditioned on the filtration $\mathcal{F}_i^E = \sigma(X_1, A_1^E, \ldots, X_{i-1}, A_{i-1}^E)$ the pairs $X_i, A_i^E$ and $\bar{X}_i, \bar{A}_i^E$ are identically and independently distributed. Notice here that we will apply this lemma in the interactive setting where the state sequence does not come from the expert occupancy measure but it is sampled from a sequence of exploratory distributions generated by the reward free exploration phase. For this reason, the sequence of states is denoted without the upper script E. Replacing the definitions of $\mathfrak{p}'$ and $g$ we can obtain

$$0 \le \frac{1}{2} \sum_{i=1}^K \log \frac{\pi_{\mathrm{E}}(A_i^E | X_i)}{\hat{\pi}^\epsilon(A_i^E | X_i)} + \log \sum_{\pi' \in \mathcal{C}_\epsilon(\log \Pi_{\mathrm{softlin}})} \frac{e^{-\sum_{i=1}^K \frac{1}{2} \log \frac{\pi_{\mathrm{E}}(A_i^E | X_i)}{\pi'(A_i^E | X_i)}}}{|\mathcal{C}_\epsilon(\log \Pi_{\mathrm{softlin}})| \prod_{i=1}^K \mathbb{E}\left[e^{-\frac{1}{2} \log \frac{\pi_{\mathrm{E}}(\bar{A}_i^E | \bar{X}_i)}{\pi'(\bar{A}_i^E | \bar{X}_i)}} \Big| \mathcal{F}_i^E\right]}$$
$$+ \log |\mathcal{C}_\epsilon(\log \Pi_{\mathrm{softlin}})| + \sum_{i=1}^K \log \mathbb{E}\left[e^{-\frac{1}{2} \log \frac{\pi_{\mathrm{E}}(\bar{A}_i^E | \bar{X}_i)}{\hat{\pi}^\epsilon(\bar{A}_i^E | \bar{X}_i)}} \Big| \mathcal{F}_i^E\right].$$

Rearranging, the above equation we obtain

$$-\frac{1}{2} \sum_{i=1}^K \log \frac{\pi_{\mathrm{E}}(A_i^E | X_i)}{\hat{\pi}^\epsilon(A_i^E | X_i)} - \log |\mathcal{C}_\epsilon(\log \Pi_{\mathrm{softlin}})| - \sum_{i=1}^K \log \mathbb{E}\left[e^{-\frac{1}{2} \log \frac{\pi_{\mathrm{E}}(\bar{A}_i^E | \bar{X}_i)}{\hat{\pi}^\epsilon(\bar{A}_i^E | \bar{X}_i)}} \Big| \mathcal{F}_i^E\right]$$
$$\le \log \sum_{\pi' \in \mathcal{C}_\epsilon(\log \Pi_{\mathrm{softlin}})} \frac{e^{-\sum_{i=1}^K \frac{1}{2} \log \frac{\pi_{\mathrm{E}}(A_i^E | X_i)}{\pi'(A_i^E | X_i)}}}{|\mathcal{C}_\epsilon(\log \Pi_{\mathrm{softlin}})| \prod_{i=1}^K \mathbb{E}\left[e^{-\frac{1}{2} \log \frac{\pi_{\mathrm{E}}(\bar{A}_i^E | \bar{X}_i)}{\pi'(\bar{A}_i^E | \bar{X}_i)}} \Big| \mathcal{F}_i^E\right]}.$$

Now, taking the exponential and the expectation with respect to the random variables $\{X_i, A_i^E\}_{i=1}^K$ we obtain

$$\mathbb{E}\left[e^{-\frac{1}{2} \sum_{i=1}^K \log \frac{\pi_{\mathrm{E}}(A_i^E | X_i)}{\hat{\pi}^\epsilon(A_i^E | X_i)} - \log |\mathcal{C}_\epsilon(\log \Pi_{\mathrm{softlin}})| - \sum_{i=1}^K \log \mathbb{E}\left[e^{-\frac{1}{2} \log \frac{\pi_{\mathrm{E}}(\bar{A}_i^E | \bar{X}_i)}{\hat{\pi}^\epsilon(\bar{A}_i^E | \bar{X}_i)}} \Big| \mathcal{F}_i^E\right]}\right]$$
$$\le \mathbb{E}\left[\sum_{\pi' \in \mathcal{C}_\epsilon(\log \Pi_{\mathrm{softlin}})} \frac{\prod_{i=1}^K \mathbb{E}\left[e^{-\frac{1}{2} \log \frac{\pi_{\mathrm{E}}(A_i^E | X_i)}{\pi'(A_i^E | X_i)}} \Big| \mathcal{F}_i^E\right]}{|\mathcal{C}_\epsilon(\log \Pi_{\mathrm{softlin}})| \prod_{i=1}^K \mathbb{E}\left[e^{-\frac{1}{2} \log \frac{\pi_{\mathrm{E}}(\bar{A}_i^E | \bar{X}_i)}{\pi'(\bar{A}_i^E | \bar{X}_i)}} \Big| \mathcal{F}_i^E\right]}\right] = 1.$$

Therefore, by the Chernoff bound, we have that

$$\mathbb{P}\left[-\frac{1}{2} \sum_{i=1}^K \log \frac{\pi_{\mathrm{E}}(A_i^E | X_i)}{\hat{\pi}^\epsilon(A_i^E | X_i)} - \log |\mathcal{C}_\epsilon(\log \Pi_{\mathrm{softlin}})| - \sum_{i=1}^K \log \mathbb{E}\left[e^{-\frac{1}{2} \log \frac{\pi_{\mathrm{E}}(\bar{A}_i^E | \bar{X}_i)}{\hat{\pi}^\epsilon(\bar{A}_i^E | \bar{X}_i)}} \Big| \mathcal{F}_i^E\right] \ge t\right] \le \frac{1}{e^t}.$$

Therefore, setting $t = \log(1/\delta)$, we have that with probability at least $1 - \delta$,

$$-\frac{1}{2} \sum_{i=1}^K \log \frac{\pi_{\mathrm{E}}(A_i^E | X_i)}{\hat{\pi}^\epsilon(A_i^E | X_i)} - \sum_{i=1}^K \log \mathbb{E}\left[e^{-\frac{1}{2} \log \frac{\pi_{\mathrm{E}}(\bar{A}_i^E | \bar{X}_i)}{\hat{\pi}^\epsilon(\bar{A}_i^E | \bar{X}_i)}} \Big| \mathcal{F}_i^E\right] \le \log \frac{|\mathcal{C}_\epsilon(\log \Pi_{\mathrm{softlin}})|}{\delta}.$$

Now, by applying Agarwal et al. (2020, Lemma 25), we have that

$$-\sum_{i=1}^{K} \log \mathbb{E}\left[e^{-\frac{1}{2}\log \frac{\pi_{\mathrm{E}}(\bar{A}_i^E|\bar{X}_i)}{\hat{\pi}^\epsilon(\bar{A}_i^E|\bar{X}_i)}}\Big|\mathcal{F}_i^E\right] \geq \frac{1}{2}\sum_{i=1}^{K}\mathbb{E}\left[D_{\mathrm{Hel}}^2(\pi_{\mathrm{E}}(\cdot|\bar{X}_i), \hat{\pi}^\epsilon(\cdot|\bar{X}_i))|\mathcal{F}_i^E\right]$$

$$= \frac{1}{2}\sum_{i=1}^{K}\mathbb{E}\left[D_{\mathrm{Hel}}^2(\pi_{\mathrm{E}}(\cdot|X_i), \hat{\pi}^\epsilon(\cdot|X_i))|\mathcal{F}_i^E\right]$$

$$= \frac{1}{2}\sum_{i=1}^{K}\mathbb{E}_{X\sim\nu_i}\left[D_{\mathrm{Hel}}^2(\pi_{\mathrm{E}}(\cdot|X), \hat{\pi}^\epsilon(\cdot|X))\right].$$

Now, setting $\hat{\pi}^\epsilon = \pi_{\mathrm{MLE}}^\epsilon$, we have that with probability at least $1-\delta$

$$\sum_{i=1}^{K}\mathbb{E}_{X\sim\nu_i}\left[D_{\mathrm{Hel}}^2(\pi_{\mathrm{E}}(\cdot|X), \pi_{\mathrm{MLE}}^\epsilon(\cdot|X))\right] \leq 2\log\frac{|\mathcal{C}_\epsilon(\log\Pi_{\mathrm{softlin}})|}{\delta} + \sum_{i=1}^{K}\log\frac{\pi_{\mathrm{E}}(A_i^E|X_i)}{\pi_{\mathrm{MLE}}^\epsilon(A_i^E|X_i)}.$$

Now, consider the following sequence of upper-bounds

$$\sum_{i=1}^{K}\mathbb{E}_{X\sim\nu_i}\left[D_{\mathrm{Hel}}^2(\pi_{\mathrm{E}}(\cdot|X), \pi_{\mathrm{MLE}}(\cdot|X))\right] \leq \sum_{i=1}^{K}\mathbb{E}_{X\sim\nu_i}\left[D_{\mathrm{Hel}}^2(\pi_{\mathrm{MLE}}(\cdot|X), \pi_{\mathrm{MLE}}^\epsilon(\cdot|X))\right]$$

$$+ \sum_{i=1}^{K}\mathbb{E}_{X\sim\nu_i}\left[D_{\mathrm{Hel}}^2(\pi_{\mathrm{E}}(\cdot|X), \pi_{\mathrm{MLE}}^\epsilon(\cdot|X))\right]$$

$$\leq \epsilon K + \sum_{i=1}^{K}\mathbb{E}\left[D_{\mathrm{Hel}}^2(\pi_{\mathrm{E}}(\cdot|X), \pi_{\mathrm{MLE}}^\epsilon(\cdot|X))\right]$$

$$\leq \epsilon K + 2\log\frac{|\mathcal{C}_\epsilon(\log\Pi_{\mathrm{softlin}})|}{\delta} + \sum_{i=1}^{K}\log\frac{\pi_{\mathrm{E}}(A_i^E|X_i)}{\pi_{\mathrm{MLE}}^\epsilon(A_i^E|X_i)}$$

$$\leq \epsilon K + 2\log\frac{|\mathcal{C}_\epsilon(\log\Pi_{\mathrm{softlin}})|}{\delta} + \sum_{i=1}^{K}\log\frac{\pi_{\mathrm{E}}(A_i^E|X_i)}{\pi_{\mathrm{MLE}}(A_i^E|X_i)}$$

$$+ \sum_{i=1}^{K}\log\frac{\pi_{\mathrm{MLE}}^\epsilon(A_i^E|X_i)}{\pi_{\mathrm{MLE}}(A_i^E|X_i)}$$

$$\leq 2\epsilon K + 2\log\frac{|\mathcal{C}_\epsilon(\log\Pi_{\mathrm{softlin}})|}{\delta} + \sum_{i=1}^{K}\log\frac{\pi_{\mathrm{E}}(A_i^E|X_i)}{\pi_{\mathrm{MLE}}(A_i^E|X_i)}.$$

If there, was no mispecification error than we would show that $\sum_{i=1}^{K}\log\frac{\pi_{\mathrm{E}}(A_i^E|X_i)}{\pi_{\mathrm{MLE}}(A_i^E|X_i)} \leq 0$ by the optimality of $\pi_{\mathrm{MLE}}$. However, since the expert Nash equilibrium might not be in $\Pi_{\mathrm{softlin}}$ we need to bound $\sum_{i=1}^{K}\log\frac{\pi_{\mathrm{E}}(A_i^E|X_i)}{\pi_{\mathrm{MLE}}(A_i^E|X_i)}$ in terms of the possible mispecification error. Towards this end, let us define $\bar{\pi} = \arg\min_{\pi\in\Pi_{\mathrm{softlin}}}\sum_{i=1}^{K}\mathbb{E}\left[D_{\mathrm{Hel}}^2(\pi(\cdot|X_i), \pi_{\mathrm{E}}(\cdot|X_i))|\mathcal{F}_i^E\right]$. Then, by definition of $\pi_{\mathrm{MLE}}$, we obtain

$$\sum_{i=1}^{K}\log\frac{\pi_{\mathrm{E}}(A_i^E|X_i)}{\pi_{\mathrm{MLE}}(A_i^E|X_i)} = \sum_{i=1}^{K}\log\pi_{\mathrm{E}}(A_i^E|X_i) - \max_{\pi\in\Pi_{\mathrm{softlin}}}\sum_{i=1}^{K}\log\pi(A_i^E|X_i)$$

$$\leq \sum_{i=1}^{K}\log\pi_{\mathrm{E}}(A_i^E|X_i) - \sum_{i=1}^{K}\log\bar{\pi}(A_i^E|X_i)$$

$$\leq \sum_{i=1}^{K}f\left(\frac{\pi_{\mathrm{E}}(A_i^E|X_i)}{\bar{\pi}(A_i^E|X_i)}\right),$$

where $f(x) = \log(x)\mathbb{1}_{\{x \geq 1\}} + (t-1)\mathbb{1}_{\{x \leq 1\}}$. The last inequality above holds because $f(x) \geq \log(x)$ for all $x \geq 0$. At this point, following the approach of Rohatgi et al. (2025, Theorem 4.2), we can apply Freedman's inequality, noticing that

$$\max_{x,a \in \mathcal{X} \times \mathcal{A}} \left| f\left(\frac{\pi_{\mathrm{E}}(a|x)}{\bar{\pi}(a|x)}\right) \right| \leq 1 + \log B_{\mathrm{ratio}},$$

to obtain that with probability at least $1 - \delta$

$$\sum_{i=1}^{K} f\left(\frac{\pi_{\mathrm{E}}(A_i^E|X_i)}{\bar{\pi}(A_i^E|X_i)}\right) \leq \sum_{i=1}^{K} \mathbb{E}\left[ f\left(\frac{\pi_{\mathrm{E}}(A_i^E|X_i)}{\bar{\pi}(A_i^E|X_i)}\right) \Big| \mathcal{F}_i^E \right] + \frac{\sum_{i=1}^{K} \mathbb{E}\left[ \left( f\left(\frac{\pi_{\mathrm{E}}(A_i^E|X_i)}{\bar{\pi}(A_i^E|X_i)}\right) \right)^2 \Big| \mathcal{F}_i^E \right]}{1 + \log B_{\mathrm{ratio}}}$$
$$+ (1 + \log B_{\mathrm{ratio}}) \log \frac{1}{\delta}. \tag{11}$$

Moreover, we can prove that

$$\mathbb{E}\left[ e^{-f\left(\frac{\pi_{\mathrm{E}}(A_i^E|X_i)}{\bar{\pi}(A_i^E|X_i)}\right)} \Big| \mathcal{F}_i^E \right] \leq \mathbb{E}\left[ e^{-\log\left(\frac{\pi_{\mathrm{E}}(A_i^E|X_i)}{\bar{\pi}(A_i^E|X_i)}\right)} \Big| \mathcal{F}_i^E \right]$$
$$= \mathbb{E}\left[ \frac{\bar{\pi}(A_i^E|X_i)}{\pi_{\mathrm{E}}(A_i^E|X_i)} \Big| \mathcal{F}_i^E \right]$$
$$= \mathbb{E}_{X_i|\mathcal{F}_i^E} \mathbb{E}_{A_i^E \sim \pi_{\mathrm{E}}(\cdot|X_i)} \left[ \frac{\bar{\pi}(A_i^E|X_i)}{\pi_{\mathrm{E}}(A_i^E|X_i)} \right]$$
$$= 1.$$

Therefore, applying Rohatgi et al. (2025, Lemma F.4) with $\eta = 1$, we obtain that

$$\sum_{i=1}^{K} \mathbb{E}\left[ \left( f\left(\frac{\pi_{\mathrm{E}}(A_i^E|X_i)}{\bar{\pi}(A_i^E|X_i)}\right) \right)^2 \Big| \mathcal{F}_i^E \right] \leq 4(2 + \log B_{\mathrm{ratio}}) \sum_{i=1}^{K} \mathbb{E}\left[ f\left(\frac{\pi_{\mathrm{E}}(A_i^E|X_i)}{\bar{\pi}(A_i^E|X_i)}\right) \Big| \mathcal{F}_i^E \right].$$

Therefore, replacing in (11), we obtain

$$\sum_{i=1}^{K} f\left(\frac{\pi_{\mathrm{E}}(A_i^E|X_i)}{\bar{\pi}(A_i^E|X_i)}\right) \leq 9 \sum_{i=1}^{K} \mathbb{E}\left[ f\left(\frac{\pi_{\mathrm{E}}(A_i^E|X_i)}{\bar{\pi}(A_i^E|X_i)}\right) \Big| \mathcal{F}_i^E \right] + (1 + \log B_{\mathrm{ratio}}) \log \frac{1}{\delta}.$$

Finally, invoking Rohatgi et al. (2025, Lemma F.5), we obtain

$$\sum_{i=1}^{K} f\left(\frac{\pi_{\mathrm{E}}(A_i^E|X_i)}{\bar{\pi}(A_i^E|X_i)}\right) \leq 9(4 + \log B_{\mathrm{ratio}}) \sum_{i=1}^{K} \mathbb{E}\left[ D_{\mathrm{Hel}}^2\left(\pi_{\mathrm{E}}(A_i^E|X_i), \bar{\pi}(A_i^E|X_i)\right) \Big| \mathcal{F}_i^E \right] + (1 + \log B_{\mathrm{ratio}}) \log \frac{1}{\delta}$$
$$\leq 9(4 + \log B_{\mathrm{ratio}}) \frac{2(A_{\max} - 1)}{1 + (A_{\max} - 1)\exp(\eta H)} K + (1 + \log B_{\mathrm{ratio}}) \log \frac{1}{\delta}$$
$$\leq 9(4 + \log B_{\mathrm{ratio}}) \frac{2}{\exp(\eta H)} K + (1 + \log B_{\mathrm{ratio}}) \log \frac{1}{\delta}.$$

Finally, using the fact that $\log B_{\mathrm{ratio}} \leq \log(1 + (A_{\max} - 1)\exp(\eta H)) \leq 2\log A_{\max} + 2\eta H$, we get

$$\sum_{i=1}^{K} \mathbb{E}\left[ D_{\mathrm{Hel}}^2(\pi_{\mathrm{E}}(\cdot|X_i), \pi_{\mathrm{MLE}}(\cdot|X_i)) | \mathcal{F}_i^E \right] \leq 2\epsilon K + 2\log \frac{|\mathcal{C}_\epsilon(\log \Pi_{\mathrm{softlin}})|}{\delta} + \frac{144(\log A_{\max} + \eta H)}{\exp(\eta H)} K$$
$$+ 4(\log A_{\max} + \eta H) \log \frac{1}{\delta}.$$

$\square$

