# OpenReview forum: "Multi-agent imitation learning with function approximation: linear Markov games and beyond"
_ICML.cc/2026/Conference — ICML 2026 regular_

### Official Review · Reviewer_DGuw · 2026-03-04

**Soundness:** 3
**Presentation:** 2
**Significance:** 3
**Originality:** 3
**Overall Recommendation:** 4
**Confidence:** 3

**Summary:**

This work presents a novel theoretical analysis of multi-agent imitation learning (MAIL) with linear transition dynamics and reward functions. They further provide a novel interactive MAIL and achieve better performance than BC in some games.

**Compliance With Llm Reviewing Policy:**

Affirmed.

**Final Justification:**

The authors addressed my concerns.

**Key Questions For Authors:**

Please see the weaknesses.

**Limitations:**

Yes

**Strengths And Weaknesses:**

Strengths:

1. The proposed interactive MAIL is efficient and achieves better performance than BC in some games.
2. They demonstrate that by using the proposed structure, all policy bias concentration coefficients at the state-action level can be replaced with concentration coefficients defined at the feature level.

Weaknesses:
1. It seems that the proposed MAIL is not only related to imitation learning but also exploratory learning. The reviewer is confused about how to implement exploratory learning solely with the given dataset since the exploratory policies may lead to unseen actions that cannot be evaluated.

2. Can the proposed theoretical results be extended to a general non-linear Markov setting? The linear setting is only a very special setting in Markov game settings. The reviewer suggests that the authors provide more discussion on the general use of the achieved results.

---

> ### Author Rebuttal · Authors · 2026-03-27
>
> We thank the reviewer for taking the time to read our paper and for their valuable feedback that helps us  improve the paper.
>
> **Exploration phase**
>
> Note that our algorithm uses reward-free RL techniques in an expert induced MDP. This means that one agent is fixed to the expert policy and the other policy is the exploratory policy. At every state visited during exploration, we query the expert for their action at that state. These expert actions are what form the imitation dataset. In other words, exploration determines which states to visit, while the expert provides the *correct actions* in those states. The exploratory agent's own actions are discarded after data collection while the expert's responses are kept for the subsequent behavioral cloning phase. Therefore, in our setup there is never a state with unknown good action during the learning phase. However, the interesting learning problem is to output a policy which is an approximate Nash equilibrium and does not require expert queries at deployment time. We hope that this clarifies the reviewer's question, and we are happy to answer additional questions that the reviewer may have.
>
> **Generalization to non-linear setting**
>
> From an applied point of view, note that we have provided an extension of our algorithm to the function approximation setting. From the theoretical side, we believe that similar ideas carry over to the general function approximation setting in Markov games. In particular, one can combine our provided ideas with theoretical results derived in [1]. We leave this exciting direction open for future works. However, this extension would unfortunately come at the price of losing any hope for a computationally efficient algorithm. We preferred to focus on the linear setting but paying attention to derive a fully efficient algorithm.
>
> We thank the reviewer again for their insightful feedback and are happy to address any further comments.
>
>  [1] Yuheng Zhang, Yu Bai, Nan Jiang, Offline Learning in Markov Games with General Function Approximation

---

> > ### Author Rebuttal · Reviewer_DGuw · 2026-04-03
> >
> > I appreciate the author's efforts to address the concerns.

---

> > > ### Author Response · Authors · 2026-04-04
> > >
> > > We thank the reviewer for confirming that our rebuttal has fully resolved their concerns, and for their positive evaluation of our work.

---

### Official Review · Reviewer_j1Sz · 2026-03-09

**Soundness:** 3
**Presentation:** 3
**Significance:** 2
**Originality:** 2
**Overall Recommendation:** 4
**Confidence:** 1

**Summary:**

This paper presents the first theoretical analysis of multi-agent imitation learning (MAIL) in linear Markov games, based on a concentrability coefficient defined at the feature level. The authors further introduce the first computationally efficient interactive MAIL algorithm for linear Markov games and prove that its sample complexity depends only on the feature dimension. They also propose a deep interactive MAIL algorithm and demonstrate its empirical performance on Tic-Tac-Toe and Connect4.

**Compliance With Llm Reviewing Policy:**

Affirmed.

**Final Justification:**

In the non-interactive setting, this paper defines a feature-level concentrability coefficient and provides the sample complexity guarantee. In the interactive setting, the paper proposes an interactive MAIL algorithm for linear Markov games and shows that its sample complexity depends only on the feature dimension. As noted in my review, I did not fully understand the overall value of the theoretical results and techniques in this seutp. The rebuttal kindly addresses this point.

**Key Questions For Authors:**

Could you provide practical motivations for multi-agent imitation learning? I am familiar with practical motivations for single-agent imitation learning, but not for the multi-agent case.

In Section 3, could you elaborate on the statement, ``we assume the expert dataset D_E is generated by a Nash equilibrium which can be recovered as the limit of the Nash equilibrium in a regularized game"?

Could authors elaborate more on the technical obstacles in the analysis and the novel aspects of your algorithm compared to prior work?

**Limitations:**

Yes

**Strengths And Weaknesses:**

First, in the non-interactive setting, this paper defines a feature-level concentrability coefficient and provides the sample complexity guarantee based on this coefficient. It also explains why this feature-level coefficient can be more favorable than the state-action-level coefficient used in prior work. In the interactive setting, the paper proposes an interactive MAIL algorithm for linear Markov games and shows that its sample complexity depends only on the feature dimension. These theoretical results are supported by experiments, and the multi-agent case is also discussed in the appendix.

Regarding the novelty of this paper, I may be missing some background, but I am not fully convinced. The work appears to be a natural extension of prior work, particularly Freihaut et al. (2025b). Although the authors emphasize the differences between the tabular and this settings, my understanding is that the technical tools needed to address these issues have already been actively studied in the single-agent setting. Therefore, while I believe this paper is valuable, I am not sure whether its level of novelty meets the bar for this conference.

---

> ### Author Rebuttal · Authors · 2026-03-27
>
> We thank the reviewer for taking the time to read our paper and for the insightful feedback. We address the concerns that the reviewer raised in detail below.
>
> **Novelty**
>
> Our algorithmic developments require a considerable amount of new techniques that we list in the following:
>
> - A change of measure at features level which exploits the fact that the occupancy measure is linear in the matrix M (see Lemma D.6).
> - A novel way to handle the total variation error of the BC policy which (i) makes sure that only the dimension $d$ appears and not the state-action cardinality and (ii) allows for the fact that the expert policy might not be exactly in $\Pi\_{\mathrm{softlin}}$ (see proof of Theorem D.5 pages 20-21).
> - For the interactive setting, we interpret the reward-free phase as regret minimization (see Lemma E.1) with time-changing rewards chosen as $\Vert\phi(x,a)\Vert\_{(\Lambda^{-n,k}\_h)^{-1}}$. This view was not used by Freihaut et al. (2025b). This new technique allows for (i) an easier-to-implement algorithm and (ii) an extension to the infinite horizon, which also requires noting that the proof by Moulin et al. (2025b) also applies to quadratic rewards. In contrast, their original analysis requires linear rewards. The infinite horizon extension was left as an open question by Freihaut et al. (2025b).
>
> - Another important innovation in the interactive setting is to control the total variation not in the fixed design iid dataset setting but for the case of an adaptively chosen dataset. This is done in Lemma J.4 and it is then applied in the proof of Lemma 4.5 (see page 28).
>
> - Finally, note that the new algorithmic idea allows for a natural deep RL extension (Explore-DQN-BC), which was not possible with previous approaches.
>
> **Expert limit assumption**
>
> We thank the reviewer for this question and are happy to provide further details on Assumption 3.1. First, note that we provide a detailed discussion of this assumption in Appendix D. This assumption ensures that the expert Nash equilibrium can be approximately realized by our softmax linear policy class​. In Appendix D (Lemma D.4), we show this is satisfied by Quantal Response Equilibria, which cover all non-isolated Nash equilibria of the game. We consider this assumption mild as analogous expert realizability assumptions are standard even in single-agent imitation learning (see, e.g., [3, Appendix C.1.2]). Additionally, note that we discuss a relaxation of this assumption as an interesting direction for future works (see Appendix C). A potential promising approach could build upon recent successful advances in the single-agent regime [4].
>
> **Practical Motivation**
>
>
> We are happy to provide further motivation for Multi-agent Imitation Learning. Potential applications of MAIL include (1) routing recommendations, where individual agents act strategically (see also [5]); (2) financial markets, where trading strategies learned from expert data must be robust against potential adversarial deviations and (3) multi-robot systems requiring coordinated, individually rational policies, and (4) in a transportation network, learning route-choice and congestion behavior from demonstrations.
>
>
> We thank the reviewer again for their insightful feedback and are happy to address any further questions or comments.
>
> [1] Freihaut, T., Viano, L., Nevali, E., Cevher, V., Geist, M., and Ramponi, G. Rate optimal learning of equilibria from data.
>
> [2] Jin, C., Krishnamurthy, A., Simchowitz, M., and Yu, T. Reward-free exploration for reinforcement learning.
>
> [3] Foster, D. J., Block, A., and Misra, D. Is behavior cloning all you need? understanding horizon in imitation learning.
>
> [4] Moulin, A., Neu, G., and Viano, L. Inverse q-learning done right: Offline imitation learning in qπ-realizable mdps,
>
> [5] Tang, J., Swamy, G., Fang, F., and Wu, Z. S. Multi-agent imitation learning: Value is easy, regret is hard.

---

> > ### Author Rebuttal · Reviewer_j1Sz · 2026-04-03
> >
> > Thank you for the clarifications. I have improved my score, but honestly, due to my limited background, I am not fully confident in my judgment and therefore have kept my confidence level unchanged.

---

> > > ### Author Response · Authors · 2026-04-04
> > >
> > > We thank the reviewer for raising their score and the confirmation that our rebuttal has addressed their questions.

---

### Official Review · Reviewer_AgEc · 2026-03-12

**Soundness:** 3
**Presentation:** 3
**Significance:** 4
**Originality:** 4
**Overall Recommendation:** 4
**Confidence:** 4

**Summary:**

This paper studies multi-agent imitation learning in Markov games to learn low-exploitability policies from expert equilibrium behavior. It extends prior tabular analyses to linear Markov games by showing that, under a suitable feature mapping, the non-interactive behavior cloning guarantee can be characterized by a feature-based hardness quantity instead of a tabular state-action one. It then gives an interactive algorithm that uses the feature mapping to guide exploration, constructs a dataset for behavior cloning, and obtains sample complexity that depends on feature dimension rather than the number of states. Beyond the linear theory, the paper also proposes a deep variant inspired by the same explore-then-clone idea. Empirical results on a small zero-sum Gridworld and on board-game environments are used to illustrate the effect of the proposed methods.

**Compliance With Llm Reviewing Policy:**

Affirmed.

**Final Justification:**

Overall, I maintain my weak accept recommendation. I find the paper technically solid, original, and significant, especially in its theoretical treatment of multi-agent imitation learning under linear function approximation. My main concerns were about the practical assumption of having a suitable feature mapping and the relatively limited support for the deep extension, but the authors’ rebuttal addressed these points reasonably well and clarified the scope of their claims. While these limitations still somewhat constrain the paper’s practical impact, they do not outweigh the strength of the core contribution, so my final assessment remains unchanged.

**Key Questions For Authors:**

1. Can the authors discuss how one might obtain a suitable feature mapping for a given Markov game in practice?

2. In Fig. 1(b), the gap between Relational LSVI-UCB-ZERO-BC and Tabular LSVI-UCB-ZERO-BC appears modest, even though their feature dimensions differ substantially (80 vs. 288). Given the theory’s dependence on feature dimension, why is the empirical advantage not larger?

**Limitations:**

yes

**Strengths And Weaknesses:**

**Strengths**

1. The paper provides a theoretical insight for non-interactive MAIL under function approximation: in linear Markov games, the BC upper bound depends on a feature-level concentrability coefficient $C_{\varphi,max⁡}$, which can be substantially smaller than the tabular state-action coefficient $C_{max}$ when the representation enables meaningful generalization across states. This sharpens our understanding of how representation quality affects the hardness of low-exploitability imitation learning.

2. The interactive result is technically strong and clearly motivated. Under a suitable feature mapping, the proposed method uses feature-guided exploration to construct a BC dataset, with a sample complexity that scales with feature dimension rather than state-space size.

**Weakness**

1. The claimed interactive efficiency gain (Theorem 4.1) relies on knowing a suitable feature mapping in advance. In practice, such a feature mapping is typically not available a priori or for free, and may require representation learning or additional computation to obtain. A more complete comparison would account for the total cost of the approach: first obtaining an appropriate feature mapping, and then using it to guide exploration. Therefore, the paper should be more cautious in presenting this advantage.

2. The deep extension is not supported as strongly as the main linear theory. It comes without a corresponding theoretical guarantee, and the empirical evaluation is limited, consisting of only two relatively small and highly structured board-game environments, with comparisons mainly against deep BC.

---

> ### Author Rebuttal · Authors · 2026-03-27
>
> We thank the reviewer for their insightful and positive feedback.
>
> **Feature map in practice**
>
> We agree that in some practical settings it is not easy to construct a meaningful feature map. However, assuming access to a feature map is standard in RL theory (MDPs [1], Markov Games [2]). Moreover, this motivates our extension to the deep setting, where features are given by the last layer of the critic and updated during exploration (e.g., via DQN), removing the need for pre-training. This may also be of independent interest for exploration in deep RL.
>
> A work that tackles the problem of learning features among those in a given feature class is, e.g., [3] While we can envision adopting similar techniques in our setting, we note that this algorithm guarantees scaling with the size of the feature class, and requires a perfect maximization oracle over it. When the feature class is a non-convex one, like in our neural network case, the algorithm becomes non-implementable. Therefore, we think that a considerable amount of fresh ideas are needed to tackle the problem of feature learning. Thank you for this insightful comment. We will make sure to mention this as an exciting future direction.
>
> **Fair comparison**
>
> The main advantage of interactive MAIL is to avoid the dependence on $\mathcal{C}\_{\phi, \max}$ compared to the non-interactive setting and both results rely on knowing the feature map. This implies, even when accounting for the additional cost of learning the features, that interactive MAIL is able to minimize the Nash gap even if $\mathcal{C}\_{\phi,\max} = \infty$, which is most likely the case for Connect4 where the number of states is more than 4 trillion and BC is unable to learn with the same feature map as our proposed interactive algorithm. Moreover, we note that BC imposes stronger conditions on the features (requires bounded $\mathcal{C}\_{\phi,\max}$). Therefore, learning good features for BC is expected to be more difficult than for LSVI-UCB-ZERO-BC.
>
> **Theoretical guarantees for the non-linear setting**
>
> It is notoriously hard to prove results for non-linear function approximation. Moreover, all analysis in the literature leads to computationally inefficient algorithms. We decided to focus on the linear case for which we derive an efficient algorithm. Moreover, as our Explore DQN demonstrates, a rigorous theoretical understanding of the linear case can provide effective guidelines also when neural network function approximation is needed.
>
> **Comparison against other algorithms**
>
> The number of MAIL algorithms, especially in the competitive setting, is limited, and, to our knowledge, no implementable interactive MAIL method exists. MAGAIL [4] relies on strong assumptions (e.g., unique Nash) and provides no guarantees for minimizing the Nash gap. We additionally evaluate it in Tic-Tac-Toe:
>
> | Dataset Size | DQN-Explore-BC |Deep BC | MAGAIL |
> |:---:|:---:|:---:|:---:|
> | 10 | 0.824 ± 0.203 | 0.154 ± 0.004 | 0.713 ± 0.001 |
> | 50 | 0.510 ± 0.365 | 0.150 ± 0.004 | 0.713 ± 0.002 |
> | 100 | 0.375 ± 0.378 | 0.155 ± 0.005 | 0.713 ± 0.001 |
> | 250 | 0.191 ± 0.336 | 0.148 ± 0.010 | 0.714 ± 0.000 |
> | 500 | 0.002 ± 0.003 | 0.150 ± 0.011 | 0.714 ± 0.002 |
> | 1000 | 0.000 ± 0.000 | 0.144 ± 0.012 | 0.713 ± 0.001 |
>
> As the algorithm is a non-interactive MAIL algorithm, it is expected that the algorithm is unable to minimize the Nash gap. Moreover, the poor performance of MAGAIL can be explained by the fact that the strong assumptions required by MAGAIL are not satisfied in Tic-Tac-Toe.
>
> **Experimental gap of relational vs. tabular**
>
> Note that for fewer expert interactions the effect is larger compared to cases with larger data, where the number of samples dominates the bound. To more clearly isolate this effect, we design a control experiment using a randomly generated linear Markov game. We run LSVI-UCB-ZERO-BC with a fixed sample size while varying the number of states, keeping the feature dimension constant. The results are shown below:
>
> | Num states | Latent (d=20) | Tabular |
> |:---:|:---:|:---:|
> | 5   | 0.0709 ± 0.0069 | 0.0672 ± 0.0068 |
> | 50  | 0.1979 ± 0.0408 | 0.2511 ± 0.0704 |
> | 100 | 0.2131 ± 0.0244 | 0.5155 ± 0.0407 |
> | 150 | 0.2233 ± 0.0274 | 0.7743 ± 0.0605 |
> | 200 | 0.2163 ± 0.0337 | 0.8043 ± 0.0961 |
>
>
> As expected, for tabular representations the exploitability increases significantly with the number of states. In contrast, under the fixed-dimensional feature map, exploitability remains relatively constant as the state space grows.
>
> We thank the reviewer again for their insightful feedback and are happy to address any further comments.
>
> [1] Wang et al., On reward-free reinforcement learning with linear function approximation
>
> [2] Zhong et al., Pessimistic minimax value iteration: Provably efficient equilibrium learning from offline datasets.
>
> [3] Ni et al., Representation Learning for General-sum Low-rank Markov Games
>
> [4] Song et al., S. Multi-agent generative adversarial imitation learning.

---

> > ### Author Rebuttal · Reviewer_AgEc · 2026-04-02
> >
> > Thank you for the detailed rebuttal. The authors have largely addressed my main concerns. My overall assessment remains unchanged, and I am keeping my score the same.

---

> > > ### Author Response · Authors · 2026-04-04
> > >
> > > We appreciate the reviewer's confirmation that our rebuttal has largely addressed their questions, and thank them for their positive assessment of our work.

---

### Official Review · Reviewer_WTNn · 2026-03-13

**Soundness:** 4
**Presentation:** 3
**Significance:** 4
**Originality:** 3
**Overall Recommendation:** 5
**Confidence:** 3

**Summary:**

This paper analyses Multi-Agent Imitation Dynamics (MAIL) in linear Markov games, providing provably efficient algorithms in both interactive and non-interactive settings. This builds upon recent work in tabular scenarios, and the convergence is numerically confirmed and compared with a deep MAIL approach.

**Compliance With Llm Reviewing Policy:**

Affirmed.

**Ethical Review Concerns:**

The author's information may be contained in some of the supplementary material (in .git/). The author may have carelessly forgotten to clear the .git folder. Otherwise, this is a good piece of research.

**Ethical Review Flag:**

Flag this paper for an ethics review.

**Ethics Expertise Needed:**

["Privacy and Security (e.g., personally identifiable information)"]

**Final Justification:**

The author has addressed my concern and question. I will keep the current assessment.

**Key Questions For Authors:**

Can you provide some justification and intuition for the beta +1 coefficient in Algorithm 2?

**Limitations:**

yes

**Strengths And Weaknesses:**

Strengths
The paper is well-written, with both the theoretical and numerical experiments justified and analysed. The assumptions are clearly stated, and their applicability is discussed. The structure is clear, with each section and result being well-motivated. The work does well to provide theoretical results on sample complexity in these linear Markov games and extends their analysis beyond using Deep MAIL. The results are a natural extension of the previous literature, namely (Freihaut et al. 2025b), which was limited to tabular games.

Weaknesses
The paper itself would benefit from related literature, which is largely omitted and placed in the Appendix. Moreover, analysis of the experimental results could go into more detail.

---

> ### Author Rebuttal · Authors · 2026-03-27
>
> We thank the reviewer for their time in reading our work and their insightful comments that help improve the paper.
>
> **Additional details**
>
> We agree with the reviewer that the work would benefit from a related work section within the main part of the paper. We will add a summary of our extensive related work section (Appendix B) from the appendix to the main paper in a revised version. The same holds true for a discussion of the experimental results, which currently are placed in Appendix I.
>
> **"Can you provide some justification and intuition for the beta +1 coefficient in Algorithm 2?"**
>
> We are happy to include further intuition on why $\beta+1$ appears in Algorithm 2. Note that in standard UCB applications, the learner has access to the true underlying reward function $r$. In contrast, in the reward-free setting, we do not have access to the true underlying reward. Instead, we define an exploratory reward, which is the UCB bonus divided by $\beta$ and can therefore be grouped together. More formally, we have that in standard RL setting that the Q-function of the update would look like $$Q\_h^k(x, a) = \underbrace{r(x, a)}\_{\text{game reward}} + (w\_h^k)^\top \phi(x, a) + \underbrace{\beta  \|\|\phi(x, a)\|\|\_{(\Lambda\_h^k)^{-1}}}\_{\text{UCB bonus}}.$$
> Now note that the artificial exploration reward has the same form:
> $$r\_h^k(x, a) = \|\|\phi(x, a)\|\|_{(\Lambda\_h^k)^{-1}}.$$
> Plugging this into our Q-function, gives
>
> $$Q\_h^k(x, a) = \underbrace{\|\|\phi(x, a)\|\|\_{(\Lambda\_h^k)^{-1}}}\_{\text{exploratory reward}} + (w\_h^k)^\top \phi(x, a) + \underbrace{\beta \|\|\phi(x, a)\|\|\_{(\Lambda\_h^k)^{-1}}}\_{\text{UCB bonus}} =  (w\_h^k)^\top \phi(x, a) + (\beta + 1)  \|\|\phi(x, a)\|\|\_{(\Lambda\_h^k)^{-1}}. $$
>
> We hope that this clarifies the reviewer’s questions and we are happy to answer any other questions they might have.

---

> > ### Author Rebuttal · Reviewer_WTNn · 2026-04-04
> >
> > Thank you for the authors’ reply.
> >
> > The author has addressed my concern and question. I will keep the current assessment.

---

> > > ### Author Response · Authors · 2026-04-04
> > >
> > > We thank the reviewer for acknowledging that our rebuttal has fully resolved their concerns, and for their positive evaluation of our work.

---

### Decision · Program_Chairs · 2026-04-30

**Decision:**

Accept (regular)

**Comment:**

This paper presents a theoretical framework and an algorithm for multi-agent imitation learning that scales by using linear features instead of relying on the total number of states. The reviewers appreciated the solid math and clear improvements over prior work. But some of them noted that relying on known feature maps and having limited real-world experiments were minor drawbacks. The authors answered these concerns during the rebuttal. The core ideas appear valuable to the field. I recommend accepting this paper.